# Mitigating Reward Over-Optimization in RLHF via Behavior-Supported Regularization

**Juntao Dai**[12]**, Taiye Chen**[4]**, Yaodong Yang**[34*]**, Qian Zheng**[12*]**, Gang Pan**[12]

[1]College of Computer Science and Technology, Zhejiang University
[2]The State Key Lab of Brain-Machine Intelligence, Zhejiang University
[3]LLM Safety Centre, Beijing Academy of Artificial Intelligence
[4]Center for AI Safety and Governance, Peking University
{juntaodai,qianzheng,gpan}@zju.edu.com
{yeyutaihan}@stu.pku.edu.cn, {yaodong.yang}@pku.edu.cn

## Abstract

Reinforcement learning from human feedback (RLHF) is an effective method for aligning large language models (LLMs) with human values. However, reward over-optimization remains an open challenge leading to discrepancies between the performance of LLMs under the reward model and the true human objectives. A primary contributor to reward over-optimization is the extrapolation error that arises when the reward model evaluates out-of-distribution (OOD) responses. However, current methods still fail to prevent the increasing frequency of OOD response generation during the reinforcement learning (RL) process and are not effective at handling extrapolation errors from OOD responses. In this work, we propose the *Behavior-Supported Policy Optimization* (BSPO) method to mitigate the reward over-optimization issue. Specifically, we define *behavior policy* as the next token distribution of the reward training dataset to model the in-distribution (ID) region of the reward model. Building on this, we introduce the behavior-supported Bellman operator to regularize the value function, penalizing all OOD values without impacting the ID ones. Consequently, BSPO reduces the generation of OOD responses during the RL process, thereby avoiding overestimation caused by the reward model's extrapolation errors. Theoretically, we prove that BSPO guarantees a monotonic improvement of the supported policy until convergence to the optimal behavior-supported policy. Empirical results from extensive experiments show that BSPO outperforms baselines in preventing reward over-optimization due to OOD evaluation and finding the optimal ID policy.

## 1 Introduction

Reinforcement Learning from Human Feedback (RLHF) has been demonstrated as an effective method for aligning large language models (LLMs) with human values (Ouyang et al., 2022b; Bai et al., 2022; Achiam et al., 2023; Yang et al., 2023; Ji et al., 2023b). A key phase of RLHF is reward modeling, where the reward model is trained on preference datasets to approximate human preferences (Stiennon et al., 2020; Ganguli et al., 2022). The reward model is subsequently used to evaluate the responses of LLMs during the reinforcement learning (RL) phase. Despite its empirical success, RLHF is criticized for its vulnerability and instability (Casper et al., 2023). One of the open challenges in RLHF is the *reward over-optimization* issue (Gao et al., 2023; Coste et al., 2023). As shown in Figure 1(a), although the performance of LLMs may seem to improve under the reward model (*proxy reward*), it can deviate from the actual human objectives (*gold reward*).

One of the primary causes of reward over-optimization is the extrapolation error that arises when the reward model evaluates out-of-distribution (OOD) responses (Eisenstein et al., 2023; Laidlaw et al., 2024; Yang et al., 2024b). Due to the exploratory nature of RL, LLMs may generate responses that fall outside the training data distribution of the reward model. Lacking the ability to accurately assess these unseen responses, the reward model may overestimate the reward signal, leading to

---

*Corresponding author.

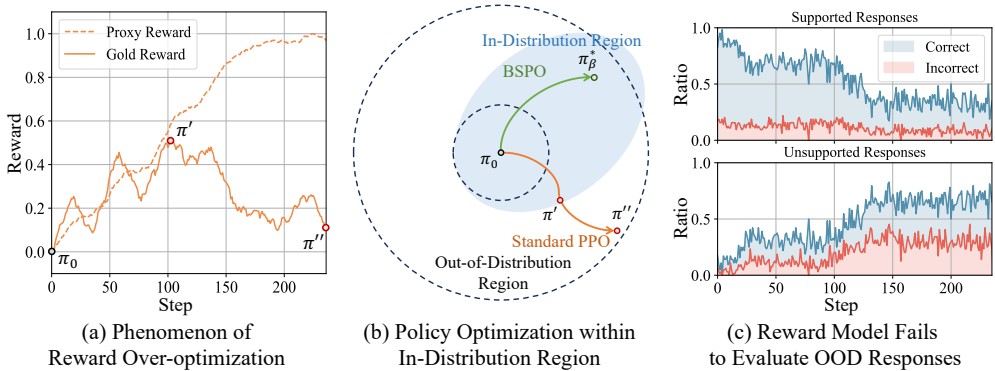

(a) Phenomenon of
Reward Over-optimization

(b) Policy Optimization within
In-Distribution Region

(c) Reward Model Fails
to Evaluate OOD Responses

Figure 1: **(a) Reward over-optimization.** Although the performance of LLMs may seem to improve under the reward model (*proxy reward*), it deviates from the actual human objectives (*gold reward*). **(b) Search in ID region.** Our algorithm guides policy iteration within the ID region of the reward model, whereas others may enter the OOD region, suffering from extrapolation errors. **(c) Hard to evaluate unsupported responses.** Responses are categorized as supported or unsupported, depending on whether they include actions unsupported by the behavior policy ($\beta(a|s) = 0$). As policy iterates, the occurrence of unsupported responses increases. "Correct/Incorrect" indicates whether the proxy model's evaluation of a generated response aligns with the gold model. The proxy model predicts preference pairs well for supported responses but struggles with unsupported ones.

severe overestimation in the value function as an incorrect policy evaluation. As the policy iteration, the frequency of OOD responses tends to increase, further amplifying the over-optimization issue.

Previous work (Gao et al., 2023) finds that enlarging the reward model and dataset may mitigate over-optimization, but it is often impractical in real-world scenarios. Thus, many others focuses on enhancing the RL process using reward regularization. A common approach is to use the Kullback-Leibler (KL) divergence as a penalty(Ouyang et al., 2022b; Touvron et al., 2023). This method limits the policy updates to remain close to the initial policy (Gao et al., 2023), such as in the dotted inner circle in Figure 1(b). However, due to the insensitivity to the in-distribution (ID) region of the reward model, it is difficult to determine an appropriate penalty strength, and it prevents full exploration of the ID region to identify the optimal solution. Similar limitations exist in methods that introduce a maximum reward constraint (Moskovitz et al., 2023) Other approaches include using uncertainty quantifiers (Zhang et al., 2024b) or ensembles (Coste et al., 2023). Yet, uncertainty quantifiers may hard to generalize to OOD regions (Nalisnick et al., 2018) while ensemble method may suffer from consistent overestimation across ensembles and higher computational costs (Eisenstein et al., 2023). Furthermore, a problem across above methods is that while they handle OOD responses pessimistically, they also impact the evaluation of ID ones, potentially leading to suboptimal solutions.

In this work, we propose the *Behavior-Supported Policy Optimization* (BSPO), a novel approach to address reward over-optimization. The core idea is to use a value regularization to guide policy iteration only within the in-distribution (ID) region, as shown in Figure 1(b). Specifically, we use the distribution of the next token to characterize the action distribution in the reward training dataset. Inspired by offline RL (Levine et al., 2020), we refer to this distribution as the *behavior policy*. Due to the auto-regressive nature of LLMs, any OOD action results in the accumulation of extrapolation errors within unsupported responses, which invalidate the reward model, as shown in Figure 1(c). Then, we introduce a behavior-supported Bellman operator to regularize the value function. The regularized value function derived from this operator penalizes all OOD values without affecting the ID ones. Our method leverages this value regularization to reduce OOD generation during RL, thereby preventing overestimation caused by the extrapolation errors of reward prediction. Theoretically, BSPO guarantees monotonic improvement of the supported policy until convergence to the optimal behavior-supported policy, while other methods lack this convergence guarantee.

We summarize our primary contributions as three folds:

- We propose a novel method that leverages the next-token distribution to characterize the reward training dataset, enabling the detection of whether a response is OOD for the reward model.

- We introduce the BSPO algorithm, the first method that uses value regularization to address reward over-optimization. It penalizes OOD values without affecting ID ones, thereby theoretically achieving the same solution as standard policy evaluation.

- We provide a lightweight implementation of BSPO using the *ScoreLM* model to predict rewards and behavior distributions. Through extensive experiments, we empirically show that BSPO outperforms baselines in avoiding reward over-optimization and finding the optimal ID policy.

## 2 BACKGROUND

**Token-level MDP** When an LLM is modeled as an RL agent, its actions and states can be represented by tokens or token sequences. Thus, we formalize language generation tasks as a token-level MDP $\mathcal{M} \doteq (\mathcal{S}, \mathcal{A}, T, \mathcal{X}, \mu, r, \gamma)$. Given the vocabulary $\mathcal{A}$ of an LLM, it also represents the action set in the MDP, where an action $a$ corresponds to generating a token of the vocabulary. $\mathcal{X}$ is the set of prompts, $\mu$ is the distribution of the prompt $x$ given to the LLM, and $\mathcal{S}$ is the set of states, where each state $s$ is composed of a prompt $x \in \mathcal{X}$ and a sequence of generated tokens $a_{0:t-1} = \bigcup_{i=0}^{t-1} a_i$. Specifically, $s \doteq x \cup a_{0:t-1}$. Thus, the state transition is defined as $T(x \cup a_{0:t-1}, a_t) = x \cup a_{0:t}$. The $r$ denotes the reward function and the $\gamma$ is the discount factor. In this framework, a stationary policy, $\pi$, is a probability distribution indicating the likelihood of the next token $a_t$ given state $s_t \doteq x \cup a_{0:t-1}$ at step $t$. Then, $\Pi$ denotes the set of all stationary policies.

The goal of reinforcement learning is to maximize a performance measure, $\mathcal{J}(\pi)$, which is typically defined as an infinite horizon discounted total return, $\mathcal{J}(\pi) \doteq \mathbb{E}_{\tau \sim \pi} [\sum_{t=0}^{\infty} \gamma^t r_t]$. Here, $\tau \doteq \{s_t, a_t, r_t\}_{t=0}^{\infty} \sim \pi$ denotes the distribution over trajectories generated by $\pi$, where $s_0 = x \sim \mu$, $a_t \sim \pi(\cdot \mid s_t)$, $s_{t+1} = T(s_t, a_t)$, and $r_t = r(s_t, a_t)$. We express the state value function of $\pi$ as $V^\pi(s) \doteq \mathbb{E}_{\tau \sim \pi} [\sum_{t=0}^{\infty} \gamma^t r_t \mid s_0 = s]$ and the state-action value function as $Q^\pi(s, a) \doteq \mathbb{E}_{\tau \sim \pi} [\sum_{t=0}^{\infty} \gamma^t r_t \mid s_0 = s, a_0 = a]$. The advantage function is $A^\pi(s, a) = Q^\pi(s, a) - V^\pi(s)$.

**Preference Modelling** The RLHF method improves the quality of LLM responses by leveraging human preference data through a reward model (Ouyang et al., 2022a; Bai et al., 2022). The reward model denoted as $R(x, y)$ is designed to align with human preferences, where $x$ represents the input prompt and $y$ is the corresponding response generated by the LLM. Human preferences are captured as pairs of responses, symbolized as $y_w \succ y_l | x$, where $y_w$ (win) denotes a response more preferred by humans than $y_l$ (lose). Using the Bradley-Terry model (Bradley & Terry, 1952), the likelihood of a preference pair can be estimated as $p(y_w \succ y_l | x) = \sigma(R(x, y_w) - R(x, y_l))$ where $\sigma(x) = \frac{1}{1+e^{-x}}$ is the logistic sigmoid function. Consequently, given the preference dataset $\mathcal{D} = \{x^i, y_w^i, y_l^i\}_{i=1}^{N}$, the reward model is trained by minimizing the negative log-likelihood loss, $\mathcal{L}(\phi, \mathcal{D}) = -\mathbb{E}_{(x, y_w, y_l) \sim \mathcal{D}}[\log \sigma(R_\phi(x, y_w) - R_\phi(x, y_l))]$. In the context of LLMs, the reward model typically includes a linear layer after the final transformer layer. During the RL stage, the reward model assigns a reward to the final token of the sequence, which is typically the EOS token.

**Related Work of Reward Over-Optimization** This phenomenon, where the policy language model exploits imperfects in the reward model, is commonly known as *reward over-optimization* (Gao et al., 2023), and is also called *reward hacking* (Amodei et al., 2016; Skalse et al., 2022) or *reward gaming* (Pang et al., 2023). Given the high cost of evaluations for studying reward over-optimization, most studies adopt the synthetic setup that employs a powerful gold model to substitute for human labeling and evaluation (Gao et al., 2023; Moskovitz et al., 2023; Coste et al., 2023).

The work most closely related to ours includes a series of reward regularization methods, such as adding a KL penalty to the reward (Kullback & Leibler, 1951) or utilizing a reward ensemble (Coste et al., 2023). KL penalty method uses a per-token KL penalty from the SFT model to mitigate over-optimization of the reward model, which is first proposed by Stiennon et al. (2022) and widely employed in the context of RLHF (Ouyang et al., 2022b; Bai et al., 2022). Similarly, building on the concept of early stopping, Moskovitz et al. (2023) introduces an approach called Constrained PPO to prevent the policy from surpassing each reward model's threshold of usefulness while minimizing KL divergence in the token distribution. However, due to insensitivity to the reward model's in-distribution (ID) region, these methods conservatively iterate near the initial model. In contrast, our algorithm leverages the behavior policy $\beta$, allowing relaxed distance constraints and broader exploration of regions where valid reward model predictions exist.

On the other hand, previous work in machine learning demonstrates that combining multiple estimators can enhance robustness (Du & Swamy, 2019). Coste et al. (2023) train several proxy models and develop ensemble-based conservative optimization methods, such as using variance for uncertainty-weighted optimization (UWO) and applying the worst reward for conservative worst-case optimization (WCO). Ramé et al. (2024) propose a more efficient ensemble-based approach by leveraging weight-averaged reward models. Zhang et al. (2024b) introduce a lightweight uncertainty-weighted optimization method that quantifies uncertainties of rewards by utilizing only the last layer embeddings of the reward model. However, without additional information about OOD responses, reward model ensembles mitigate but do not eliminate reward hacking (Nalisnick et al., 2018; Eisenstein et al., 2023). All ensemble proxy models may exhibit consistent error patterns in the OOD region.

Another route to mitigating reward over-optimization is by training more generalized and robust reward models (Wang et al., 2024a;b; Shen et al., 2024). Gao et al. (2023) demonstrated in a synthetic setup that increasing the reward model size and the training data volume can mitigate this over-optimization issue. Notably, methods aimed at enhancing the generalization at the reward modeling stage run parallel to our approach and can be combined. In our implementation, the training of the reward model incorporates the hidden state regularization technique (Yang et al., 2024b).

## 3 ANALYSIS: BEHAVIOR-SUPPORTED METHOD

### 3.1 BEHAVIOR POLICY FOR OOD DETECTION OF REWARD PREDICTION

Since reward learning is inherently data-driven, it faces significant challenges in evaluating responses that lie outside the distribution of the training dataset. However, current OOD detection techniques (Yang et al., 2024a) require additional information for the OOD part (Nalisnick et al., 2018), and are not applicable to reward prediction in LLM. Therefore, we introduce an OOD detection approach based on the power of well-pretrained LLMs for next-token prediction.

Based on the token-level MDP modeling in RL for LLMs, we use the distribution of the next token, $\beta : \mathcal{S} \times \mathcal{A} \to [0, 1]$, to characterize the action distribution in the preference dataset. Inspired by offline RL (Levine et al., 2020; Wu et al., 2022), we refer to this distribution as the *behavior policy*. The behavior policy divides actions at a given state $s$ into two categories, defined as follows:

**Definition 1** (Behavior-Supported Action). *In a given state $s$, an action $a$ is considered supported by the behavior distribution $\beta$ if and only if $\beta(a|s) > 0$.*

Due to the auto-regressive nature of LLMs, we believe that any OOD action (i.e., one not a behavior-supported action) leads to the accumulation of extrapolation errors in the resulting unsupported responses. To validate this hypothesis, we collect all responses generated during the RLHF process under the same experimental settings of the experiment section. These responses are categorized into supported responses and unsupported responses, depending on whether they contain behavior-unsupported actions (i.e., $\beta(a|s) = 0$). These responses, along with those generated by the initial model given the same prompt, construct comparison pairs. We use these comparison pairs and the gold model as ground truth to test whether the proxy model can correctly predict preferences.

Figure 1(c) shows that as the policy iterates, the proportion of unsupported responses generated by the LLM increases. For comparison pairs consisting of supported responses, the proxy model accurately predicts preferences, achieving an average accuracy of 75.91%, which is comparable to its performance on the test set (Figure 2(b)). However, for comparison pairs composed of unsupported responses, the proxy model's predictive accuracy drops significantly to 58.10%. These results indicate whether the response contains only behavior-supported actions is an effective indicator for measuring whether it is OOD for reward prediction. More results are provided in Appendix D.1.

### 3.2 BEHAVIOR-SUPPORTED REGULARIZATION

A natural idea to mitigate reward over-optimization during the RL is to avoid taking actions that are not supported by the behavior policy. In this paper, we consider applying regularization to the value function in order to reduce the ranking of actions that are not supported by the behavior distribution during policy evaluation. Specifically, we define $r_{\min} \doteq \min_{s \in \mathcal{S}, a \in \mathcal{A}} [r(s, a)]$ and

$Q_{\min} \doteq \sum_{t=0}^{\infty} \gamma^t r_{\min} = \frac{r_{\min}}{1-\gamma}$. Then, we propose the behavior-supported Bellman operator:

$$\mathcal{T}_\beta^\pi Q(s,a) \doteq \begin{cases} \mathcal{T}^\pi Q(s,a), & \text{if} \quad \beta(a|s) > 0, \\ Q_{\min}, & \text{otherwise.} \end{cases} \tag{1}$$

where $\mathcal{T}^\pi Q(s,a) \doteq r(s,a) + \gamma \mathbb{E}_{a' \sim \pi(\cdot|s')} [Q(s',a')]$ represents the standard Bellman operator, and $s' = T(s,a)$ is the next state. This new operator has the following properties:

**Theorem 1** (Contraction of $\mathcal{T}_\beta^\pi$). *For any policy $\pi \in \Pi$, the operator $\mathcal{T}_\beta^\pi$ is $\gamma$-contraction with respect to the $\mathcal{L}_\infty$ norm over the space $\mathcal{S} \times \mathcal{A}$.*

The proofs are provided in the Appendix A. The $\gamma$-contraction property of the operator $\mathcal{T}_\beta^\pi$ guarantees that, for any initial Q-values, iterative application of $\mathcal{T}_\beta^\pi$ converges to a unique fixed point at a rate of $\gamma$. Thus, we find this fixed point by iteratively solving the problem, $\min_Q \mathbb{E}_{\tau \sim \pi}(Q(s,a) - \mathcal{T}_\beta^\pi Q(s,a))^2$, and use it for policy evaluation to optimize the policy $\pi$.

Intuitively, an observation from Equation 1 is that, at each iteration $k$, $\mathcal{T}_\beta^\pi$ yields the same $Q^{k+1}(s,a)$ as the standard operator $\mathcal{T}^\pi$ for any behavior-supported actions (i.e., ID actions). However, for actions that are not supported (i.e., OOD actions), $\mathcal{T}_\beta^\pi$ yields smaller $Q^{k+1}(s,a)$. Through this iterative process, the convergence to a fixed point ensures the validity of the following theorem:

**Theorem 2** (Fixed Points). *For any policy $\pi \in \Pi$, the fixed point $Q_\beta^\pi$ of $\mathcal{T}_\beta^\pi$ satisfies*

$$\begin{cases} Q_{min} \le Q_\beta^\pi(s,a) \le Q^\pi(s,a), & \text{if} \quad \beta(a|s) > 0, \\ Q_\beta^\pi(s,a) = Q_{min}, & \text{otherwise.} \end{cases} \tag{2}$$

*where $Q^\pi(s,a)$ is fixed point of standard Bellman operator $\mathcal{T}^\pi$.*

We refer to the fixed point $Q_\beta^\pi$ as the behavior-supported Q-value function. Theorem 2 indicates that for any policy $\pi$, using the behavior-supported Q-value function for policy evaluation underestimates the future returns of OOD actions, thereby disadvantaging these actions across the action space. This leads the policy iteration process to reinforce behavior-supported actions and weakens unsupported ones. Now, we consider a common policy iteration using behavior-supported value, as follows:

$$\pi_{k+1} = \arg\max_{\pi \in \Pi} \mathbb{E}_{\tau \sim \pi_k} \left[ \frac{\pi(a_t|s_t)}{\pi_k(a_t|s_t)} A_\beta^{\pi_k}(s_t, a_t) \right]. \tag{3}$$

where $V_\beta^{\pi_k}(s) = \mathbb{E}_{a \sim \pi_k(\cdot|s)} \left[ Q_\beta^{\pi_k}(s,a) \right]$, and $A_\beta^{\pi_k}(s,a) = Q_\beta^{\pi_k}(s,a) - V_\beta^{\pi_k}(s)$. As the policy iterates, the action selection will eventually consist only of behavior-supported actions. We define such a policy as a behavior-supported policy. Specifically, for any state $s$, the supported action space of state $s$ is denoted as $\text{supp}(\beta(\cdot|s)) \doteq \{a \in \mathcal{A} \mid \beta(a|s) > 0\}$. The set of all behavior-supported policies is defined as

$$\Pi_\beta \doteq \{\pi \in \Pi \mid \pi(a|s) = 0, \quad \forall s \in \mathcal{S}, a \notin \text{supp}(\beta(\cdot|s))\}. \tag{4}$$

Furthermore, the following corollary guarantees that the policy optimization method in Equation 3 yields behavior-supported policies $\pi \in \Pi_\beta$ through regularized policy evaluation $Q_\beta^\pi$.

**Corollary 1** (Supported Policy Optimization). *The policy optimization method mentioned in Equation 3 yields behavior-supported policy $\pi \in \Pi_\beta$.*

Theoretically, we demonstrate that policy iteration with the regularized value function guarantees that each iteration results in a behavior-supported policy. This restricts the search for the optimal policy to the ID region of the reward model. As a result, the reward model does not need to evaluate OOD responses generated with behavior-unsupported actions, thereby preventing the reward over-optimization issue caused by extrapolation errors.

Furthermore, the following corollary holds for behavior-supported policy iterations:

**Corollary 2.** *For any behavior-supported policy $\pi \in \Pi_\beta$, the fixed point $Q_\beta^\pi$ of $\mathcal{T}_\beta^\pi$ satisfies*

$$Q_\beta^\pi(s,a) = \begin{cases} Q^\pi(s,a), & \text{if} \quad \beta(a|s) > 0, \\ Q_{min}, & \text{otherwise.} \end{cases} \tag{5}$$

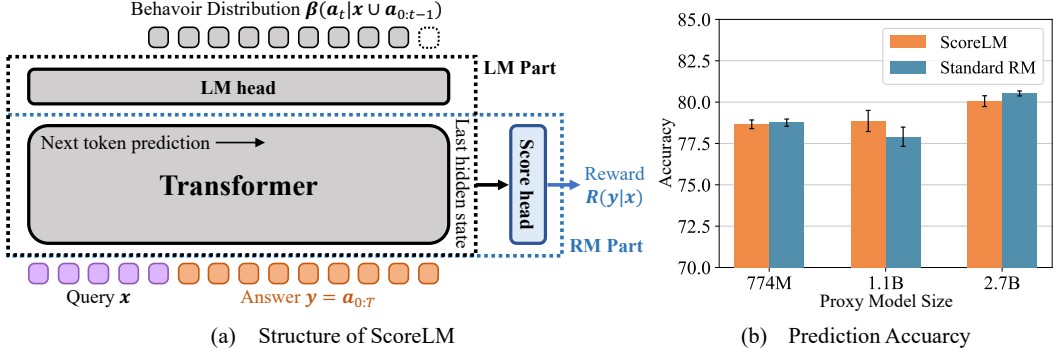

(a)    Structure of ScoreLM

(b)    Prediction Accuarcy

Figure 2: **(a) Structure of our ScoreLM model.** We retain the original language model head to predict the next-token distribution and initialize a score head to predict the reward. **(b) Compare with Standard RM.** The performance of ScoreLM is comparable to standard reward models under three scales on the test set. The short vertical lines indicate the standard deviation of four repetitions.

Corollary 2 indicates that the behavior-supported policy evaluation provides the same unbiased Q-values for all ID actions as the standard Bellman operator $\mathcal{T}^{\pi}$. Together with Corollary 1, for any state $s$, the optimal behavior-supported action is the same under the $Q_{\beta}^{\pi}$ and $Q^{\pi}$ value functions during behavior-supported policy iteration. This ensures that the method in Equation 3 converges to the optimal behavior-supported policy $\pi_{\beta}^{*}$. Specifically, the following theorem holds:

**Theorem 3** (Monotonicity to Optimality). *The behavior-supported policy optimization method mentioned in Equation 3 results in a strictly monotonic improvement of policy performance, continuing until the optimal behavior-supported policy $\pi_{\beta}^{*}$ is attained.*

**Notably**, all proofs of this section are provided in the Appendix A.

## 4    IMPLEMENTATION: BEHAVIOR-SUPPORTED POLICY OPTIMIZATION

We introduce the behavior-supported method that uses regularized Q-values to guide the policy iteration within the ID region of the reward model. In this section, we present an implementation of the *Behavior-Supported Policy Optimization* (BSPO) algorithm for the RLHF training of LLMs.

**Behavior Distribution Prediction**    Since pre-trained LLMs have already been exposed to an extensive amount of data, we can directly fit the distribution of the next token in the reward training dataset without introducing additional OOD response information. In the standard RLHF paradigm, the reward model is derived by converting the LLMs, replacing the language model head with a scalar head for scoring. In our work, we retain the original language model head, as shown in Figure 2, to predict the behavior distribution. Specifically, we train this *ScoreLM* model by minimizing the following loss function, which consists of two components:

$$\mathcal{L}_{\text{ScoreLM}}(\phi; \mathcal{D}) = -\underbrace{\mathbb{E}_{\mathcal{D}}\Big[ \log \sigma \Big( R(x, y_w; \phi) - R(x, y_l; \phi) \Big) \Big]}_{\text{Preference Loss}} -\alpha \underbrace{\mathbb{E}_{\mathcal{D}}\Big[ \log \beta(a_t | x \cup a_{0:t-1}; \phi) \Big]}_{\text{Supervised Loss}},$$

(6)

where $\alpha$ is a hyperparameter to balance the two losses. We integrate reward prediction and behavior distribution prediction into the same model for two main reasons. First, experimental results indicate that this integration has almost no impact on prediction accuracy, as shown in Figure 2(b). Additionally, the supervised learning loss helps preserve the language capabilities of the transformer, thereby improving the generalization of the reward model (Yang et al., 2024b). Second, compared to standard PPO, using ScoreLM introduces negligible additional memory and computational overhead.

**Behavior-Supported Value Function**    In our implementation of value regularization, we predict behavior-supported V-values instead of Q-values to achieve greater stability while maintaining

equivalent policy evaluation. The deterministic state transition of the token-level MDP (i.e., given a state $s = a_{0:t-1}$ and a next token $a_t$, the next state $s' = a_{0:t}$ is determined) ensures this equivalence. It is important to clarify that this determinism inherently holds due to the auto-regressive nature of next-token prediction in LLMs. Specifically, the behavior-supported Bellman V-operator is

$$\mathcal{T}_{\beta,V}^{\pi} V(x \cup a_{0:t-1}) \doteq \begin{cases} \mathcal{T}_V^{\pi} V(x \cup a_{0:t-1}), & \text{if} \quad \beta(a_{t-1}|x \cup a_{0:t-2}) > 0, \\ \frac{1}{\gamma} \left[ Q_{\min} - r(x \cup a_{0:t-2}, a_{t-1}) \right], & \text{otherwise,} \end{cases} \tag{7}$$

where $\mathcal{T}_V^{\pi} V(x \cup a_{0:t-1}) \doteq \mathbb{E}_{a_t \sim \pi(\cdot|x \cup a_{0:t-1})} \left[ r(x \cup a_{0:t-1}, a_t) + \gamma V(x \cup a_{0:t}) \right]$ is the standard Bellman operator for the V-value function. Then, the following theorem and corollary hold:

**Theorem 4** (Contraction of $\mathcal{T}_{\beta,V}^{\pi}$). *For any policy $\pi \in \Pi$, the operator $\mathcal{T}_{\beta,V}^{\pi}$ is $\gamma$-contraction with respect to the $\mathcal{L}_{\infty}$ norm over the state space $\mathcal{S}$.*

**Corollary 3** (Equivalent Policy Evaluation). *$\forall \pi \in \Pi$, denote $V_{\beta}^{\pi}$ as the fixed point of the Bellman operator $\mathcal{T}_{\beta,V}^{\pi}$ in Equation 7. Then, its corresponding state-action value function $Q_{\beta}^{\pi}(s, a) = r(s, a) + \gamma V_{\beta}^{\pi}(T(s, a))$ is equal to the fixed point of the Bellman operator $\mathcal{T}_{\beta}^{\pi}$ in Equation 1.*

The proofs of Theorem 4 and Corollary 3 are provided in the Appendix B. These results guarantee that the $\mathcal{T}_{\beta,V}^{\pi} V$ converges to a unique fixed point, $V_{\beta}^{\pi}$, which provides policy evaluation equivalent to that of the behavior-supported Q-values, $Q_{\beta}^{\pi}$. For a parameterized critic model $V_{\varphi}(x \cup a_{0:t-1})$, we train it by minimizing the loss function with $\mathcal{T}_{\beta,V}^{\pi} V_{\varphi}$ calculated from ScoreLM $R_{\phi}$ at step $k$:

$$\mathcal{L}_V(\varphi; \pi) = \mathbb{E}_{\tau \sim \pi_k} \left[ \left( V_{\varphi}(x \cup a_{0:t-1}) - \mathcal{T}_{\beta,V}^{\pi_k} V_{\varphi}(x \cup a_{0:t-1}) \right)^2 \right]. \tag{8}$$

**Behavior-Supported Policy Optimization**   We combine the behavior-supported method with the widely used Proximal Policy Optimization (PPO) (Schulman et al., 2017) algorithm. Specifically, for the parameterized LLM $\pi_{\theta}$, we optimize it by minimizing the following loss function at step $k$:

$$\mathcal{L}_{\pi}(\theta) = -\mathbb{E}_{\tau \sim \pi_{\theta_k}} \left[ \min \left( \rho_t(\theta) A^{\pi_{\theta_k}}(s_t, a_t), \text{clip}(\rho_t(\theta), 1 - \epsilon, 1 + \epsilon) A^{\pi_{\theta_k}}(s_t, a_t) \right) \right] \tag{9}$$

where $\rho_t(\theta) = \frac{\pi_{\theta}(a_t|s_t)}{\pi_{\theta_k}(a_t|s_t)}$, and $A^{\pi_{\theta_k}}$ represents the advantage estimates, which are calculated based on the predictions from behavior-supported value model $V_{\phi_k}$.

**Importantly,** PPO ensures the accuracy of gradient prediction when reusing data by clipping the gradients where the policy ratio difference exceeds a certain threshold, thereby enhancing sample efficiency. If we focus solely on the portion used to update the policy gradient, Equation 9 provides an unbiased estimate of the optimization objective in Equation 3. Together with the equivalent policy evaluation demonstrated in Corollary 3, our implementation of the *Behavior-Supported Policy Optimization* algorithm (BSPO) is theoretically consistent with the analysis in Section 3.

Bringing everything together, the pseudo-code of our algorithm is shown in Algorithm 1.

## 5 EXPERIMENTS

In this section, we present experiments to demonstrate the effectiveness of the BSPO algorithm. Specifically, we focus on the following three aspects:

- BSPO outperforms baseline algorithms, proving its capacity to better mitigate reward over-optimization and find the optimal in-distribution policies (Section 5.2).
- BSPO reduces the generation of OOD responses during the RL, thereby avoiding overestimation caused by the extrapolation errors of the reward prediction (Section 5.3).
- BSPO effectively avoids over-optimization at larger KL divergence distances (Section 5.4).

### 5.1 EXPERIMENTAL SETTINGS

To rigorously evaluate our algorithm, we compare it with five baseline methods across three proxy model scales in the synthetic setup. The specific experimental settings are as follows:

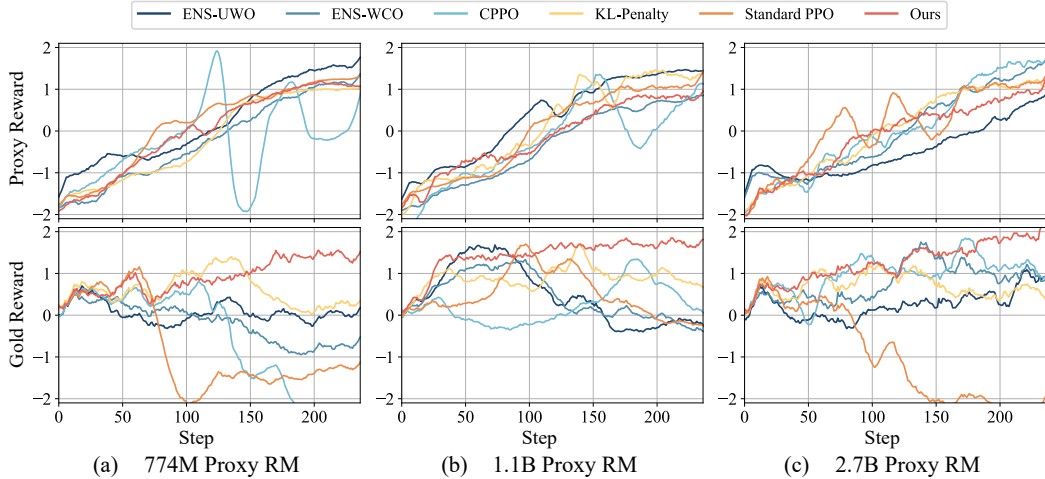

(a)   774M Proxy RM          (b)   1.1B Proxy RM          (c)   2.7B Proxy RM

Figure 3: **Main results.** The training curves of various algorithms across three experimental settings show upward trends in proxy rewards. Most baselines suffer from reward over-optimization. In contrast, our BSPO algorithm effectively mitigates this issue and achieves the highest gold reward.

**Synthetic Setup**   Given the high cost of human evaluations for studying reward over-optimization, we employ a widely used synthetic setup in over-optimization research (Gao et al., 2023; Moskovitz et al., 2023; Coste et al., 2023). In this setup, labels are generated by a "gold-standard" reward model (*gold model*) rather than by humans. Meanwhile, the *proxy model* is a proxy of the ground truth, trained to fit the labels from the gold model.

In our experiments, we train the gold model based on Llama3-8B (Dubey et al., 2024). For the proxy model, we utilize three smaller models: 774M (GPT-2-large (Radford et al., 2019)), 1.1B (TinyLlama (Zhang et al., 2024a)), and 2.7B (ShearedLlama (Xia et al., 2023)), employing the ScoreLM architecture as described in Section 4. The gold model is trained using 57k preference pairs from the binarized UltraFeedback dataset (Cui et al., 2023). The gold model re-annotates 30k data points, which are then used for training the proxy models, whose training curves are provided in Appendix D.2. During RL, Alpaca-7B (Taori et al., 2023) is employed as the initial actor model. All algorithms are trained using the same proxy model and evaluated against the same gold model.

**Baselines**   We implement five baseline algorithms for comparison (details in Appendix C.1):

- `Standard PPO`: The standard implementation of Proximal Policy Optimization (PPO) (Schulman et al., 2017) without the KL penalty in the LLM RLHF (Ouyang et al., 2022a).
- `KL-Penalty`: Add a per-token KL penalty to the standard PPO algorithm (Gao et al., 2023).
- `CPPO`: Constrained PPO identifies potentially over-optimized proxy points through experimentation and constrains the reward value to be below those points during RL (Moskovitz et al., 2023).
- `ENS-UWO` & `ENS-WCO`: The ensemble baseline (ENS) integrates four reward models, utilizing variance for uncertainty-weighted optimization (UWO) or employing the worst reward for conservative worst-case optimization (WCO) (Eisenstein et al., 2023; Coste et al., 2023).

## 5.2   MAIN RESULTS

Figure 3 presents the training curves of different algorithms across three proxy model scales. Our BSPO algorithm effectively mitigates reward over-optimization across all parameter sizes, as evidenced by the consistent upward trends in both the *proxy reward* and the *gold reward*. One possible explanation is that BSPO reduces the OOD generation, thereby minimizing extrapolation errors from the proxy model, which we empirically validate in Section 5.3. We also observe that BSPO more reliably finds the optimal policies achieving the highest gold reward.

In contrast, most baseline algorithms continue to suffer from the reward over-optimization problem. First, the inability of ensemble methods to handle OOD responses, as noted in Eisenstein et al.

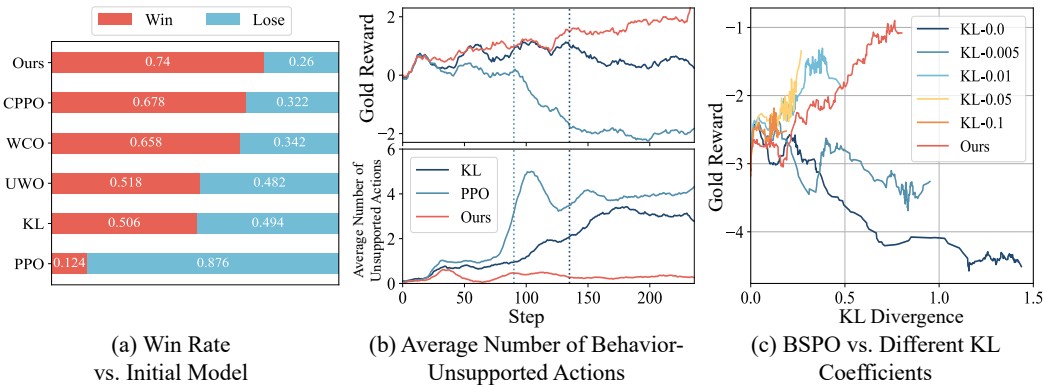

Figure 4: **(a) Win Rates.** The win rates of different algorithms against the initial SFT model using the 2.7B proxy model. **(b) Count behavior-unsupported actions during RL.** We track the average number of actions (tokens) that are not supported by the behavior policy for each response during RL. Over-optimization in traditional methods leads to a sharp rise in unsupported actions, as shown by the dashed line, while our BSPO algorithm keeps these actions consistently low during training. **(c) Compare with KL penalty.** Compared with the KL penalty method using different penalty coefficients, BSPO effectively avoids reward over-optimization at larger KL divergence distances.

(2023), remains a concern. These methods mitigate overestimation within the ID setting, but for OOD responses, multiple models may still consistently overestimate. As the policy iteration, the frequency of OOD responses tends to increase, further amplifying this issue. Second, methods that apply KL penalties or maximum constraints to the reward can be viewed as limiting the policy search to a specific region. However, unlike BSPO, these methods do not explicitly model the ID region of the reward model, and thus may only mitigate over-optimization in the overlapping region (typically near the initial policy). Third, while the constraints of CPPO successfully keep the proxy reward near the threshold, the gold reward continues to change. Therefore, relying solely on the proxy reward may be insufficient to establish a threshold that indicates when over-optimization occurs. Furthermore, CPPO introduces instability inherent to the Lagrangian approach (Platt & Barr, 1987).

Finally, to further validate the results, we evaluate the final model on the test set and calculate its win rate against the initial Alpaca-7B, as shown in Figure 4(a). Unlike other baseline methods that adjust the sequence-wise reward, BSPO penalizes action-wise OOD values without affecting ID ones, allowing it to find the same optimal ID policies as standard policy evaluation. This may explain why the model trained with BSPO outperforms the other baselines shown in the figure.

### 5.3 ABLATION ON THE BEHAVIOR SUPPORTED PREDICTION

We employ the behavior policy $\beta$, which denotes the next token distribution in the reward model's training dataset, as a method for OOD detection of the reward model. To experimentally illustrate how BSPO mitigates the issue of over-optimization, we track the average number of actions (tokens) that are not supported by the behavior policy for each response during the RL process.

The results are shown in Figure 4(b). We observe that the occurrence of the over-optimization phenomenon in traditional methods is accompanied by a significant increase in the number of behavior-unsupported actions, as indicated by the dashed line. In contrast, our BSPO algorithm maintains a consistently low number of such actions throughout the training process. This supports one of our core insights: regularizing the value function reduces the generation of OOD responses, thereby avoiding the overestimation problem caused by the extrapolation error from OOD evaluation.

### 5.4 FURTHER COMPARISON WITH DISTANCE CONTROL

We further highlight the necessity of modeling the ID region of the reward model. Figure 4(c) presents a comparison between the KL penalty method, using different penalty coefficients, and BSPO. The KL penalty method can be seen as an approach that constrains the policy from deviating

too far from the initial one under a distance measure (Gao et al., 2023). The results show that BSPO avoids over-optimization even at larger KL divergence distances, whereas the KL penalty method fails at closer distances, only ensuring a consistent increase of gold reward within a proximal region.

This phenomenon suggests that the region defined by the distance constraint does not align with the ID region of the proxy model. As a result, valid reward prediction without extrapolation error can only be guaranteed within the proximal region that is fully covered by the ID region (as the inner dashed circle in Figure 1(b)). In contrast, our algorithm models the ID region directly through the behavior policy $\beta$, enabling us to relax the distance constraint and fully explore the entire region where the reward model can generate valid predictions.

## 6 CONCLUSION AND DISCUSSION

A primary cause of reward over-optimization is the extrapolation error that occurs when the reward model evaluates OOD responses. However, current methods still fail to prevent the increasing generation of OOD responses during the RL process and are not effective at handling extrapolation errors from their reward prediction. Furthermore, they apply reward regularization to pessimistically handle OOD responses which, meanwhile, introduces unintended changes to ID ones, potentially leading to suboptimal solutions. In this work, we propose the *Behavior-Supported Policy Optimization* (BSPO), a method grounded in two core insights. First, we define *behavior policy* as the next token distribution of the reward training dataset to model the ID region of the reward model. Second, we introduce the behavior-supported Bellman operator to regularize the value function, penalizing all OOD values without impacting the ID ones. It illustrates that BSPO iterates the policy in the ID region of the reward model, thereby avoiding the reward model to evaluate the OOD response. Theoretically, we prove that BSPO ensures a monotonic improvement of the supported policy until convergence to the optimal ID policy, a guarantee that other methods lack. Extensive experiments demonstrate that BSPO outperforms baselines and exhibits two key strengths. First, BSPO reduces the generation of OOD responses during the RL, thereby avoiding overestimation caused by the extrapolation errors. Second, BSPO effectively avoids reward over-optimization at larger distances, facilitating the search for the optimal ID policy.

### 6.1 LIMITATION AND FUTURE WORK

This study has several notable limitations. First, due to the unaffordable costs of human analyzing for studying the reward over-optimization, we employ the widely accepted synthetic setup (Gao et al., 2023; Coste et al., 2023; Moskovitz et al., 2023). However, there are some differences between the synthetic setup and real-world scenarios. For example, model-based evaluations tend to exhibit higher consistency, while human preferences are subject to greater variability. Therefore, further validation experiments in real-world settings are concerned. In Appendix D.5, We emulate human behavior by a LLM to conduct experiments in a realistic setting and engage in further discussions.

Second, our algorithm primarily addresses the issue of reward over-optimization during the RL phase caused by OOD responses in evaluations (Eisenstein et al., 2023; Laidlaw et al., 2024; Yang et al., 2024b), assuming the use of a well-trained reward model by default. However, if the reward model fails to accurately capture preferences during the reward learning phase, this could also lead to deviations from human values (Miao et al., 2024). It is important to recognize that this issue parallels our concerns and is at a different stage of training. Another aspect to be considered additionally is the presence of an OOD prompt in the RL phase, as there may be no response for the reward model ID on these OOD prompt. This requires a further study of the data distribution shift between the reward model phase and the RL phase.

Third, a current research direction in LLM alignment involves multiple reward models (Moskovitz et al., 2023; Ji et al., 2023a; Dai et al., 2023a). Extending our algorithm to multi-objective scenarios to mitigate the over-optimization of each reward model presents a promising direction.

## ACKNOWLEDGEMENTS

This research is supported by the STI 2030 Major Projects (2021ZD0200403) and the Natural Science Foundation of China (No. 61925603)

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

ETHICS STATEMENT

This paper presents work that aims to advance the field of machine learning, specifically focusing on mitigating reward over-optimization in RLHF. There are many potential societal consequences of our work, none of which we feel must be specifically highlighted here.

# A  SUPPLEMENTARY DETAILS OF ANALYSIS SECTION

## A.1  PROOFS

**Notations.** For any state $s$, the supported action space of state $s$ is denoted as $\text{supp}\,(\beta(\cdot|s)) \doteq \{a \in \mathcal{A} \mid \beta(a|s) > 0\}$. The set of all behavior-supported policies is denoted as

$$\Pi_\beta \doteq \{\pi \in \Pi \mid \pi(a|s) = 0 \quad, \forall a \notin \text{supp}\,(\beta(\cdot|s))\}. \tag{10}$$

We define $r_{\min} \doteq \min_{s \in \mathcal{S}, a \in \mathcal{A}} [r(s,a)]$ and $Q_{\min} \doteq \sum_{t=0}^{\infty} \gamma^t r_{\min} = \frac{r_{\min}}{1-\gamma}$. Then, we propose the following behavior-supported Bellman operator:

$$\mathcal{T}_\beta^\pi Q(s,a) \doteq \begin{cases} \mathcal{T}^\pi Q(s,a), & \text{if} \quad \beta(a|s) > 0, \\ Q_{\min}, & \text{otherwise.} \end{cases} \tag{11}$$

where $\mathcal{T}^\pi Q(s,a) \doteq r(s,a) + \gamma \mathbb{E}_{a' \sim \pi(\cdot|s')}[Q(s',a')]$ is the traditional Bellman operator and $s' = T(s,a)$ is the next state. This new operator has the following properties:

**Theorem 1** (Contraction of $\mathcal{T}_\beta^\pi$)**.** *For any policy $\pi \in \Pi$, the operator $\mathcal{T}_\beta^\pi$ is $\gamma$-contraction with respect to the $\mathcal{L}_\infty$ norm over the space $\mathcal{S} \times \mathcal{A}$.*

*Proof.* For any functions $f_1, f_2 : \mathcal{S} \times \mathcal{A} \to \mathbb{R}$, any policy $\pi$, and $\forall s \in \mathcal{S}, a \in \mathcal{A}$, if $\beta(a|s) = 0$ we have

$$\left|\mathcal{T}_\beta^\pi f_1(s,a) - \mathcal{T}_\beta^\pi f_2(s,a)\right| = |Q_{\min} - Q_{\min}| = 0 \le \gamma \|f_1 - f_2\|_\infty; \tag{12}$$

else if $\beta(a|s) > 0$ we have

$$\begin{aligned}
\left|\mathcal{T}_\beta^\pi f_1(s,a) - \mathcal{T}_\beta^\pi f_2(s,a)\right| &= \left|\gamma \mathbb{E}_{a' \sim \pi(\cdot|s')}[f_1(s',a') - f_2(s',a')]\right| \\
&\le \gamma \mathbb{E}_{a' \sim \pi(\cdot|s')}|f_1(s',a') - f_2(s',a')| \\
&\le \gamma \max_{s \in \mathcal{S}, a \in \mathcal{A}}|f_1(s,a) - f_2(s,a)| \\
&= \gamma \|f_1 - f_2\|_\infty
\end{aligned} \tag{13}$$

where $s' = T(s,a)$.

Thus, we have $\left\|\mathcal{T}_\beta^\pi f_1 - \mathcal{T}_\beta^\pi f_2\right\|_\infty \le \gamma \|f_1 - f_2\|_\infty$, that is, $\mathcal{T}_\beta^\pi$ is $\gamma$-contraction. $\square$

The $\gamma$-contraction property of the operator $\mathcal{T}_\beta^\pi$ guarantees that, for any initial Q-values, iterative application of $\mathcal{T}_\beta^\pi$ converges to a unique fixed point at a rate of $\gamma$. Thus, we find this fixed point by iteratively solving the problem, $\min_Q \mathbb{E}_{\tau \sim \pi}(Q(s,a) - \mathcal{T}_\beta^\pi Q(s,a))^2$, and use it for policy evaluation to optimize the policy $\pi$.

The fixed point $Q_\beta^\pi$ has the following properties:

**Theorem 2** (Fixed Points)**.** *For any policy $\pi \in \Pi$, the fixed point $Q_\beta^\pi$ of $\mathcal{T}_\beta^\pi$ satisfies*

$$\begin{cases} Q_{min} \le Q_\beta^\pi(s,a) \le Q^\pi(s,a), & \text{if} \quad \beta(a|s) > 0, \\ Q_\beta^\pi(s,a) = Q_{min}, & \text{otherwise.} \end{cases} \tag{2}$$

*where $Q^\pi(s,a)$ is fixed point of standard Bellman operator $\mathcal{T}^\pi$.*

*Proof.* By Theorem 1, $\mathcal{T}_\beta^\pi$ is $\gamma$-contraction. Assume the $Q_\beta^\pi$ is the fixed point, then we have

$$Q_\beta^\pi(s,a) = \mathcal{T}_\beta^\pi Q_\beta^\pi(s,a) = \begin{cases} \mathcal{T}^\pi Q_\beta^\pi(s,a), & \text{if} \quad \beta(a|s) > 0, \\ Q_{\min}, & \text{otherwise.} \end{cases} \tag{14}$$

where $s' = T(s, a)$.

Denoting $(\hat{s}, \hat{a})$ is the minimum point of $Q_\beta^\pi(s, a)$ when $\beta(a|s) > 0$, that is, we define $(\hat{s}, \hat{a}) \doteq \arg\min_{s \in \mathcal{S}, a \in \text{supp}(\beta(\cdot|s))} Q_\beta^\pi(s, a)$. Then, we have

$$
\begin{aligned}
Q_\beta^\pi(\hat{s}, \hat{a}) &= r(\hat{s}, \hat{a}) + \gamma \mathbb{E}_{a' \sim \pi(\cdot|s')} \left[ Q_\beta^\pi(s', a') \right] \qquad \text{where} \quad s' \doteq T(\hat{s}, \hat{a}) \\
&= r(\hat{s}, \hat{a}) + \gamma \left[ \sum_{a' \in \text{supp}(\beta(\cdot|s'))} \pi(a'|s') \mathcal{T}^\pi Q_\beta^\pi(s', a') + \sum_{a' \notin \text{supp}(\beta(\cdot|s'))} \pi(a'|s') Q_{\min} \right] \\
&= r(\hat{s}, \hat{a}) + \gamma \left[ \sum_{a' \in \text{supp}(\beta(\cdot|s'))} \pi(a'|s') Q_\beta^\pi(s', a') + \sum_{a' \notin \text{supp}(\beta(\cdot|s'))} \pi(a'|s') Q_{\min} \right] \\
&\geq r(\hat{s}, \hat{a}) + \gamma \left[ \sum_{a' \in \text{supp}(\beta(\cdot|s'))} \pi(a'|s') \right] Q_\beta^\pi(\hat{s}, \hat{a}) + \gamma \left[ \sum_{a' \notin \text{supp}(\beta(\cdot|s'))} \pi(a'|s') \right] Q_{\min}
\end{aligned}
$$
(15)

The last inequality in Equation 15 holds since $(\hat{s}, \hat{a})$ is the minimum point of $Q_\beta^\pi(s, a)$. To simplify the expression, we define $\lambda \doteq \sum_{a' \sim \text{supp}(\beta(\cdot|s'))} \pi(a'|s')$. Then, $1 - \lambda = \sum_{a' \notin \text{supp}(\beta(\cdot|s'))} \pi(a'|s')$. Bringing it into the above inequality and since $\gamma \in [0, 1), \lambda \in [0, 1]$, we have

$$
\begin{aligned}
Q_\beta^\pi(\hat{s}, \hat{a}) &\geq r(\hat{s}, \hat{a}) + \gamma\lambda Q_\beta^\pi(\hat{s}, \hat{a}) + \gamma(1 - \lambda)Q_{\min} \\
(1 - \gamma\lambda)Q_\beta^\pi(\hat{s}, \hat{a}) &\geq r(\hat{s}, \hat{a}) + \gamma(1 - \lambda)Q_{\min} \\
Q_\beta^\pi(\hat{s}, \hat{a}) &\geq \frac{r(\hat{s}, \hat{a}) + \gamma(1 - \lambda)Q_{\min}}{1 - \gamma\lambda}
\end{aligned}
$$
(16)

Since $r(\hat{s}, \hat{a}) \geq r_{\min} = (1 - \gamma)Q_{\min}$, it follows that

$$
Q_\beta^\pi(\hat{s}, \hat{a}) \geq \frac{(1 - \gamma)Q_{\min} + \gamma(1 - \lambda)Q_{\min}}{1 - \gamma\lambda} = Q_{\min}
$$
(17)

Since $(\hat{s}, \hat{a})$ is the minimum point of $Q_\beta^\pi(s, a)$, the following equation holds

$$
\forall s \in \mathcal{S}, a \in \text{supp}(\beta(\cdot|s)), \quad Q_\beta^\pi(s, a) \geq Q_\beta^\pi(\hat{s}, \hat{a}) \geq Q_{\min}
$$
(18)

Then, since $\forall s \in \mathcal{S}, a \notin \text{supp}(\beta(\cdot|s)), Q_\beta^\pi(s, a) = Q_{\min}$, we have $\forall s \in \mathcal{S}, a \in \mathcal{A}, Q_\beta^\pi(s, a) \geq Q_{\min}$.

Next, we will prove the relationship between $Q_\beta^\pi(s, a)$ and $Q^\pi(s, a)$.

$\forall s \in \mathcal{S}, a \in \mathcal{A}$, we have

$$
\begin{aligned}
\mathcal{T}^\pi Q_\beta^\pi(s, a) &= r(s, a) + \gamma \mathbb{E}_{a' \sim \pi(\cdot|s')} \left[ Q_\beta^\pi(s', a') \right] \qquad \text{where} \quad s' \doteq T(\hat{s}, \hat{a}) \\
&\geq r(s, a) + \gamma \mathbb{E}_{a' \sim \pi(\cdot|s')} \left[ Q_{\min} \right] \\
&= (1 - \gamma)Q_{\min} + \gamma Q_{\min} \\
&= Q_{\min}
\end{aligned}
$$
(19)

Thus, for any $s \in \mathcal{S}, a \in \mathcal{A}$, we have

$$
\begin{aligned}
\mathcal{T}_\beta^\pi Q_\beta^\pi(s, a) &= \begin{cases} \mathcal{T}^\pi Q_\beta^\pi(s, a), & \text{if} \quad \beta(a|s) > 0, \\ Q_{\min} \leq \mathcal{T}^\pi Q_\beta^\pi(s, a), & \text{otherwise} \end{cases} \\
&\leq \mathcal{T}^\pi Q_\beta^\pi(s, a)
\end{aligned}
$$
(20)

$\forall s \in \mathcal{S}, a \in \mathrm{supp}(\beta(\cdot|s))$, it holds that

$$
\begin{aligned}
Q_\beta^\pi(s,a) &= \mathcal{T}_\beta^\pi Q_\beta^\pi(s,a) \\
&= \mathcal{T}^\pi Q_\beta^\pi(s,a) \\
&= r(s,a) + \gamma \mathbb{E}_{a' \sim \pi(\cdot|s')}\left[Q_\beta^\pi(s',a')\right] \quad \text{where } s' \doteq T(\hat{s},\hat{a}) \\
&= r(s,a) + \gamma \mathbb{E}_{a' \sim \pi(\cdot|s')}\left[\mathcal{T}_\beta^\pi Q_\beta^\pi(s',a')\right] \\
&\leq r(s,a) + \gamma \mathbb{E}_{a' \sim \pi(\cdot|s')}\left[\mathcal{T}^\pi Q_\beta^\pi(s',a')\right] \\
&= r(s,a) + \gamma \mathbb{E}_{a' \sim \pi(\cdot|s')}\left[r(s',a') + \gamma \mathbb{E}_{a'' \sim \pi(\cdot|s'')}\left[Q_\beta^\pi(s'',a'')\right]\right] \quad \text{where } s'' \doteq T(\hat{s}',\hat{a}') \\
&= r(s,a) + \gamma \mathbb{E}_{a' \sim \pi(\cdot|s')}\left[r(s',a') + \gamma \mathbb{E}_{a'' \sim \pi(\cdot|s'')}\left[\mathcal{T}_\beta^\pi Q_\beta^\pi(s'',a'')\right]\right] \\
&\leq r(s,a) + \gamma \mathbb{E}_{a' \sim \pi(\cdot|s')}\left[r(s',a') + \gamma \mathbb{E}_{a'' \sim \pi(\cdot|s'')}\left[\mathcal{T}^\pi Q_\beta^\pi(s'',a'')\right]\right] \\
&\cdots \\
&\leq \mathbb{E}_\pi\left[\sum_{i=0}^\infty \gamma^i r(s_i,a_i) \mid s_0=s, a_0=a\right] \\
&= Q^\pi(s,a)
\end{aligned}
\tag{21}
$$

Combining Equation 18 and Equation 21, we prove that $\forall \pi \in \Pi$, the fixed point $Q_\beta^\pi$ of $\mathcal{T}_\beta^\pi$ satisfies

$$
\begin{cases}
Q_{\min} \leq Q_\beta^\pi(s,a) \leq Q^\pi(s,a), & \text{if} \quad \beta(a|s) > 0, \\
Q_\beta^\pi(s,a) = Q_{\min}, & \text{otherwise.}
\end{cases}
\tag{22}
$$

$\square$

We refer to the fixed point $Q_\beta^\pi$ as the behavior-supported Q-value function. It underestimates the future returns of OOD actions, thereby disadvantaging these actions across the action space. Now, we consider a common policy iteration using behavior-supported value, as follows:

$$
\pi_{k+1} = \arg\max_{\pi \in \Pi} \mathbb{E}_{\tau \sim \pi_k}\left[\frac{\pi(a_t|s_t)}{\pi_k(a_t|s_t)} A_\beta^{\pi_k}(s_t,a_t)\right].
\tag{23}
$$

where $V_\beta^{\pi_k}(s) = \mathbb{E}_{a \sim \pi_k(\cdot|s)}\left[Q_\beta^{\pi_k}(s,a)\right]$, and $A_\beta^{\pi_k}(s,a) = Q_\beta^{\pi_k}(s,a) - V_\beta^{\pi_k}(s)$. Furthermore, the following corollary guarantees that the policy optimization method in Equation 3 yields behavior-supported policies $\pi \in \Pi_\beta$ through regularized policy evaluation.

**Corollary 1** (Supported Policy Optimization). *The policy optimization method mentioned in Equation 3 yields behavior-supported policy $\pi \in \Pi_\beta$.*

*Proof.* For any $s \in \mathcal{S}$,

$$
d_\mu^{\pi_k}(s) = d_\mu^{\pi_k}(x \cup a_{0:t-1}) = \sum_{x \in \mathcal{X}} \mu(x) \prod_{i=0}^{t-1} \pi_k(a_i|x \cup a_{0:i-1}).
\tag{24}
$$

Then, we have

$$
\begin{aligned}
\pi_{k+1}(a|s) &= \arg\max_{\pi \in \Pi} \mathbb{E}_{\tau \sim \pi_k}\left[\frac{\pi(a_t|s_t)}{\pi_k(a_t|s_t)} A_\beta^{\pi_k}(s_t,a_t)\right], \\
&= \arg\max_{\pi \in \Pi} \sum_{s \in \mathcal{S}} d_\mu^{\pi_k}(s) \sum_{a \in \mathcal{A}} \pi_k(a|s)\left[\frac{\pi(a|s)}{\pi_k(a|s)} A_\beta^{\pi_k}(s,a)\right], \\
&= \arg\max_{\pi \in \Pi} \sum_{s \in \mathcal{S}} d_\mu^{\pi_k}(s) \sum_{a \in \mathcal{A}} \pi(a|s)\left[A_\beta^{\pi_k}(s,a)\right], \\
&= \mathbb{I}\left[a = \arg\max_{a' \in \mathcal{A}} A_\beta^{\pi_k}(s,a')\right], \\
&= \mathbb{I}\left[a = \arg\max_{a' \in \mathcal{A}} \left(Q_\beta^{\pi_k}(s,a') - V_\beta^{\pi_k}(s)\right)\right], \\
&= \mathbb{I}\left[a = \arg\max_{a' \in \mathcal{A}} Q_\beta^{\pi_k}(s,a')\right].
\end{aligned}
\tag{25}
$$

Since $Q_\beta^{\pi_k}(s, a)$ is equal to the fixed point of the $\mathcal{T}_\beta^{\pi_k}$, it holds that, $\forall s \in \mathcal{S}, \forall a' \in \text{supp}(\beta(\cdot|s))$, $\forall a'' \notin \text{supp}(\beta(\cdot|s))$,

$$Q_\beta^{\pi_k}(s, a') \geq Q_{\min} = Q_\beta^{\pi_k}(s, a'') \tag{26}$$

where the inequality holds because of Theorem 2. Therefore, the maximum solution will only choose behavior-supported actions. It follows that

$$\pi_{k+1}(a|s) = \mathbb{I}\left[a = \arg\max_{a' \in \mathcal{A}} Q_\beta^{\pi_k}(s, a')\right] = \mathbb{I}\left[a = \arg\max_{a' \in \text{supp}(\beta(\cdot|s)} Q_\beta^{\pi_k}(s, a')\right] \in \Pi_\beta \tag{27}$$

In conclusion, for arbitrary policy $\pi_k$, the policy optimization method mentioned in Equation 3 yields a supported next policy $\pi_{k+1} \in \Pi_\beta$. □

Theoretically, we demonstrate that policy iteration with the regularized value function guarantees that each iteration results in a behavior-supported policy. This restricts the search for the optimal policy to the ID region of the reward model. As a result, the reward model does not need to evaluate trajectories generated by OOD actions, thereby preventing the over-optimization issue caused by extrapolation errors. For behavior-supported policy iterations, the following corollary holds:

**Corollary 2.** *For any behavior-supported policy* $\pi \in \Pi_\beta$, *the fixed point* $Q_\beta^\pi$ *of* $\mathcal{T}_\beta^\pi$ *satisfies*

$$Q_\beta^\pi(s, a) = \begin{cases} Q^\pi(s, a), & \text{if} \quad \beta(a|s) > 0, \\ Q_{min}, & \text{otherwise}. \end{cases} \tag{5}$$

*Proof.* By Theorem 1, $\mathcal{T}_\beta^\pi$ is $\gamma$-contraction. Assume the $Q_\beta^\pi$ is the fixed point, then we have

$$Q_\beta^\pi(s, a) = \mathcal{T}_\beta^\pi Q_\beta^\pi(s, a) = \begin{cases} \mathcal{T}^\pi Q_\beta^\pi(s, a), & \text{if} \quad \beta(a|s) > 0, \\ Q_{\min}, & \text{otherwise}. \end{cases} \tag{28}$$

where $s' = T(s, a)$.

For all supported policy $\pi \in \Pi_\beta$, it holds that $\forall s \in \mathcal{S}, a \notin \text{supp}(\beta(\cdot|s), \pi(a|s) = 0$. Then, we have

$$
\begin{aligned}
\mathbb{E}_{a \sim \pi(\cdot|s)}\left[\mathcal{T}_\beta^\pi Q_\beta^\pi(s, a)\right] &= \sum_{a \in \text{supp}(\beta(\cdot|s))} \pi(a|s)\mathcal{T}^\pi Q_\beta^\pi(s, a) + \sum_{a \notin \text{supp}(\beta(\cdot|s))} \pi(a|s)Q_{\min} \\
&= \sum_{a \in \text{supp}(\beta(\cdot|s))} \pi(a|s)\mathcal{T}^\pi Q_\beta^\pi(s, a) \\
&= \sum_{a \in \text{supp}(\beta(\cdot|s))} \pi(a|s)\mathcal{T}^\pi Q_\beta^\pi(s, a) + \sum_{a \notin \text{supp}(\beta(\cdot|s))} \pi(a|s)\mathcal{T}^\pi Q_\beta^\pi(s, a) \\
&= \mathbb{E}_{a \sim \pi(\cdot|s)}\left[\mathcal{T}^\pi Q_\beta^\pi(s, a)\right]
\end{aligned} \tag{29}
$$

Thus, $\forall s \in \mathcal{S}, a \in \text{supp}(\beta(\cdot|s))$, it holds that

$$
\begin{aligned}
Q_\beta^\pi(s, a) &= \mathcal{T}_\beta^\pi Q_\beta^\pi(s, a) \\
&= \mathcal{T}^\pi Q_\beta^\pi(s, a) \\
&= r(s, a) + \gamma\mathbb{E}_{a' \sim \pi(\cdot|s')}\left[Q_\beta^\pi(s', a')\right] \quad \text{where } s' \doteq T(\hat{s}, \hat{a}) \\
&= r(s, a) + \gamma\mathbb{E}_{a' \sim \pi(\cdot|s')}\left[\mathcal{T}_\beta^\pi Q_\beta^\pi(s', a')\right] \\
&= r(s, a) + \gamma\mathbb{E}_{a' \sim \pi(\cdot|s')}\left[\mathcal{T}^\pi Q_\beta^\pi(s', a')\right] \\
&= r(s, a) + \gamma\mathbb{E}_{a' \sim \pi(\cdot|s')}\left[r(s', a') + \gamma\mathbb{E}_{a'' \sim \pi(\cdot|s'')}\left[Q_\beta^\pi(s'', a'')\right]\right] \quad \text{where } s'' \doteq T(\hat{s}', \hat{a}') \\
&= r(s, a) + \gamma\mathbb{E}_{a' \sim \pi(\cdot|s')}\left[r(s', a') + \gamma\mathbb{E}_{a'' \sim \pi(\cdot|s'')}\left[\mathcal{T}_\beta^\pi Q_\beta^\pi(s'', a'')\right]\right] \\
&= r(s, a) + \gamma\mathbb{E}_{a' \sim \pi(\cdot|s')}\left[r(s', a') + \gamma\mathbb{E}_{a'' \sim \pi(\cdot|s'')}\left[\mathcal{T}^\pi Q_\beta^\pi(s'', a'')\right]\right] \\
&\cdots \\
&= \mathbb{E}_\pi\left[\sum_{i=0}^\infty \gamma^i r(s_i, a_i) \mid s_0 = s, a_0 = a\right] \\
&= Q^\pi(s, a)
\end{aligned} \tag{30}
$$

$\square$

Corollary 2 indicates that the behavior-supported policy evaluation provides the same unbiased Q-values for all ID actions as the standard Bellman operator $\mathcal{T}^\pi$. Together with Corollary 1, for any state $s$, the optimal behavior-supported action is the same under the $Q_\beta^\pi$ and $Q^\pi$ value functions during behavior-supported policy iteration.

Furthermore, we use the following lemma to prove the monotonicity of our method:

**Lemma 1** (Performance Difference, Lemma 6.1 in (Kakade & Langford, 2002)). *For any policies $\pi$ and $\pi'$ and any starting state distribution $\mu$,*

$$\mathcal{J}(\pi') - \mathcal{J}(\pi) = \frac{1}{1-\gamma} \mathbb{E}_{s \sim d_\mu^{\pi'}, a \sim \pi'(\cdot|s)} [A^\pi(s,a)] \tag{31}$$

**Theorem 3** (Monotonicity to Optimality). *The behavior-supported policy optimization method mentioned in Equation 3 results in a strictly monotonic improvement of policy performance, continuing until the optimal behavior-supported policy $\pi_\beta^*$ is attained.*

*Proof.* First, we prove the monotonicity. Given a sequence of policies $\{\pi_i\}_{i=0}^\infty$ generated by the policy optimization method in Equation 3. From Lemma 1, for any $k \geq 0$, we have

$$
\begin{aligned}
\mathcal{J}(\pi_{k+1}) - \mathcal{J}(\pi_k) &= \frac{1}{1-\gamma} \mathbb{E}_{s \sim d_\mu^{\pi_{k+1}}, a \sim \pi_{k+1}(\cdot|s)} [A^{\pi_k}(s,a)] \\
&= \frac{1}{1-\gamma} \sum_{s \in \mathcal{S}} d_\mu^{\pi_{k+1}}(s) \sum_{a \in \mathcal{A}} \pi_{k+1}(a|s) A^{\pi_k}(s,a) \\
&= \frac{1}{1-\gamma} \sum_{s \in \mathcal{S}} d_\mu^{\pi_{k+1}}(s) \sum_{a \in \mathcal{A}} \pi_{k+1}(a|s) \left[ Q_\beta^{\pi_k}(s,a) - V_\beta^{\pi_k}(s) \right] \\
&= \frac{1}{1-\gamma} \sum_{s \in \mathcal{S}} d_\mu^{\pi_{k+1}}(s) \left[ \sum_{a \in \mathcal{A}} \pi_{k+1}(a|s) Q_\beta^{\pi_k}(s,a) - V_\beta^{\pi_k}(s) \right] \\
&= \frac{1}{1-\gamma} \sum_{s \in \mathcal{S}} d_\mu^{\pi_{k+1}}(s) \left[ \sum_{a \in \mathcal{A}} \pi_{k+1}(a|s) Q_\beta^{\pi_k}(s,a) - \sum_{a \in \mathcal{A}} \pi_k(a|s) Q_\beta^{\pi_k}(s) \right] \\
&= \frac{1}{1-\gamma} \mathbb{E}_{s \sim d_\mu^{\pi_{k+1}}} \left[ \mathbb{E}_{a \sim \pi_{k+1}(\cdot|s)} Q_\beta^{\pi_k}(s,a) - \mathbb{E}_{a \sim \pi_k(\cdot|s)} Q_\beta^{\pi_k}(s,a) \right] \\
&\geq 0
\end{aligned}
\tag{32}
$$

where the last inequality holds since $\pi_{k+1}$ is the greedy policy with respect to $Q_\beta^{\pi_k}$, i.e., $\pi_{k+1}(a|s) = \mathbb{I}\left[ a = \arg\max_{a' \in \text{supp}(\beta(\cdot|s))} Q_\beta^{\pi_k}(s,a') \right]$.

Second, we prove the convergence. For any $k \geq 0$, if $\mathcal{J}(\pi_{k+1}) - \mathcal{J}(\pi_k) = 0$, since $\forall s \in \mathcal{S}, d_\mu^{\pi_{k+1}}(s) > 0$, we have

$$\mathbb{E}_{a \sim \pi_{k+1}(\cdot|s)} Q_\beta^{\pi_k}(s,a) = \mathbb{E}_{a \sim \pi_k(\cdot|s)} Q_\beta^{\pi_k}(s,a) \tag{33}$$

Since $Q_\beta^{\pi_k}$ is equal to the fixed point of the Bellman Operator $\mathcal{T}_\beta^{\pi_k}$, it holds that

$$Q_\beta^{\pi_k}(s,a) = \mathcal{T}_\beta^{\pi_k} Q_\beta^{\pi_k}(s,a) = \begin{cases} \mathcal{T}^{\pi_k} Q_\beta^{\pi_k}(s,a), & \text{if} \quad \beta(a|s) > 0, \\ Q_{\min}, & \text{otherwise.} \end{cases} \tag{34}$$

As a result, $\forall s \in \mathcal{S}, a \in \mathcal{A}, \beta(a|s) > 0$, we have

$$
\begin{aligned}
Q_\beta^{\pi_k}(s,a) &= \mathcal{T}^{\pi_k} Q_\beta^{\pi_k}(s,a) \\
&= r(s,a) + \gamma \mathbb{E}_{a' \sim \pi_k(\cdot|s')}\left[Q_\beta^{\pi_k}(s',a')\right] \quad \text{where } s' = T(s,a) \\
&= r(s,a) + \gamma \mathbb{E}_{a' \sim \pi_{k+1}(\cdot|s')}\left[Q_\beta^{\pi_k}(s',a')\right] \quad \text{from Equation 33} \\
&= r(s,a) + \gamma \sum_{a \in \mathcal{A}} \mathbb{I}\left[a = \underset{a' \in \mathrm{supp}(\beta(\cdot|s'))}{\arg\max} Q_\beta^{\pi_k}(s',a')\right] Q_\beta^{\pi_k}(s',a) \quad \text{from Equation 27} \\
&= r(s,a) + \gamma \max_{a' \in \mathrm{supp}(\beta(\cdot|s'))} Q_\beta^{\pi_k}(s',a') \\
&= \mathcal{T}_\beta^* Q_\beta^{\pi_k}(s,a)
\end{aligned}
$$

(35)

Thus, $Q_\beta^{\pi_k}$ is the fixed point of the Bellman optimality operator $\mathcal{T}_\beta^*$, i.e., $Q_\beta^{\pi_k} = Q_\beta^*$. From Equation 27, we have,

$$
\pi_{k+1}(a|s) = \mathbb{I}\left[a = \underset{a' \sim \mathrm{supp}(\beta(\cdot|s))}{\arg\max} Q_\beta^{\pi_k}(s,a')\right] = \mathbb{I}\left[a = \underset{a' \sim \mathrm{supp}(\beta(\cdot|s))}{\arg\max} Q_\beta^*(s,a')\right] = \pi_\beta^*(a|s)
$$

(36)

$\square$

## B  SUPPLEMENTARY DETAILS OF IMPLEMENTATION SECTION

### B.1  PROOFS

**Behavior-Supported Value Function**  The core of our algorithm involves training a critic model to perform behavior-supported policy evaluation. In our implementation, we predict behavior-supported V-values instead of Q-values to achieve greater stability while maintaining equivalent policy evaluation. The deterministic state transition equation of a token-level MDP in LLM, $T(x \cup a_{0:t-1}, a_t) = x \cup a_{0:t}$, ensures this equivalence. Specifically, the behavior-supported Bellman V-operator is defined as:

$$
\mathcal{T}_{\beta,V}^\pi V(x \cup a_{0:t-1}) \doteq
\begin{cases}
\mathcal{T}_V^\pi V(x \cup a_{0:t-1}), & \text{if } \beta(a_{t-1}|x \cup a_{0:t-2}) > 0, \\
\frac{1}{\gamma}\left[Q_{\min} - r(x \cup a_{0:t-2}, a_{t-1})\right], & \text{otherwise,}
\end{cases}
$$

(37)

where $\mathcal{T}_V^\pi V(x \cup a_{0:t-1}) \doteq \mathbb{E}_{a_t \sim \pi(\cdot|x \cup a_{0:t-1})}\left[r(x \cup a_{0:t-1}, a_t) + \gamma V(x \cup a_{0:t})\right]$ is the standard Bellman operator for the V-value function. This operator has the following properties:

**Theorem 4** (Contraction of $\mathcal{T}_{\beta,V}^\pi$). *For any policy $\pi \in \Pi$, the operator $\mathcal{T}_{\beta,V}^\pi$ is $\gamma$-contraction with respect to the $\mathcal{L}_\infty$ norm over the state space $\mathcal{S}$.*

*Proof.* $\forall f_1, f_2 : \mathcal{S} \to \mathbb{R}, \forall \pi \in \Pi$, and $\forall s \in \mathcal{S}$, denoting $s = x \cup a_{0:t-1}$, if $\beta(a_{t-1}|x \cup a_{0:t-2})=0$, it follows that

$$
\left|\mathcal{T}_{\beta,V}^\pi f_1(s) - \mathcal{T}_{\beta,V}^\pi f_2(s)\right| = |V_{\min}(s) - V_{\min}(s)| = 0 \le \gamma \|f_1 - f_2\|_\infty ;
$$

(38)

if $\beta(a_{t-1}|x \cup a_{0:t-2}) > 0$, we have

$$
\begin{aligned}
\left|\mathcal{T}_{\beta,V}^\pi f_1(s) - \mathcal{T}_{\beta,V}^\pi f_2(s)\right| &= \left|\mathbb{E}_{a \sim \pi(\cdot|s)}\left[r(s,a) + \gamma f_1(s')\right] - \mathbb{E}_{a \sim \pi(\cdot|s)}\left[r(s,a) + \gamma f_2(s')\right]\right| \\
&= \left|\gamma \mathbb{E}_{a \sim \pi(\cdot|s)}\left[f_1(s') - f_2(s')\right]\right| \\
&\le \gamma \mathbb{E}_{a \sim \pi(\cdot|s)}\left|f_1(s') - f_2(s')\right| \\
&\le \gamma \max_{s \in \mathcal{S}}|f_1(s) - f_2(s)| \\
&= \gamma \|f_1 - f_2\|_\infty
\end{aligned}
$$

(39)

where $s' = T(s,a)$. Thus, $\|\mathcal{T}_{\beta,V}^\pi f_1 - \mathcal{T}_{\beta,V}^\pi f_2\|_\infty \le \gamma \|f_1 - f_2\|_\infty$, that is, $\mathcal{T}_{\beta,V}^\pi$ is a $\gamma$-contraction operator in the $\mathcal{L}_\infty$ norm within the $\mathcal{S}$ space. $\square$

We refer to the fixed point $V_\beta^\pi$ as the behavior-supported V-value function. Now, we will prove that $Q_\beta^\pi$ and $V_\beta^\pi$ have the equivalent policy evaluation.

**Corollary 3** (Equivalent Policy Evaluation). *$\forall \pi \in \Pi$, denote $V_\beta^\pi$ as the fixed point of the Bellman operator $\mathcal{T}_{\beta,V}^\pi$ in Equation 7. Then, its corresponding state-action value function $Q_\beta^\pi(s,a) = r(s,a) + \gamma V_\beta^\pi(T(s,a))$ is equal to the fixed point of the Bellman operator $\mathcal{T}_\beta^\pi$ in Equation 1.*

*Proof.* By Theorem 4, $\mathcal{T}_{\beta,V}^\pi$ is a $\gamma$-contraction operator in the $\mathcal{L}_\infty$ norm within the $\mathcal{S}$ space. Assume the $V_\beta^\pi$ is the fixed point, then we have $\forall s = x \cup a_{0:t-1} \in \mathcal{S}$,

$$V_\beta^\pi(x \cup a_{0:t-1}) = \begin{cases} \mathcal{T}_V^\pi V(x \cup a_{0:t-1}), & \text{if} \quad \beta(a_{t-1}|x \cup a_{0:t-2}) > 0, \\ \frac{1}{\gamma}\left[Q_{\min} - r(x \cup a_{0:t-2}, a_{t-1})\right], & \text{otherwise}, \end{cases} \tag{40}$$

$\forall s = x \cup a_{0:t-1} \in \mathcal{S}, a_t \in \mathcal{A}$, if $\beta(a_t|x \cup a_{0:t-1}) = 0$, it holds that

$$\begin{aligned} Q_\beta^\pi(x \cup a_{0:t-1}, a_t) &= r(x \cup a_{0:t-1}, a_t) + \gamma V_\beta^\pi(x \cup a_{0:t}) \\ &= r(x \cup a_{0:t-1}, a_t) + \gamma \cdot \frac{1}{\gamma}\left[Q_{\min} - r(x \cup a_{0:t-1}, a_t)\right] \tag{41} \\ &= Q_{\min} \end{aligned}$$

if $\beta(a_t|x \cup a_{0:t-1}) > 0$, it holds that

$$\begin{aligned} Q_\beta^\pi(x \cup a_{0:t-1}, a_t) &= r(x \cup a_{0:t-1}, a_t) + \gamma V_\beta^\pi(x \cup a_{0:t}) \\ &= r(x \cup a_{0:t-1}, a_t) + \gamma \mathcal{T}_V^\pi V(x \cup a_{0:t}) \\ &= r(x \cup a_{0:t-1}, a_t) + \gamma \mathbb{E}_{a_{t+1} \sim \pi(\cdot|x \cup a_{0:t})}\left[r(x \cup a_{0:t}, a_{t+1}) + \gamma V_\beta^\pi(x \cup a_{0:t+1})\right] \\ &= r(x \cup a_{0:t-1}, a_t) + \gamma \mathbb{E}_{a_{t+1} \sim \pi(\cdot|x \cup a_{0:t})}\left[Q_\beta^\pi(x \cup a_{0:t}, a_{t+1})\right] \\ &= \mathcal{T}^\pi Q_\beta^\pi(x \cup a_{0:t-1}, a_t) \end{aligned} \tag{42}$$

Therefore, we have

$$\begin{aligned} Q_\beta^\pi(x \cup a_{0:t-1}, a_t) &= \mathcal{T}_\beta^\pi Q_\beta^\pi(x \cup a_{0:t-1}, a_t) \\ &= \begin{cases} \mathcal{T}^\pi Q_\beta^\pi(x \cup a_{0:t-1}, a_t), & \text{if} \quad \beta(a_t|x \cup a_{0:t-1}) > 0, \\ Q_{\min}, & \text{otherwise}. \end{cases} \end{aligned} \tag{43}$$

which means $Q_\beta^\pi$ is the fixed point of the Bellman operator $\mathcal{T}_\beta^\pi$ in Equation 1.

On the other hand, we will prove that if $Q_\beta^\pi$ is the fixed point of the Bellman operator $\mathcal{T}_\beta^\pi$ in Equation 1, then its corresponding state value function $V_\beta^\pi$ is the fixed point of the Bellman operator $\mathcal{T}_{\beta,V}^\pi$ in Equation 7.

Assume $Q_\beta^\pi$ is the fixed point of the Bellman operator $\mathcal{T}_\beta^\pi$. $\forall s = x \cup a_{0:t-1} \in \mathcal{S}, \forall a_t \in \mathcal{A}$, it holds that

$$Q_\beta^\pi(x \cup a_{0:t-1}, a_t) = \begin{cases} \mathcal{T}^\pi Q_\beta^\pi(x \cup a_{0:t-1}, a_t), & \text{if} \quad \beta(a_t|x \cup a_{0:t-1}) > 0, \\ Q_{\min}, & \text{otherwise}. \end{cases} \tag{44}$$

Since $Q_\beta^\pi(x \cup a_{0:t-2}, a_{t-1}) = r(x \cup a_{0:t-2}, a_{t-1}) + \gamma V_\beta^\pi(x \cup a_{0:t-1})$, $\forall s = x \cup a_{0:t-1} \in \mathcal{S}$, if $\beta(a_{t-1}|x \cup a_{0:t-2}) = 0$, it follows that

$$\begin{aligned} V_\beta^\pi(x \cup a_{0:t-1}) &= \frac{1}{\gamma}\left[Q_\beta^\pi(x \cup a_{0:t-2}, a_{t-1}) - r(x \cup a_{0:t-2}, a_{t-1})\right] \\ &= \frac{1}{\gamma}\left[Q_{\min} - r(x \cup a_{0:t-2}, a_{t-1})\right] \end{aligned} \tag{45}$$

if $\beta(a_{t-1}|x \cup a_{0:t-2}) > 0$, it follows that

$$\begin{aligned} V_\beta^\pi(x \cup a_{0:t-1}) &= \mathbb{E}_{a_t \sim \pi(\cdot|x \cup a_{0:t-1})}\left[Q_\beta^\pi(x \cup a_{0:t-1}, a_t)\right] \\ &= \mathbb{E}_{a_t \sim \pi(\cdot|x \cup a_{0:t-1})}\left[r(x \cup a_{0:t-1}, a_t) + \gamma V_\beta^\pi(x \cup a_{0:t})\right] \end{aligned} \tag{46}$$

Therefore, $\forall s \in \mathcal{S}$, we have $V_\beta^\pi(s) = \mathcal{T}_{\beta,V}^\pi V_\beta^\pi(s)$ which means $V_\beta^\pi$ is the fixed point of the Bellman operator $\mathcal{T}_{\beta,V}^\pi$ in Equation 7. $\qquad \square$

## B.2 PSEUDO-CODE

We provide the pseudo-code of the implementation of our BSPO algorithm as follows:

---

**Algorithm 1** Behavior-Supported Policy Optimization

---

1: **Require:** Reward function $r(x, y)$, behavior function $\beta(a|s)$, initial policy $\pi_0$, initial value function $V_0$
2: **for** step $k = 1, \ldots, K$ **do**
3:     Generate responses

$$\hat{\boldsymbol{y}}_b = (a_{b,0}, \ldots, a_{b,T-1}) \sim p_{\pi_{k-1}}(\boldsymbol{y}_b|\boldsymbol{x}_b) = \prod_{t=0}^{T-1} \pi_{k-1}(a_{b,t}|s_{b,t})$$

    where $s_{b,0} = \boldsymbol{x}_b$, and $b = 1, \cdots, B$ refers to the index within the batch.
4:     Compute reward with KL penalty†:

$$\boldsymbol{r}_{b,T-1}^{\text{RM}} = r(\boldsymbol{x}_b, \hat{\boldsymbol{y}}_b),$$

$$r_{b,t}^{\text{KL}} = -\log \frac{\pi_k(a_{b,t}|\boldsymbol{x}_b \cup a_{b,0:t-1})}{\pi_0(a_{b,t}|\boldsymbol{x}_b \cup a_{b,0:t-1})}, \quad (0 \le t \le T-1),$$

$$\hat{r}_{b,t} = r_{b,t}^{\text{RM}} + \nu \cdot r_{b,t}^{\text{KL}}, \quad (0 \le t \le T-1),$$

    where $\nu$ represents the KL penalty coefficient.
5:     Compute advantage estimates:

$$\hat{A}_{b,t} = \delta_{b,t} + \gamma\lambda\delta_{b,t+1} + \cdots + (\gamma\lambda)^{T-t-1}\delta_{b,T-1},$$

    where $\delta_{b,t} \doteq \hat{r}_{b,t} + \gamma V_{k-1}(s_{b,t+1}) - V_{k-1}(s_{b,t})$.
6:     Compute actor loss:

$$\mathcal{L}_{\text{PPO}} = -\frac{1}{BT}\sum_{b=1}^{B}\sum_{t=0}^{T-1}\min\{\rho_{b,t}(k)\hat{A}_{b,t}, \text{clip}(\rho_{b,t}(k), 1-\epsilon, 1+\epsilon)\hat{A}_{b,t}\},$$

    where $\rho_{b,t}(k) = \frac{\pi_k(a_{b,t}|s_{b,t})}{\pi_{k-1}(a_{b,t}|s_{b,t})}$.
7:     Compute behavior-supported value function and critic loss‡:

$$\mathcal{T}_{\beta,V}^{\pi_{\boldsymbol{\theta}_k}} V_k(x_b \cup a_{b,0:t-1}) \doteq \begin{cases} \hat{r}_{b,t} + \gamma V_k(x_b \cup a_{b,0:t}) & \text{if} \quad \beta(a_{b,t-1} \mid x_b \cup a_{b,0:t-2}) > \epsilon_\beta, \\ V_{min}, & \text{otherwise,} \end{cases}$$

$$\mathcal{L}_V = \frac{1}{BT}\sum_{b=1}^{B}\sum_{t=0}^{T-1}\left[\left(V(s_{b,t}) - \mathcal{T}_{\beta,V}^{\pi_{k-1}} V_k(s_{b,t})\right)^2\right].$$

    where $\epsilon_\beta$ and $V_{min}$ are constant minimal values representing the unsupported action's threshold and the minimum of the value function.
8:     Update actor model and critic model with $\mathcal{L}_{\text{PPO}}$ and $\mathcal{L}_V$
9: **end for**
10: **Return** $\pi_K$

---

†: Since our value regularization method is compatible with the reward regularization method based on KL divergence control, we preserve this component of the standard RLHF implementation in the pseudo-code. Notably, we ensure a fair comparison against the KL-only method in main experiments and provide a detailed analysis of the differences between the two methods in Section 5.4.

‡: In the implementation, a parameterized method predicts the behavior policy. To prevent modeling errors, an infinitesimal value $\epsilon_\beta$ is used as a threshold, instead of zero, to determine whether an action is supported.

## C  Supplementary Details of Experiments

### C.1  Baselines

In this section, we elaborate on baseline algorithms and provide details of our implementation.

#### C.1.1  Standard PPO

We adopt the training procedure proposed by Ouyang et al. (2022b), with a focus on the RL objective. In this setup, the output from the score head of the ScoreLM model serves as the reward function for RL training of baselines to ensure a fair comparison. Given a prompt $x \sim \mathcal{D}_{\text{prompt}}$, the current actor model $\pi_\theta(y|x)$ generates a corresponding response $y = a_{0:T-1}$ with length $T$. Then the reward for tokens $a_{0:T-1}$ is defined as:

$$
r_t^{\text{RM}} = \begin{cases} 0, & 0 \leq t < T - 1, \\ R_\phi(x, y), & t = T - 1, \end{cases} \tag{47}
$$

In the RLHF fine-tuning phase, we use the Proximal Policy Optimization (PPO) algorithm (Schulman et al., 2017) to train the LLM. The surrogate PPO clip loss for the RL training objective is formulated as:

$$
\mathcal{L}^{\text{RL}}(\theta; \mathcal{D}_{\text{prompt}}) = -\mathbb{E}_{x \sim \mathcal{D}_{\text{prompt}}, y \sim \pi_\theta(y|x)} \left[ \mathbb{E}_t \left[ \min \left( \rho_t(\theta) \hat{A}^{\hat{r}_t}, \text{clip} \left( \rho_t(\theta), 1 - \epsilon, 1 + \epsilon \right) \hat{A}^{\hat{r}_t} \right) \right] \right] \tag{48}
$$

where $\rho_t(\theta) = \frac{\pi_\theta(a_t|x \cup a_{0:t-1})}{\pi_{\theta_{\text{old}}}(a_t|x \cup a_{0:t-1})}$ is the importance sampling weight and $\theta_{\text{old}}$ is model parameters from the previous gradient update, $\epsilon \in (0, 1)$ is the PPO clip ratio. $\hat{A}_t^{\hat{r}}$ is the advantage of the reward estimated by the GAE method (Schulman et al., 2018).

#### C.1.2  KL Penalty

The primary difference from standard PPO is the addition of a per-token KL penalty to the reward function used in the RL training. With the prompt $x \sim \mathcal{D}_{prompt}$ and the corresponding response $y = a_{1:T}$, the reward function can be expressed as:

$$
r_t^{\text{KL}} = -\log \frac{\pi_\theta(a_t|x \cup a_{0:t-1})}{\pi_{\text{ref}}(a_t|x \cup a_{0:t-1})}, \quad (0 \leq t \leq T - 1), \tag{49}
$$

$$
\hat{r}_t = r_t^{\text{RM}} + \beta \cdot r_t^{\text{KL}}, \quad (0 \leq t \leq T - 1), \tag{50}
$$

where $\pi_{\text{ref}}(\cdot \mid x)$ denotes the reference model, and $\beta \geq 0$ represents the KL penalty coefficient. For each token, a dense reward is applied, penalized by the KL divergence between the current actor model and the reference model. The score head of the ScoreLM model provides a sparse reward only on the final token. The reference model is a frozen LLM initialized with the actor model's parameters at the start of the RLHF phase.

#### C.1.3  Ensemble

We propose to learn an ensemble of reward models $\{R_1, ..., R_k\}$ in the reward model training stage. During policy optimization, we combine the reward estimates from different reward models within the ensemble according to the following two methods which have been proved to be more effective.

**Worst-Case Optimization**  Worst-case optimization (WCO) creates a conservative estimate by choosing the lowest reward from the ensemble at every step. Choosing the lowest reward helps ensure that as long as at least one ensemble member does not overestimate the true reward, policy optimization will not result in over-optimization.

$$R_{WCO}(x,y) := \min_i R_i(x,y) \tag{51}$$

A major advantage of WCO is that it does not have any hyperparameters that might require tuning. However, it can sometimes result in a performance penalty due to its highly conservative nature.

**Uncertainty-Weighted Optimization**   In uncertainty-weighted optimization (UWO), the reward for a sample is calculated by combining the average reward across all models in an ensemble with the intra-ensemble variance, weighted by a coefficient $\lambda$. Intuitively, UWO works by penalizing the policy for generating responses for which there is high disagreement among reward models within the ensemble. This helps prevent the exploitation of a single faulty reward model which might be erroneously assigning high rewards to incorrect or low-quality responses. Mathematically, this objective is given by:

$$R_{UWO}(x,y) = \underbrace{\frac{1}{k}\sum_i R_i(x,y)}_{mean} - \lambda \underbrace{\frac{1}{k}\sum_i \left( R_i(x,y) - \frac{1}{k}\sum_i R_i(x,y) \right)^2}_{variance} \tag{52}$$

where $\lambda$ is a hyperparameter which controls the weight of the uncertainty component.

### C.1.4   CONSTRAINED PPO

The basic idea of Constrained PPO (Moskovitz et al., 2023) is to model the over-optimization problem as a constraint in the RLHF process and then solve it using the Safe RL (Ji et al., 2024b; 2023c; Yang et al., 2022; Dai et al., 2023b; Meng et al., 2021; Dai et al., 2024; Achiam et al., 2017) domain approach. Unlike the original paper which uses several rule-based rewards, we employ an LLM-based reward model. To identify proxy point, we train PPO agents (Schulman et al., 2017) to maximize the reward model (without KL regularization) and plot the resulting evaluation scores against the gold model. We observe that the evaluation score initially increase before falling, so we identify the point where further optimization will lead to a decrease in the gold score. After performing the above steps multiple times, we select the most conservative proxy point to ensure that roo will not occur.

Once one has identified proxy point for the reward model, the next question is how to train agents to maximize the reward until they hit this critical value. In constrained reinforcement learning, an agent seeks to maximize its value while adhering to constraints on its behavior. Mathematically, this problem is formalized as a constrained MDP (Altman, 1999). For clarity, we will hereafter refer to r0 as the "task reward" rather than just the reward, and the CMDP optimization problem is given by:

$$\max_\pi v_0^\pi \text{ s.t. } v_i^\Pi \geq \theta_i, i = 1, \ldots, N \tag{53}$$

As mentioned in the paper, $r_0$ is selected as kl divergence, and we have only one reward model, that is, N=1. Given our possible objectives, we can now consider how to optimize them. One popular approach to solving constrained problems is to use Lagrangian relaxation:

$$\max_\pi \min_{\mu \geq 0} v_0^\pi + \sum_{i=1}^N \mu_i(v_i^\pi - \theta_i) \tag{54}$$

where the weights on the value of each RMs $\mu = [\mu_1, \ldots, \mu_N]^T \in R_{\geq 0}^N$ are the Lagrange multipliers associated with each constraint.

We implemented the algorithm referred to as $\xi - PPO$ in the paper, which is also the best-performing among the several algorithms proposed. It is designed to stay close to $\pi_0$ and ensure reward models hit the target, with an objective of the following form:

$$\max_\pi v_{KL}^\pi \text{ s.t. } v_j = \theta_j \forall j \neq i \tag{55}$$

With this objective, we then design mixed advantages which are a convex combination of the task and constraint advantages:

$$A_\mu^\pi(s, a) = (N - \sum_{i=1}^{N} \mu_i) A_0^\pi(s, a) + \sum_{i=1}^{N} \mu_i A_i^\pi(s, a) \tag{56}$$

This equation has the intuitive interpretation of placing more weight on optimizing constraint reward $r_i > 0$ when $\mu_i > 0$ is high (indicating a constraint violation), and more weight on task reward $r_0$ when $\mu_{1:N}$ are low (indicating that constraints are satisfied).

The original paper employs a tanh function to constrain the Lagrange multipliers between -1 and 1. However, in our experiment, the total number of training steps is relatively low due to the large batch size. Using the tanh function slows down the optimization of the Lagrange multipliers, causing them to remain elevated for too long when constraints are satisfied, and vice versa. Therefore, we replace the tanh function with a simple clipping of the Lagrange multipliers between -1 and 1.

## C.2 HYPER-PARAMETERS

In this section, we provide all the hyper-parameters used in our experiments.

Table 1: Hyper-parameters of three experimental settings of BSPO.

| Hyper-parameters | Proxy-774M | Proxy-1.1B | Proxy-2.7B |
|---|---|---|---|
| epochs | 3 | 3 | 3 |
| max_length | 512 | 512 | 512 |
| temperature | 1.2 | 1.2 | 1.2 |
| top_p | 1 | 1 | 1 |
| num_return_sequences | 1 | 1 | 1 |
| repetition_penalty | 1.0 | 1.0 | 1.0 |
| per_device_prompt_batch_size | 8 | 8 | 8 |
| per_device_train_batch_size | 8 | 8 | 8 |
| gradient_accumulation_steps | 1 | 1 | 1 |
| actor_lr | 1.00E-5 | 1.00E-5 | 1.00E-5 |
| actor_weight_decay | 0.01 | 0.01 | 0.01 |
| actor_lr_scheduler_type | cosine | cosine | cosine |
| actor_lr_warmup_ratio | 0.03 | 0.03 | 0.03 |
| actor_gradient_checkpointing | TRUE | TRUE | TRUE |
| critic_lr | 5.00E-06 | 5.00E-06 | 5.00E-06 |
| critic_weight_decay | 0.0 | 0.0 | 0.0 |
| critic_lr_scheduler_type | constant | constant | constant |
| critic_lr_warmup_ratio | 0.03 | 0.03 | 0.03 |
| critic_gradient_checkpointing | TRUE | TRUE | TRUE |
| clip_range_ratio ($\epsilon$) | 0.2 | 0.2 | 0.2 |
| unsupported_value | -15 | -15 | -15 |
| epsilon_beta ($\epsilon_\beta$) | 1.00E-4 | 1.00E-4 | 1.00E-4 |
| bf16 | TRUE | TRUE | TRUE |
| tf32 | TRUE | TRUE | TRUE |

Table 2: Hyper-parameters of Reward Model Training.

| Hyper-parameters | Gold-8B | Proxy-774M | Proxy-1.1B | Proxy-2.7B |
|---|---|---|---|---|
| epochs | 2 | 2 | 2 | 2 |
| max_length | 1024 | 1024 | 1024 | 1024 |
| per_device_train_batch_size | 16 | 16 | 16 | 16 |
| per_device_eval_batch_size | 16 | 16 | 16 | 16 |
| gradient_accumulation_steps | 1 | 1 | 1 | 1 |
| gradient_checkpointing | TRUE | TRUE | TRUE | TRUE |
| regularization | 0.001 | 0.001 | 0.001 | 0.001 |
| lr | 2.00E-05 | 2.00E-05 | 2.00E-05 | 2.00E-05 |
| lr_scheduler_type | cosine | cosine | cosine | cosine |
| lr_warmup_ratio | 0.03 | 0.03 | 0.03 | 0.03 |
| weight_decay | 0.1 | 0.1 | 0.1 | 0.1 |
| lm_coef ($\alpha$) | NULL | 0.01 | 0.01 | 0.01 |
| bf16 | TRUE | TRUE | TRUE | TRUE |
| tf32 | TRUE | TRUE | TRUE | TRUE |

## C.3 RUNTIME ENVIRONMENT

All experiments in this paper utilize the following runtime environment. The server's CPU is an Intel(R) Xeon(R) Platinum 8358P CPU @ 2.60GHz with 128 cores, and the graphics cards are NVIDIA A100-SXM4-80GB ×8, with NVLink support and the graphics driver version being 550.54.15.

# D   MORE EXPERIMENTAL RESULTS

## D.1   PROXY MODEL FAILS TO EVALUATE UNSUPPORTED RESPONSES

Since reward learning is inherently data-driven, it faces significant challenges in evaluating responses that lie outside the distribution of the reward training dataset. As described in Section 3, we introduce an OOD detection approach for reward prediction of RLHF based on the power of well-pretrained LLMs for next-token prediction.

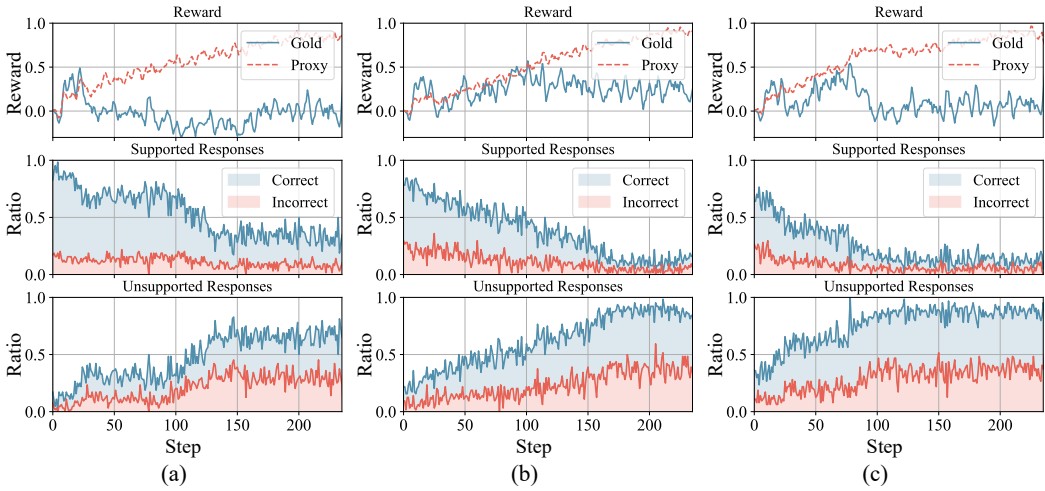

Figure 5: The prediction accuracy of the proxy model on pairs of supported and unsupported responses is evaluated across three repeated standard PPO experiments, each conducted with different random seeds.

We collect all responses generated during RLHF under the same experimental settings and categorize them as supported or unsupported based on whether the behavior policy supports the actions. These responses, along with those from the initial model given the same prompts, form comparison pairs. We then test whether the proxy model can correctly predict preferences using the gold model as ground truth. The results are provided in Figure 5.

Figure 5 shows that as the policy iterates, the proportion of unsupported responses generated by the LLM increases. For comparison pairs consisting of supported responses, the proxy model predicts preferences with accuracies of 75.91%, 71.56%, and 74.81%, which are comparable to its performance on the test set (Figure 2(b)). In contrast, for comparison pairs involving unsupported responses, the model's accuracy drops markedly to 58.10%, 62.25%, and 59.01%. These findings suggest that whether the response contains only behavior-supported actions is an effective indicator for measuring whether it is OOD for reward prediction. Furthermore, the occurrence of reward over-optimization is also directly related to the percentage of unsupported responses and the validity of the evaluation of responses (extrapolation error).

## D.2   MORE TRAINING CURVES OF SCORELM

We train three ScoreLM models for our algorithm and baselines using the loss function in Equation 6 with different model scales, as shown in Figure 6. The accuracies on the evaluation dataset demonstrate a scaling law, where larger pretrained models consistently achieve better performance on the downstream task.

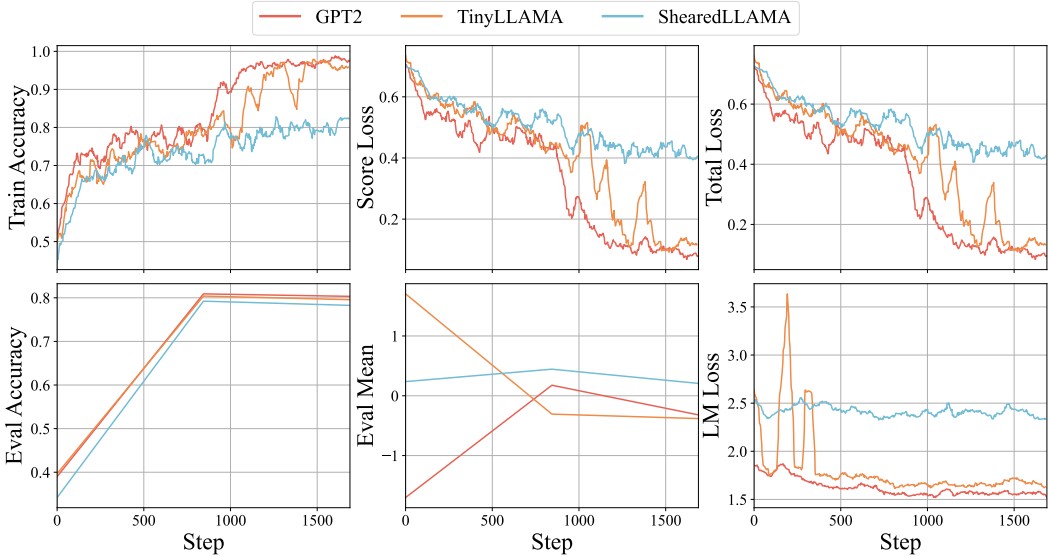

Figure 6: Training curves of ScoreLM models

## D.3 MORE TRAINING CURVES OF ENSEMBLE BASELINES

The ensemble baselines, ENS-UWO and ENS-WCO, require multiple reward or ScoreLM models. Therefore, we train four reward models for each pretrained model the same experimental settings, varying only the random seed. The training curves and evaluation curves are presented in Figure 7.

## D.4 RESPONSE LENGTH

Improvements in reward are often driven by longer responses, as LLMs exploit human raters' preference for detailed content (Singhal et al., 2024; Shen et al., 2023). Several methods aim to control this length bias (Dubois et al., 2024a; Singhal et al., 2024; Shen et al., 2023; Chen et al., 2024; Ji et al., 2025; 2024a). Our method provides a more comprehensive solution, effectively mitigating various biases, including response length.

We collect response length statistics for different methods during training, as shown in Figure 8. Both the PPO and ENS-UWO algorithms approach the model's maximum generation length, while ENS-WCO generates shorter responses due to conservation optimization. KL-penalty PPO produces shorter responses by strictly enforcing KL-divergence constraints. Constrained PPO is the most unstable, with response lengths varying unpredictably. Although there is no control for length, our proposed approach strikes a balance and achieves an appropriately concise and detailed response.

## D.5 NON-SYNTHETIC SETTING

Studying reward over-optimization requires continuous evaluation during the training and effective comparison with the proxy reward. However, human evaluation is not only expensive but also unable to provide timely scalar feedback. As a result, to the best of our knowledge, nearly all prior works (Gao et al., 2023; Coste et al., 2023; Eisenstein et al., 2023) rely on synthetic settings.

The differences between synthetic settings and real-world scenarios are an important concern. In this section, we emulate human behavior by the GPT-4o (Achiam et al., 2023) to conduct experiments in a realistic setting and engage in further discussions.

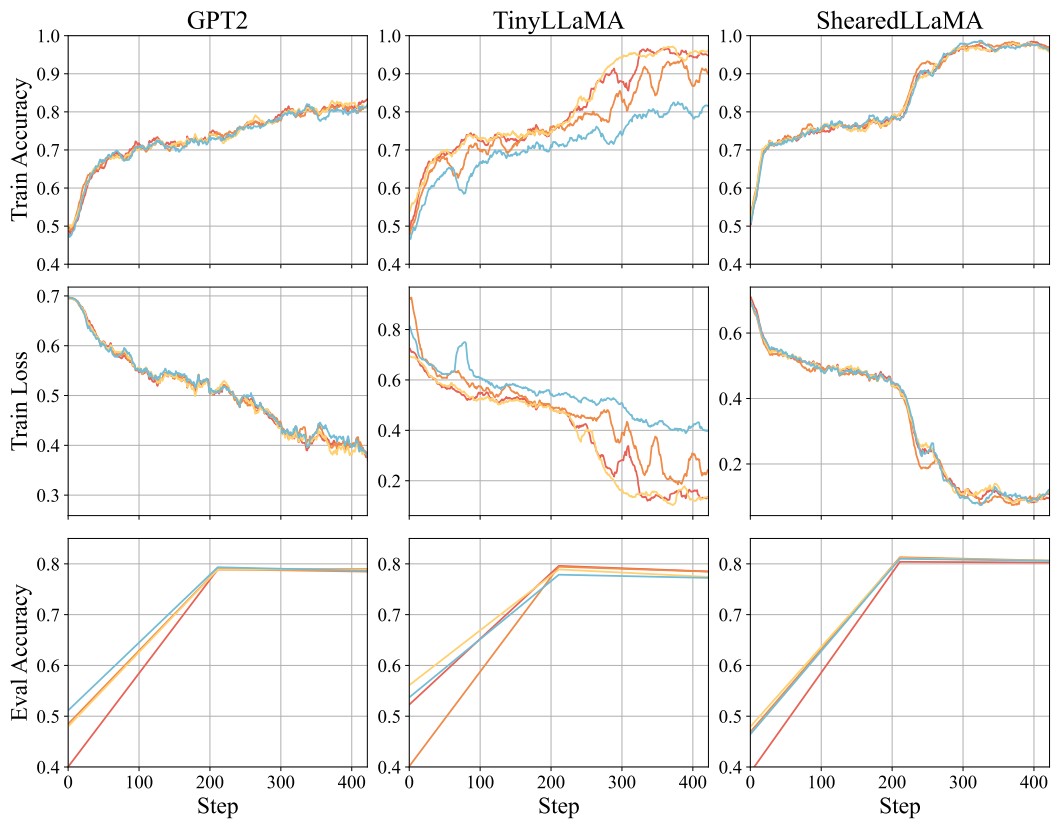

Figure 7: Training curves of four reward models used in ensemble baselines, with variations only in random seeds.

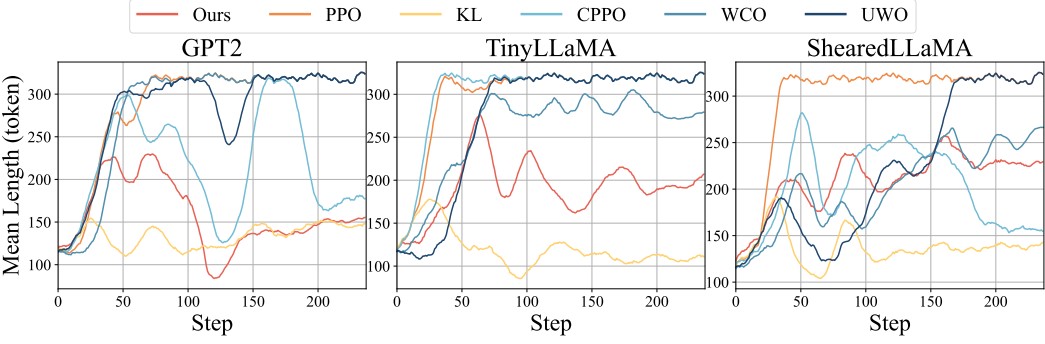

Figure 8: Mean generated length in different algorithms

### D.5.1 IMPLEMENTATION DETAILS

Assigning a scalar score to responses through human evaluation is inherently challenging. Therefore, an effective alternative involves comparing the responses generated by different checkpoints on an evaluation dataset. Specifically, we train a 7B reward model from Alpaca-7B (Taori et al., 2023) on the UltraFeedback preference dataset (Cui et al., 2023), achieving a test set accuracy of 82.94%. This reward model was subsequently used to conduct training for BSPO and the baseline methods (PPO (Ouyang et al., 2022a), WCO (Coste et al., 2023)).

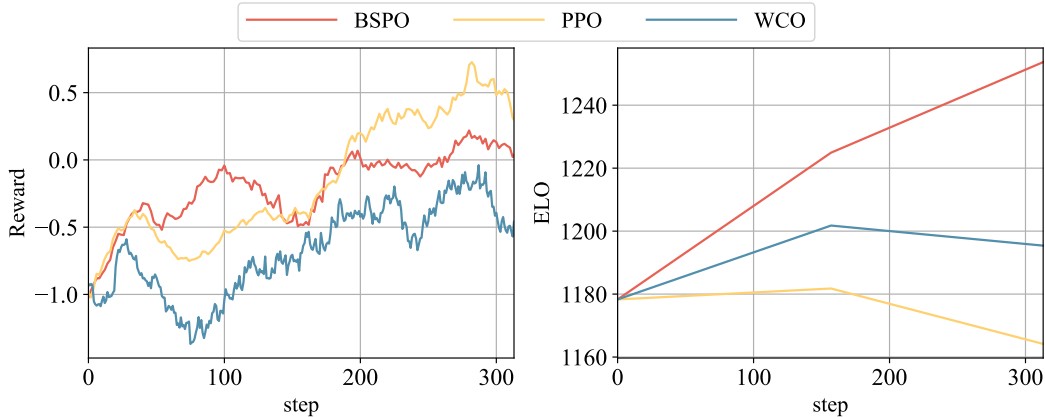

Figure 9: The training curves and ELO scores of different algorithms in the non-synthetic setup.

Table 3: The win rate of different checkpoints

| Win Ratio | Initial | PPO:79step | PPO:158step | WCO:79step | WCO:158step | BSPO:79step | BSPO:158step |
|---|---|---|---|---|---|---|---|
| Initial | 0.5 | 0.4921 | 0.5316 | 0.4683 | 0.4821 | 0.4246 | 0.3849 |
| PPO:79step | 0.5079 | 0.5 | 0.5258 | 0.4722 | 0.4942 | 0.4227 | 0.3952 7 |
| PPO:158step | 0.4684 | 0.4742 | 0.5 | 0.4643 | 0.4603 | 0.4048 | 0.369 |
| WCO:79step | 0.5317 | 0.5278 | 0.5357 | 0.5 | 0.5161 | 0.4724 | 0.4365 |
| WCO:158step | 0.5179 | 0.5058 | 0.5397 | 0.4839 | 0.5 | 0.4782 | 0.4325 |
| BSPO:79step | 0.5754 | 0.5773 | 0.5952 | 0.5276 | 0.5218 | 0.5 | 0.4484 |
| BSPO:158step | 0.6151 | 0.6048 | 0.631 | 0.5635 | 0.5675 | 0.5516 | 0.5 |

To quantify evaluation results, we compared the checkpoints on the test set to calculate the win rate between checkpoints and fit an ELO score (Askell et al., 2021) as the scalar evaluation metric. The ELO update formula is as follows:

$$R'_A = R_A + K \cdot \left( S_{AB} - \frac{1}{1 + 10^{(R_B - R_A)/400}} \right) \tag{57}$$

where $S_{AB}$ represents the win rate of Model A over Model B on the test set, $K$ is the update coefficient, and $R_A$ and $R_B$ are the ELO scores of Models A and B, respectively.

### D.5.2 EXPERIMENTAL RESULTS

The training curves and ELO scores can be found in Figure 9 and Table 4. The specific win rates between different checkpoints are presented in Table 3.

Our experiments reveal interesting insights. We observed that while the proxy reward exhibits consistently increases across the three approaches, their performance under human evaluation reveals differences. Naïve PPO shows performance improvement at 79 steps but suffers a severe decline by 158 steps, highlighting the issue of reward over-optimization. WCO maintains stable human evaluation performance across both steps, indicating its effectiveness in mitigating over-optimization. However, it does not identify the optimal solution as effectively as BSPO. BSPO demonstrates continuous improvement in both proxy reward and human evaluation, indicating its capability to address over-optimization while identifying optimal policies.

### D.6 DATA WITH LABEL NOISE

Many studies (Gao et al., 2023; Coste et al., 2023; Eisenstein et al., 2023) utilize re-annotated data with label noise to train their proxy models. Similarly, we conducted experiments to demonstrate the robustness of our algorithm. By introducing 15% label noise to the training data of the proxy reward models (TinyLlama-1.1B (Zhang et al., 2024a)), we obtained the results shown in Figure 10.

We observed the following phenomena. The inclusion of additional noisy data results in a decrease in the performance of all algorithms when evaluated on the gold reward, compared to the Figure 3 of

Table 4: The ELO scores of different checkpoints

| Model | Initial | PPO:79step | PPO:158step | WCO:79step | WCO:158step | BSPO:79step | BSPO:158step |
|---|---|---|---|---|---|---|---|
| ELO | 1178.30 | 1181.77 | 1164.17 | 1201.77 | 1195.39 | 1224.92 | 1253.68 |

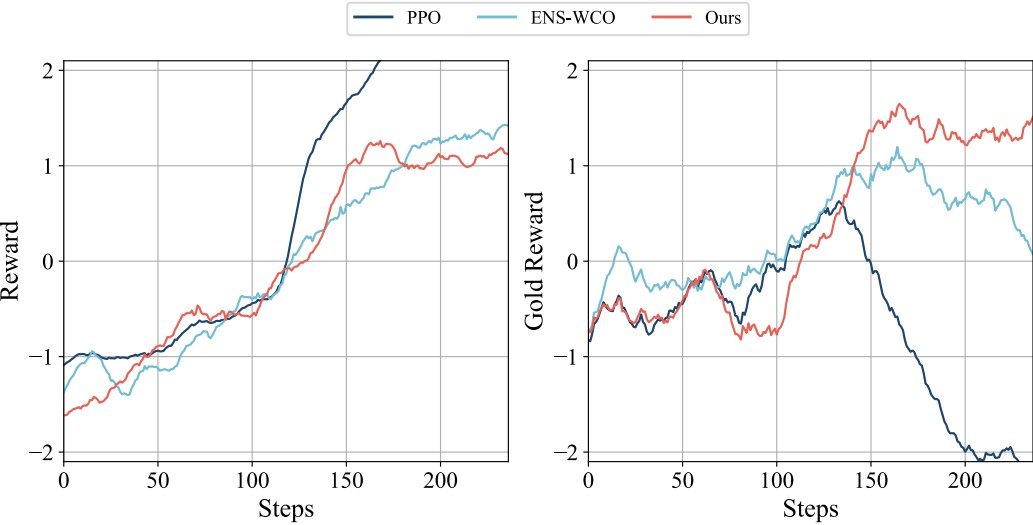

Figure 10: Training curves of different algorithms with label noise

the paper. Despite this, BSPO consistently mitigates reward over-optimization and achieves better performance than the baseline methods, demonstrating its robustness to noise.

### D.7 ABLATION STUDIES OF HYPER-PARAMETERS

#### D.7.1 ROBUSTNESS OF $V_{\mathrm{MIN}}$

$V_{\min}$ is designed to suppress the $V$-values corresponding to OOD actions, necessitating that $V_{\min}$ be lower than all possible $V$-values.

In general, $V_{\min}$ can be determined by calculating the minimum $V$-value, such as $V_{\min} = \frac{1}{1-\gamma} r_{\min}$, where $r_{\min}$ denotes the minimum value of the reward function. Specifically, for RLHF of LLMs, where only the final token receives a nonzero reward, we have $V(s) = \mathbb{E}_{\tau \sim \pi}[\gamma^{\|\tau\|} r(\tau) \mid s_0 = s] \geq \min(0, r_{\min}) = V_{\min}$.

On the other hand, empirical evidence suggests that as long as $V_{\min}$ remains smaller than all possible $V$-values, the performance of BSPO is not sensitive to the $V_{\min}$. To evaluate this, we conducted experiments under the condition $r_{\min} = -10$, testing a range of $V_{\min}$ values: $V_{\min} = -10, -15, -20, -25$.

The resulting experimental outcomes are summarized in Figure 11. We observe that, across different values of $V_{\min}$, the training curves of BSPO exhibit similar behavior, consistently improving both the proxy reward and the gold reward while effectively addressing the issue of over-optimization. These results demonstrate the robustness of the $V_{\min}$ selection.

#### D.7.2 ROBUSTNESS OF $\epsilon_\beta$

Since we use a parameterized model to predict the behavior policy. Thus, to prevent modeling errors, an infinitesimal value $\epsilon_\beta$ is used as a threshold, instead of 0, to determine whether a response is supported. To analyze the sensitivity of the BSPO algorithm to $\epsilon_\beta$, we conducted experiments with a range of values: $\epsilon_\beta = 1e-3, 1e-4, 1e-5, 1e-6, 1e-7$.

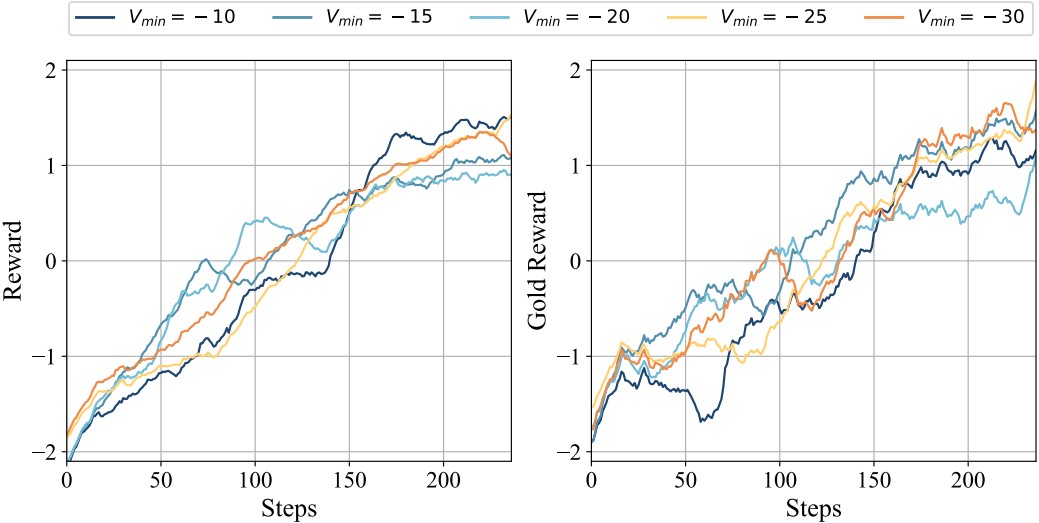

Figure 11: Training curves of different $V_{min}$ values

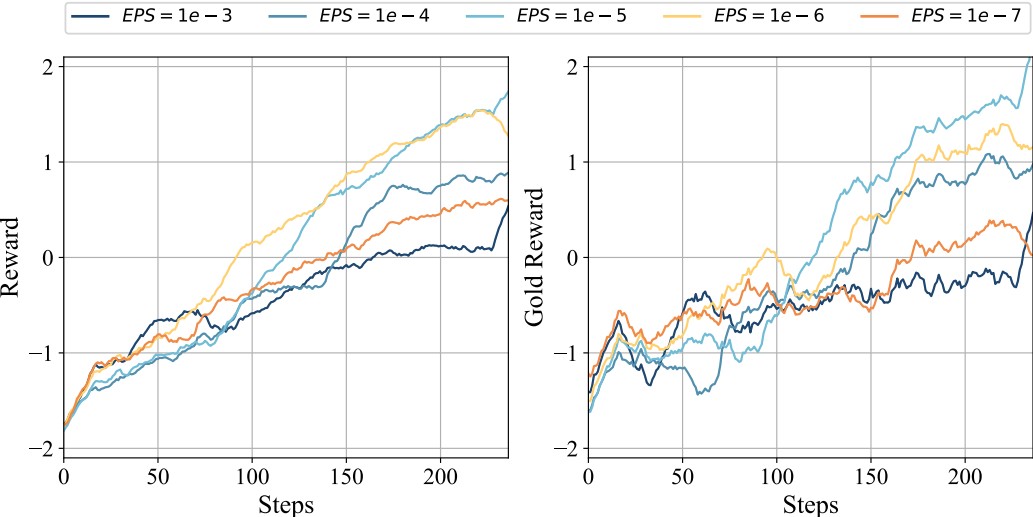

Figure 12: Training curves of different $\epsilon_\beta$ values

The experimental results are presented in the Figure 12. Our observations indicate that for $\epsilon_\beta = 1e-4, 1e-5, 1e-6$, BSPO mitigates reward over-optimization and identifies the optimal policy. However, the performance for $\epsilon_\beta = 1e-3$ and $\epsilon_\beta = 1e-7$ shows slower improvement. This behavior may result from excessively large $\epsilon$ values, which can incorrectly classify some actions as behavior-unsupported, or excessively small $\epsilon$ values, which may fail to account for certain OOD actions. In conclusion, BSPO is robust to $\epsilon_\beta$ within a reasonable range, but extreme values should be avoided.

### D.8 COMPARISON WITH DPO

Direct Preference Optimization (DPO) (Rafailov et al., 2024) is an efficient algorithm for aligning LLMs with human preferences. DPO learns from preference data with the following objective:

$$\mathcal{L}_{\text{DPO}}(\pi_\theta; \pi_{\text{ref}}) = -\mathbb{E}_{x,y_w,y_l \sim \mathcal{D}}\left[\log \sigma \left(\beta \log \frac{\pi_\theta(y_w|x)}{\pi_{\text{ref}}(y_w|x)} - \beta \log \frac{\pi_\theta(y_l|x)}{\pi_{\text{ref}}(y_l|x)}\right)\right] \quad (58)$$

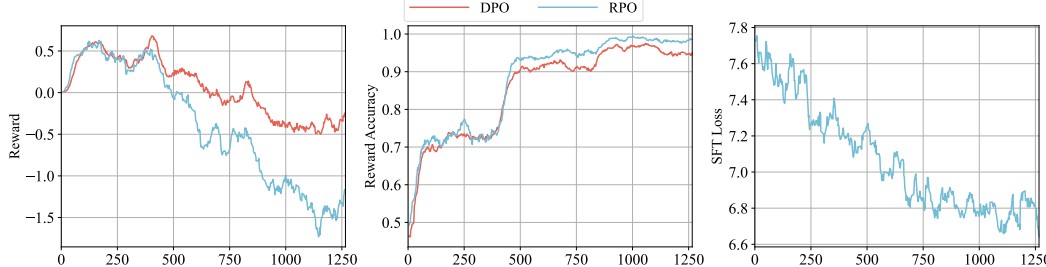

Figure 13: Training curves of DPO and RPO

Table 5: The win rate of DPO and RPO compared to other algorithms

| vs. | vs. Alpaca-7B (ref) | vs. PPO | vs. BSPO (ours) |
|-----|---------------------|---------|-----------------|
| DPO | 0.6592 | 0.6226 | 0.2627 |
| RPO | 0.7058 | 0.6664 | 0.297 |

DPO-based algorithms have also been reported to have over-optimization issues (Liu et al., 2024; Xu et al., 2024). However, the over-optimization problems of DPO-based algorithms differ from the reward over-optimization problem that our work addresses:

- Over-optimization studies for DPO focus on mitigating overfitting to the training dataset in DPO-based algorithms, which results in reduced generalization to the test set. DPO-based algorithms do not use a reward model to evaluate responses generated from exploration. Instead, the policy learns preferences solely from the "chosen" and "rejected" labels in the training dataset. Consequently, the reward over-optimization problem solved in our work does not arise in this context.

- In contrast, our work addresses a distinct challenge: in RL-based algorithms, exploration can produce OOD responses for the reward model, leading to reward overestimation and, ultimately, reward hacking.

To tackle the issue of over-optimization in DPO-based algorithm, Liu et al. (2024) proposes Regularized Preference Optimization (RPO). This method aims to identify the optimal policy by leveraging an adversarially chosen reward model that minimizes a weighted combination of its expected value and the maximum likelihood estimation (MLE) loss. The corresponding policy objective is

$$\mathcal{L}_{\text{RPO}}(\theta) = \eta\beta \cdot \mathbb{E}_{x\sim d_0, a^0\sim\pi_{\text{base}}(\cdot|x)}\left[-\log(\pi_\theta(a^0|x))\right] + \mathcal{L}_{\mathcal{D}}\left(\beta\cdot\log\left(\frac{\pi_\theta(\cdot|\cdot)}{\pi_{\text{ref}}(\cdot|\cdot)}\right)\right) \tag{59}$$

The training curves is shown in Figure 13. Although our work does not focus on the same problem, we compare the performance of the two approaches with our algorithm and baselines in Table 5.

Both DPO and RPO exhibit improvements compared to the original Alpaca-7B model, with the integration of SFT loss further enhancing their performance. The performance of PPO training is relatively poor due to severe reward over-optimization issues. In contrast, DPO and RPO do not rely on the reward model during training, thereby avoiding reward over-optimization problems and outperforming PPO. Nevertheless, because DPO and RPO lack the ability to explore new responses and depend heavily on the quality of the training dataset (Xu et al., 2024), their performance falls short of RL-based BSPO.

For this phenomenon, we believe the following is a plausible discussion for the comparison between BSPO and DPO-based methods:

- As demonstrated in Theorem 3, although we apply regularization to the value of OOD actions to avoid overestimation, BSPO can still converge to the same optimal solution for the ID region as the standard value. BSPO retains the exploration capability inherent in RL-based methods. Therefore, the comparison between BSPO and DPO-based algorithms primarily reflects the broader distinction between RL-based and DPO-based approaches.

- As reported in Xu et al. (2024), compared to RL-based methods, DPO-based methods have several limitations: over-fitting to the training dataset (a different manner of over-optimization), generating a biased policy that favors OOD responses, and sensitivity to the quality of preference data.

## D.9 OTHER DATASET

We also applied our pipeline to the AlpacaFarm dataset (Dubois et al., 2024b). Our gold reward model, based on Llama3-8B (Dubey et al., 2024), is trained on a combined preference dataset comprising UltraFeedback and a 20k preference dataset annotated by GPT-4 from AlpacaFarm. The gold model with the best performance over three training epochs is selected. For the proxy reward model, we used TinyLlama (Zhang et al., 2024a), training it on the 20k re-annotated AlpacaFarm dataset. This proxy model was then employed in the training of PPO, WCO, and BSPO. The results are presented in Figure 14.

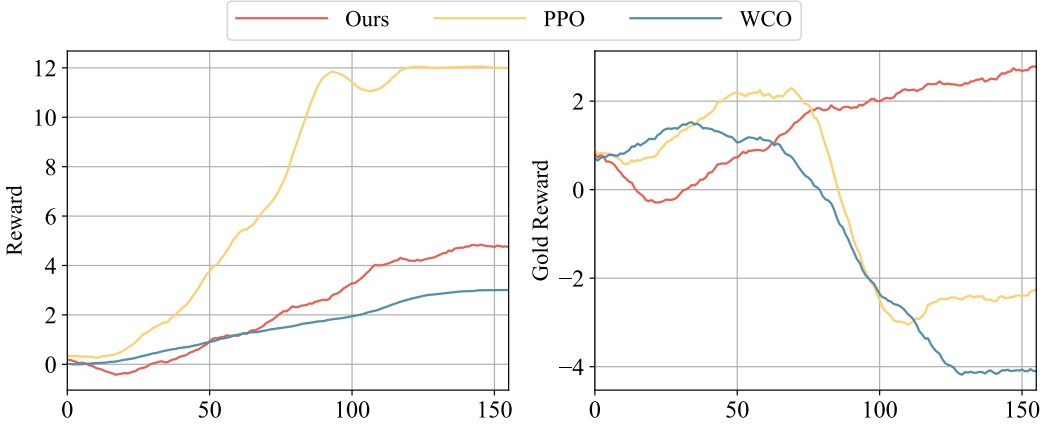

Figure 14: Training curves of different algorithms using AlpacaFarm

Due to the relatively smaller size of the AlpacaFarm dataset, the accuracy of the ScoreLM proxy model is lower compared to when using the UltraFeedback dataset. This limitation results in an earlier occurrence of the roo phenomenon in both PPO and WCO. However, only BSPO shows continuous improvement in both proxy rewards and gold rewards, highlighting its ability to mitigate over-optimization while identifying optimal policies effectively.

