# OpenReview forum: "Mitigating Reward Over-Optimization in RLHF via Behavior-Supported Regularization"
_ICLR.cc/2025/Conference — ICLR 2025 Poster_

### Official Review · Reviewer_JZFB · 2024-10-28

**Soundness:** 3
**Presentation:** 3
**Contribution:** 3
**Rating:** 6
**Confidence:** 3

**Summary:**

This paper investigates the issue of reward over-optimization in RLHF and introduces Behavior-Supported Policy Optimization (BSPO) to prevent the generation of out-of-distribution responses during RL training. By learning the value function using in-dataset actions, the author refines the policy through PPO updates. Empirical results demonstrate the effectiveness of the proposed method.

**Strengths:**

* The studied problem is important for RLHF and the proposed method is well-motivated.

* The idea of using reward model training distribution to regularize the value function learning for LLMs is novel, to the best of my knowledge.

* The empirical evidence shows promising results to alleviate the overoptimization issue.

**Weaknesses:**

Despite the strengths of this paper, I identified several limitations that require further attention:

1. The paper claims that prior constraint optimization methods "only suppress overestimation in the ID region.", which I disagree. The worst-case optimization method with ensemble has been shown to mitigate over-optimization for OOD actions [1]. During PPO training, a worst-case reward can also penalize OOD responses generated by the trained LLM. The authors should provide a more accurate comparison of BSPO with previous methods.

2. There seems to be a problem in the proof of Corollary 1. The authors suggest that optimizing the objective in Eq 23 forces the learned policy to choose behavior-support actions. However, there are cases where all actions are OOD for a given prompt $s$, which could inevitably introduce OOD actions. This situation can arise when the SFT model and reward training use different datasets. Thus, the argument may not hold.

3. The current BSPO can only be applied to token-level RLHF, while many studies focus on response-level RLHF. Can the proposed method also be adapted for response-level RLHF?

4. The evaluation is restricted to reward models smaller than 3B. Validating on 7/8B reward models would strengthen the experiments.

5. The authors should explore how the two threshold hyperparameters, $V_{min}, \epsilon_{\beta}$, affect the method's performance.

6. Prior work [2][3] considers a noisy label setting, where the over-optimization problem is more severe. Can BSPO be effective in this challenging setting?

7. Typos: In Eq 36, the use of Q function is not correct.

**References**

[1] An G, Moon S, Kim J H, et al. Uncertainty-based offline reinforcement learning with diversified q-ensemble[J]. Advances in neural information processing systems, 2021, 34: 7436-7447.

[2] Ramé A, Vieillard N, Hussenot L, et al. Warm: On the benefits of weight averaged reward models[J]. arXiv preprint arXiv:2401.12187, 2024.

[3] Yang R, Ding R, Lin Y, et al. Regularizing Hidden States Enables Learning Generalizable Reward Model for LLMs[J]. arXiv preprint arXiv:2406.10216, 2024.

**Questions:**

Please refer to the Weaknesses section.

---

> ### Author Response · Authors · 2024-11-21
> **Official Reply to Reviewer JZFB (1/5)**
>
> > **W1:** The paper claims that prior constraint optimization methods "only suppress overestimation in the ID region.", which I disagree. The worst-case optimization method with ensemble has been shown to mitigate over-optimization for OOD actions. During PPO training, a worst-case reward can also penalize OOD responses generated by the trained LLM. The authors should provide a more accurate comparison of BSPO with previous methods.
>
> **Reply to W1:** Thank you very much for highlighting this question. In fact, due to space limitations in the Introduction, we combined the descriptions of uncertainty quantifier methods and ensemble methods, which may result in a lack of clarity and potential misunderstanding. Thus, we provide a more detailed explanation here and will refine the wording in the revised manuscript to improve clarity.
>
> - Uncertainty quantifier methods[1]: These method leverage generative models to directly predict uncertainty. However, as highlighted in prior work [2], these methods face challenges in accurately predicting uncertainty in OOD regions without additional data incorporated.
> - Ensemble methods [3]: As you mentioned, ensemble methods can indeed mitigate reward over-optimization to some extent, especially in the ID region. However, recent findings [4] indicate that in the RLHF of LLMs, ensemble methods may still face challenges, such as consistent reward overestimation across multiple models, leading to over-optimization issues. **In summary, reward model ensembles mitigate but do not eliminate reward hacking [4].** Additionally, ensemble approaches necessitate training multiple branches, resulting in **higher computational and memory costs**.
>
> We hope this clarification addresses your concerns and provides a more accurate comparison. We will refine the wording in the revised manuscript to improve clarity.
> If you feel that we have not explained this point clearly, we welcome further discussion.
>
> **References:**
>
> [1] Zhang et al. Mitigating reward overoptimization via lightweight uncertainty estimation.
>
> [2] Nalisnick et al. Do deep generative models know what they don’t know?
>
> [3] Coste et al. Reward model ensembles help mitigate overoptimization.
>
> [4] Eisenstein et al. Helping or herding? reward model ensembles mitigate but do not eliminate reward hacking.

---

> ### Author Response · Authors · 2024-11-21
> **Official Reply to Reviewer JZFB (2/5)**
>
> > **W2:** There seems to be a problem in the proof of Corollary 1. The authors suggest that optimizing the objective in Eq 23 forces the learned policy to choose behavior-support actions. However, there are cases where all actions are OOD for a given prompt $s$, which could inevitably introduce OOD actions. This situation can arise when the SFT model and reward training use different datasets. Thus, the argument may not hold
>
> **Reply to W2:** Thank you for your valuable questions. We are glad to discuss this issue with you.
>
> Theoretically, **Corollary 1 is valid because the ID of the reward model is defined as the next-token distribution over the reward training dataset by a behavior policy $\beta$.**
> For a LLMs with a finite vocabulary, the condition $\sum_{a \in \mathcal{A}} \beta(a|s) = 1$ always holds.
> Consequently, for any $s \in \mathcal{S}$, there exists at least one $a$ such that $\beta(a|s) > 0$. Therefore, given a specified behavior policy, a behavior-support action that satisfies the conditions of Corollary 1 can always be identified.
>
> However, the situation you mentioned is indeed worth discussing. **In the limitations section** of our paper, we emphasized that **our algorithm primarily focuses on mitigating the issue of reward over-optimization during the RL phase, which arises due to OOD responses in reward evaluation.** Thus, if the prompt set used for RL training includes prompts outside the distribution of the reward model's training prompts, it may lead to unforeseen biases in the behavior model. Nevertheless, we remain optimistic about addressing this issue for the following reasons:
>
> - In typical scenarios, such as those described in [1][2], the prompt sets used for RL training and reward model training are carefully controlled and are drawn from the same distribution. The presence of OOD prompts during RL training negatively impacts most algorithms, making it essential to avoid such situations.
> - The behavior model is built upon a pretrained LLM that has undergone extensive pretraining on large-scale datasets and fine-tuning on the same next-token prediction task. Consequently, it exhibits a degree of generalization capability.
>
> Based on this discussion, we think this is **a good future work to explore the data distribution shift between the reward model phase and the RL phase**.
> In the revised manuscript, we will incorporate additional discussion of OOD prompts within the Limitations section. We hope our explanation addresses your concerns. Should you have any further questions, we are open to continued discussion.
>
> **References:**
>
> [1] Ouyang et al. Training language models to follow instructions with human feedback.
>
> [2] Bai et al. Training a helpful and harmless assistant with reinforcement learning from human feedback.

---

> > ### Author Response · Authors · 2024-11-21
> > **Official Reply to Reviewer JZFB (3/5)**
> >
> > > **W3:** The current BSPO can only be applied to token-level RLHF, while many studies focus on response-level RLHF. Can the proposed method also be adapted for response-level RLHF?
> >
> > **Reply to W3:** Thank you for your valuable questions. We appreciate the opportunity to engage in this discussion.
> >
> > From our perspective, token-level RLHF and response-level RLHF primarily emphasize different theoretical analysis frameworks. In practice, however, RLHF consistently utilizes response-level reward values to compute token-level advantage functions and optimizes the LLM at the token level. This is achieved using the following commonly used PPO loss function [1][2]:
> > $$
> >   \mathcal{L}\_\pi(\theta) = - \mathbb{E}\_{\tau\sim{\pi\_{\theta\_{k}}}} \left[
> >   \min\left(
> >   \rho\_t(\theta)A^{\pi\_{\theta\_{k}}}(s\_t,a\_t),
> >   \text{clip}(\rho\_t(\theta), 1-\epsilon, 1+\epsilon)A^{\pi\_{\theta_{k}}}(s\_t,a\_t)
> >   \right)
> >   \right]
> > $$
> > where $\rho\_t(\theta) = \frac{\pi\_{\theta}(a\_t|s\_t)}{\pi\_{\theta\_{k}}(a\_t|s\_t)}$, and $A^{\pi_{\theta_{k}}}$ represents the advantage function, which are calculated based on the predictions from value model $V_{\phi_{k}}$. Based on the above equation, almost all PPO-based RLHF algorithms are implemented at the token level, aligning closely with the transformer architecture's next-token prediction mechanism.
> > Consequently, **BSPO's token-level value regularization approach is inherently aligned with the training framework of PPO-based RLHF and the structure of transformer-based LLMs.** This alignment represents an advantage of BSPO.
> >
> > On the other hand, if you are referring to whether the judgment of OOD token can be extended to OOD response judgment. In theory, it is possible, because under the MDP condition, the following equation holds:
> > $$
> > \log P(y|x) = \sum_{t=1}^T \log P(a_{t:T}|a_{0:t-1},x)=\sum_{t=1}^T \log \beta(a_t|a_{0:t-1},x)
> > $$
> > However, due to the relatively long length of responses, there will be a cumulative error in practice, making it challenging to implement.
> >
> > If you have further questions, please feel free to continue the discussion.
> >
> > **References:**
> >
> > [1] Ouyang et al. Training language models to follow instructions with human feedback.
> >
> > [2] Bai et al. Training a helpful and harmless assistant with reinforcement learning from human feedback.

---

> > > ### Author Response · Authors · 2024-11-21
> > > **Official Reply to Reviewer JZFB (4/5)**
> > >
> > > > **W4:** The evaluation is restricted to reward models smaller than 3B. Validating on 7/8B reward models would strengthen the experiments.
> > >
> > > Thank you for your constructive feedback on our paper. We sincerely appreciate the opportunity to refine our work based on your suggestions.
> > >
> > > Gold models require a larger scale of parameters compared to proxy reward models to resist exploitation by OOD responses. However, the computational cost becomes prohibitive for gold models with significantly larger sizes, especially those exceeding 7B parameters.
> > >
> > > Therefore, **to evaluate the performance of different algorithms using a 7B proxy reward model, we explored using GPT-4o as evaluators.** A key issue remains: LLMs cannot provide global scalar scores in a timely manner to reflect over-optimization during training. To address this, we considered storing policy checkpoints throughout training. By comparing these checkpoints on the test set to compute the win rate and fitting an ELO score [1] as a global measure, we can effectively observe the over-optimization phenomenon.
> > >
> > > Below, we provide a summary of our implementation and results:
> > >
> > > **Implementation Details:** We train a 7B reward model (based on Alpaca-7B) on a preference dataset (UltraFeedback), achieving a test set accuracy of 82.94%.
> > > This reward model was used to conduct BSPO and baselines (PPO, WCO) training.
> > > To quantify evaluation results, we compared the checkpoints on the test set to calculate the win rate between checkpoints and fit an ELO score [1] as the scalar evaluation metric.
> > >
> > > **Experimental results:**
> > >
> > > The curves of proxy reward and elo score in training see link: [7B Proxy Reward](https://anonymous.4open.science/r/rebuttal-iclr2025-8235/human/README.md)
> > >
> > > ***Table 1: The ELO scores for different checkpoints:***
> > > | Model | Initial | PPO:79step | PPO:158step | WCO:79step | WCO:158step | BSPO:79step | BSPO:158step |
> > > | ----- | ------- | ---------- | ----------- | ---------- | ----------- | ----------- | ------------ |
> > > | ELO   | 1178.30 | 1181.77    | 1164.17     | 1201.77    | 1195.39     | 1224.92     | 1253.68      |
> > >
> > >
> > >
> > > **Our experiments reveal interesting insights:** We observed that while the proxy reward exhibits consistently increases across the three approaches, their performance under human evaluation reveals differences:
> > >
> > > - **Naïve PPO** shows performance improvement at 79 steps but suffers a severe decline by 158 steps, highlighting the issue of reward over-optimization.
> > > - **WCO** maintains stable human evaluation performance across both steps, indicating its effectiveness in mitigating over-optimization. However, it does not identify the optimal solution as effectively as BSPO.
> > > - **BSPO** demonstrates continuous improvement in both proxy reward and human evaluation, indicating its capability to address over-optimization while identifying optimal policies.
> > >
> > > We hope these experimental results address your concerns and strengthen our paper's contributions to the field of RLHF.
> > >
> > > **References:**
> > >
> > > [1] Askell et al. A General Language Assistant as a Laboratory for Alignment

---

> ### Author Response · Authors · 2024-11-21
> **Official Reply to Reviewer JZFB (5/5)**
>
> > **W5:** The authors should explore how the two threshold hyperparameters, $V_\text{min}$, $\epsilon_\beta$, affect the method's performance.
>
> **Reply to W5:** We sincerely thank you for your valuable question. Completing the experiments related to these two hyperparameters indeed enhances the contribution of our paper. Below, we provide a detailed explanation of the hyerparameter selection, experimental resulsts, and analysis.
>
> - $V\_\text{min}$:
>
> This hyperparameter is designed to suppress the $V$-values corresponding to OOD actions, necessitating that $V_\text{min}$ be lower than all possible $V$-values.
>
> In general, **$V_\text{min}$ can be determined by calculating the minimum $V$-value**, such as $V_\text{min} = \frac{1}{1-\gamma} r_\text{min}$, where $r_\text{min}$ denotes the minimum value of the reward function.
> Specifically, for RLHF of LLMs, where only the final token receives a nonzero reward, we have $V(s) = \mathbb{E}\_{\tau \sim \pi}[\gamma^{\|\tau\|} r(\tau) \mid s\_0 = s] \geq \min(0, r\_\text{min}) = V_\text{min}$.
>
> On the other hand, empirical evidence suggests that **as long as $V_\text{min}$ remains smaller than all possible $V$-values, the performance of BSPO is not sensitive to the $V_\text{min}$**.
> To evaluate this, we conducted experiments under the condition $r_\text{min} = -10$, testing a range of $V_\text{min}$ values: $V_\text{min} = -10, -15, -20, -25$. The resulting experimental outcomes are summarized in the figure below:
>
> [Robustness of Vmin](https://anonymous.4open.science/r/rebuttal-iclr2025-8235/robust/README.md)
>
> We observe that, across different values of $V_\text{min}$, the training curves of BSPO exhibit similar behavior, consistently improving both the proxy reward and the gold reward while effectively addressing the issue of over-optimization.
> These results demonstrate the robustness of the $V_\text{min}$ selection.
>
> - $\epsilon_\beta$:
>
> Since we use a parameterized model to predict the behavior policy. Thus, to prevent modeling errors, an infinitesimal value $\epsilon_\beta$ is used as a threshold, instead of 0, to determine whether a response is supported.
>
> To analyze the sensitivity of the BSPO algorithm to $\epsilon_\beta$, we conducted experiments with a range of values: $\epsilon_\beta = 1e-3, 1e-4, 1e-5, 1e-6, 1e-7$. The experimental results are presented in the figure below:
>
> [robustness of eps](https://anonymous.4open.science/r/rebuttal-iclr2025-8235/robust/README.md)
>
> Our observations indicate that for $\epsilon_\beta = 1e-4, 1e-5, 1e-6$, BSPO mitigates reward over-optimization and identifies the optimal policy. However, the performance for $\epsilon_\beta = 1e-3$ and $\epsilon_\beta = 1e-7$ shows slower improvement. This behavior may result from excessively large $\epsilon$ values, which can incorrectly classify some actions as behavior-unsupported, or excessively small $\epsilon$ values, which may fail to account for certain OOD actions.
> In conclusion, **BSPO is robust to $\epsilon_\beta$ within a reasonable range, but extreme values should be avoided.**
>
> ---
>
> > **W6:** Prior work [2][3] considers a noisy label setting, where the over-optimization problem is more severe. Can BSPO be effective in this challenging setting?
>
> **Reply to W6:** Thank you for your valuable suggestion. In response, we have conducted a new set of experiments with larger noise ratio. Specifically, we modified the training dataset of proxy model by replacing 15% of the samples with noisy data and conducted experiments under the same settings described in the paper.
>
> The results of these experiments can be found at the following link: [Noise experiment](https://anonymous.4open.science/r/rebuttal-iclr2025-8235/noise/README.md).
>
> We observed the following phenomena:
>
> - The inclusion of additional noisy data results in a decrease in the performance of all algorithms when evaluated on the gold reward, compared to the Figure 3 of the paper.
> - Despite this, BSPO consistently mitigates reward over-optimization and achieves better performance than the baseline methods, demonstrating its robustness to noise.
>
> We hope the experimental results addresses your concerns and further validate the effectiveness of BSPO.
> Should you have any additional questions, we welcome further discussion.
>
> ---
>
> > **W7:** Typos: In Eq 36, the use of Q function is not correct.
>
> Thank you very much for pointing out this typo. We will correct it in the latest manuscript.
>
> ---
>
> Finally, we express our sincere gratitude for the insightful questions and suggestions, which have been helpful to our work and will be integral to our revised manuscript. **If our efforts can address your concerns, we sincerely hope you will recognize our work.**

---

> > ### Author Response · Authors · 2024-11-24
> > **Hope to Get Your Reply**
> >
> > Dear Reviewer JZFB,
> >
> > As the deadline is nearing, we wanted to gently follow up on our recent submission. Your feedback is highly valuable to us, and we would appreciate any updates or further guidance you might have regarding our revisions and responses.
> >
> > Thank you for your time and consideration.

---

> > > ### Comment · Reviewer_JZFB · 2024-11-25
> > >
> > > I appreciate the authors' response and new experimental results. Regarding W2, I am curious whether there is any underlying assumption about state coverage in the proof; if there is, it would be better to explicitly state it. Additionally, the theoretical analysis appears to have a gap when compared to realistic situations, as we would prefer our policy to generalize to unseen prompts during PPO training to more effectively leverage the generalization capabilities of reward models.
> > >
> > > For W5, as the authors note, "for $\epsilon_\beta = 1e-4, 1e-5, 1e-6$, BSPO mitigates reward over-optimization and identifies the optimal policy." However, I have concerns that the small values of $\epsilon_\beta$ may not be robust to numerical noise, particularly during model training, which can lead to instability.

---

> ### Author Response · Authors · 2024-11-26
> **Further Discussion with Reviewer JZFB**
>
> Thank you very much Reviewer for your response. We appreciate the opportunity to address your two additional questions.
>
> > Regarding W2, I am curious whether there is any underlying assumption about state coverage in the proof; if there is, it would be better to explicitly state it. Additionally, the theoretical analysis appears to have a gap when compared to realistic situations, as we would prefer our policy to generalize to unseen prompts during PPO training to more effectively leverage the generalization capabilities of reward models.
>
> Thanks for the reminder about state coverage.
> As outlined in our Preliminaries section, in token-level MDP, the state is defined by the prompt $x \in \mathcal{X}$ and the generated action sequences $a_i \in \mathcal A, i=0,1,2,\cdots$.
> In our study, the focus is on the shift in action distribution, which leads to reward over-optimization.
> Consequently, the assumption that requires further clarification in the Analysis section is that **prompts are identically distributed during both the reward learning phase and the RL phase**.
> We will explicitly state it in latest maniscript.
>
> The reasoning behind this assumption is as follows:
>
> 1. Similar to the need for caution when addressing OOD actions, we argue that conversation data containing OOD prompts is more challenging for the reward model to evaluate accurately. Since the prompt set in RLHF is predefined and controllable, excluding OOD prompts directly is a more practical approach. Indeed, **in many RLHF implementations [1][2], the prompts used for reward learning and RL are identically distributed**—or the former explicitly includes the latter.
> 2. Throughout the paper (e.g., in the Abstract, Introduction, and Limitations sections), we clearly stated that our research focuses on the issue of reward over-optimization caused by training-time shifts in the token (action) distribution of a LLM, which results in OOD responses relative to the reward model. As noted in the Related Work section, **research on the generalizability of reward models is parallel to our work and can be used in a composite way**.
>
> Secondly, regarding the theoretical and practical gap, we believe it can be addressed for the following reasons:
>
> 1. As we mentioned above, in prior practical work [1][2], the prompts for reward learning and RL are typically i.i.d. In this case, our theoretical setting aligns with reality.
> 2. In cases where a reward model must generalize to OOD prompts, it becomes necessary to establish a mechanism that determines whether the reward model can correctly evaluate subsequent responses. Several prior studies have highlighted that the generalizability of reward models largely depends on the language capabilities of their pre-trained foundations [3][4]. Our proposed ScoreLM approach retains language capabilities while modeling behavior distribution using the same hidden state. Therefore, our method is well-positioned to complement ongoing research on reward model generalization. This could represent a promising and valuable direction for future work.
>
> **Reference**
>
> [1] Ouyang et al. Training language models to follow instructions with human feedback. OpenAI. (Section 3.2)
>
> [2] Bai et al. Training a Helpful and Harmless Assistant with Reinforcement Learning from Human Feedback. Anthropic. (Appendix B.1)
>
> [3] Lambert et al. Evaluating Reward Models for Language Modeling
>
> [4] Yang et al. Regularizing hidden states enables learning generalizable reward model for llms.
>
>
> > For W5, as the authors note, "for $\epsilon_\beta=1e-4, 1e-5, 1e-6$, BSPO mitigates reward over-optimization and identifies the optimal policy." However, I have concerns that the small values of $\epsilon_\beta$ may not be robust to numerical noise, particularly during model training, which can lead to instability.
>
> In fact, as stated in the paper (`lines 1186–1187`), **it is precisely because of numerical noise considerations that we introduce $\epsilon_\beta$ rather than strictly following the theory and comparing the probability of an action to 0**. Specifically, $\epsilon_\beta$ serves as a tolerance threshold for numerical noise in probabilistic predictions. While this approach differs from the strict theoretical formulation, it enhances the stability and reliability of the algorithm in practical applications.
>
> Furthermore, we validate its robustness through various experiments:
>
> 1. As shown in our ablation study (Figure 4(b)), the prediction of behavior-unsupported actions aligns with the occurrence of reward over-optimization.
> 2. From our experiments addressing prior W5 responses, we observe that $\epsilon_\beta$ remains robust across a sufficiently large range.
>
> ---
>
> We sincerely hope that above responses will address your concerns and help to recognize our work. If you have any further questions, we would be glad to discuss them with you

---

> > ### Comment · Reviewer_JZFB · 2024-11-28
> >
> > Thank you for your response. That makes sense to me, and I have raised my score accordingly. If space permits, the authors are expected to include some important new experiments and the state distribution assumption in the main paper for the revision.

---

> > > ### Author Response · Authors · 2024-11-28
> > > **Gratitude to Reviewer JZFB for Recognition**
> > >
> > > We sincerely appreciate your recognition of our work and the increase in our score. It is an honor for us to address your concerns, and your invaluable insights will be integrated into the final revised version. If space permits, we will include some important part of our discussion in the main text of the revision.

---

### Official Review · Reviewer_DPgD · 2024-10-29

**Soundness:** 4
**Presentation:** 3
**Contribution:** 3
**Rating:** 8
**Confidence:** 3

**Summary:**

This paper addresses reward over-optimization in Reinforcement Learning from Human Feedback (RLHF), where large language models (LLMs) sometimes align poorly with human objectives due to extrapolation errors when evaluating out-of-distribution (OOD) responses. The authors propose Behavior-Supported Policy Optimization (BSPO), which models the in-distribution (ID) region by defining a behavior policy based on the reward training dataset’s next token distribution. To regularize the value function, BSPO introduces a behavior-supported Bellman operator, penalizing OOD values while leaving ID values unaffected, thus reducing OOD response generation. BSPO is shown theoretically to guarantee monotonic improvement of the policy within the ID region, leading to convergence at the optimal policy. Empirical results confirm BSPO’s superiority over baselines in preventing reward over-optimization, making it effective in aligning LLMs with human intent.

**Strengths:**

- The research focus is clearly presented, providing readers with the essential background needed to understand the main contributions of the paper.

- The core idea of Behavior-Supported Policy Optimization (BSPO) is intuitive and easy to implement, with a strong theoretical ground covering key properties like contractivity, monotonicity, and convergence.

- The synthetic experiments are well-designed to demonstrate the rigor of the proposed method. The results are presented and interpreted clearly, making the comparisons with baseline methods easy to follow and effectively highlighting the strength of BSPO.

- The literature review is thorough, giving readers, even those unfamiliar with reward over-optimization, a clear view of the related research landscape.

**Weaknesses:**

- The experimental results presented in this paper are demonstrated exclusively on the UltraFeedback dataset, making it essential to evaluate the generalizability of the proposed method by conducting experiments on additional benchmark datasets.

- The paper emphasizes the concept of the "In-Distribution (ID) region of the reward training dataset" as a key to avoiding reward over-optimization, with BSPO’s foundational idea built on this concept. Given this emphasis, it would have been helpful to see a performance comparison with DPO-based methods, which directly utilize the ID region of reward training datasets (e.g., win response $y_w$ and lose response $y_l$), to further contextualize BSPO’s effectiveness.

- As noted in the limitations, significant differences may exist between human preferences and model-predicted preferences, underscoring the importance of human evaluation for generated responses. In cases where human judgment is not feasible, using an LLM-as-a-judge (please refer to Reference) could serve as a practical alternative for conducting pseudo-human evaluation.

- Some visual content is challenging to interpret at first glance. For instance, it seems unclear in Figure 1(c) what "correct/incorrect" specifically indicates, as this is not explicitly mentioned in the main text. Additionally, Figure 2(a) is somewhat confusing, as it places the "query+answer (given past tokens)" at the top and the "behavior distribution (next tokens)" at the bottom, unlike the conventional layout where the past context is on the bottom and the future sequence is on the top. A reversed figure layout might have improved clarity.

- Some explanations were not detailed enough. For example, it would be helpful if there were some sentences describing where the (pretrained) proxy reward model $R\_{\phi}(\cdot)$, which is optimized by $\mathcal{L}\_{\text{ScoreLM}}(\phi; \mathcal{D})$, is placed at the calculation of $\mathcal{L}\_{V}(\phi; \pi)$.

**Reference**
- Zheng, L., Chiang, W. L., Sheng, Y., Zhuang, S., Wu, Z., Zhuang, Y., ... & Stoica, I. (2023). Judging llm-as-a-judge with mt-bench and chatbot arena. Advances in Neural Information Processing Systems, 36, 46595-46623.

**Response to Rebuttal**
- I truly appreciate the authors' efforts in addressing my concerns. I will maintain my original score

**Questions:**

- In Eq (6), is the supervised loss calculated with respect to both $y_w$ and $y_l$?

- Regarding the main results in Figure 3, how were the remaining 27k samples, which were not used for training the proxy reward model, utilized? What was the rationale behind splitting the dataset into 30k and 27k?

- As noted in the Weakness section, is there a specific reason for not comparing BSPO with direct preference optimization methods like DPO, KTO, and CTO?
  - In what ways do you consider BSPO to be superior to DPO-related methods?

- In the first line of Eq (15) in the Appendix, should it be $a' \sim \pi(\cdot|s')$ instead of $a' \sim \pi(\cdot|\pi)$?

---

> ### Author Response · Authors · 2024-11-21
> **Official Reply to Reviewer DPgD (1/3)**
>
> > **W1:** The experimental results presented in this paper are demonstrated exclusively on the UltraFeedback dataset, making it essential to evaluate the generalizability of the proposed method by conducting experiments on additional benchmark datasets.
>
> **Reply to W1:** We sincerely appreciate your valuable comments, which have greatly helped us improve the paper. In response, we have conducted additional experiments using the AlpacaFarm dataset [1], as detailed in the revised manuscript. In this experiment, our gold reward model, based on Llama3-8B, is trained on a combined preference dataset consisting of UltraFeedback and a 20k preference dataset annotated by GPT-4 from AlpacaFarm. The gold model with the best performance over three training epochs is selected. For the proxy reward model, we utilized TinyLlama, training it on the 20k re-annotated AlpacaFarm dataset. This proxy model is subsequently used to train PPO, WCO, and BSPO. The results are presented in the figure below.
>
> [Alpaca Farm](https://anonymous.4open.science/r/rebuttal-iclr2025-8235/alpacafarm/README.md)
>
> Due to the relatively smaller size of the AlpacaFarm dataset, the accuracy of the ScoreLM proxy model is lower compared to when using the UltraFeedback dataset. This limitation results in an earlier occurrence of the roo phenomenon in both PPO and WCO. However, only BSPO shows continuous improvement in both proxy rewards and gold rewards, highlighting its ability to mitigate over-optimization while identifying optimal policies effectively.
>
> [1] Bubois et al. AlpacaFarm: A Simulation Framework for Methods that Learn from Human Feedback
>
> ---
>
> > **W2:** The paper emphasizes the concept of the "In-Distribution (ID) region of the reward training dataset" as a key to avoiding reward over-optimization, with BSPO’s foundational idea built on this concept. Given this emphasis, it would have been helpful to see a performance comparison with DPO-based methods, which directly utilize the ID region of reward training datasets (e.g., win response $y_w$ and lose response $y_l$), to further contextualize BSPO’s effectiveness.
> >
> > **Q3:** As noted in the Weakness section, is there a specific reason for not comparing BSPO with direct preference optimization methods like DPO, KTO, and CTO? In what ways do you consider BSPO to be superior to DPO-related methods?
>
> **Reply to W2,Q3:** We sincerely thank the reviewer for reminding us of the relationship between our work and DPO-based methods.
>
> In the initial version of the paper, we did not include DPO-based methods in the comparison because we focused primarily on RL-based algorithms.
> **While DPO-based methods also utilize preference data, they have a completely different training framework than RL-based methods.**
> As you mentioned, DPO-based methods do not use a reward model during training, relying instead on the preference dataset. Therefore, they are not subject to the reward over-optimization problem.
>
> **DPO-based vs. BSPO (RL-based)**: We believe the following is a plausible discussion for the comparison between BSPO and DPO-based methods:
>
> - As demonstrated in Theorem 3, although we apply regularization to the value of OOD actions to avoid overestimation, BSPO can still converge to the same optimal solution for the ID region as the standard value. BSPO retains the exploration capability inherent in RL-based methods. Therefore, **the comparison between BSPO and DPO-based algorithms primarily reflects the broader distinction between RL-based and DPO-based approaches.**
> - **As reported in [1], compared to RL-based methods, DPO-based methods have several limitations**: over-fitting to the training dataset (a different manner of over-optimization), generating a biased policy that favors OOD responses, and sensitivity to the quality of preference data.
>
> Based on the above differences, we did not previously discuss the DPO-based methods. However, following the reviewer’s suggestion, we recognize that it is valuable to compare BSPO with DPO-based methods. The experimental results are shown below.
>
> **Experimental Results:** The win rate of DPO results compared to other algorithms is shown in the table below:
>
> | vs. | vs. Alpaca-7B (ref.) | vs. PPO | vs. BSPO (ours) |
> | --- | -------------------- | ------- | --------------- |
> | DPO | 0.6592 | 0.6226  | 0.2627 |
>
> **Our experiments reveal interesting insights:**
>
> - Both DPO exhibit improvements compared to the original Alpaca-7B model.
> - The performance of PPO training is relatively poor due to severe reward over-optimization issues. In contrast, DPO do not rely on the reward model during training, thereby avoiding reward over-optimization problems and outperforming PPO.
> - Nevertheless, because DPO lacks the ability to explore new responses and depend heavily on the quality of the training dataset (similar to supervised learning) [1], its performance falls short of RL-based BSPO.
>
> [1] Xu et al. Is DPO Superior to PPO for LLM Alignment? A Comprehensive Study

---

> ### Author Response · Authors · 2024-11-21
> **Official Reply to Reviewer DPgD (2/3)**
>
> > **W3:** As noted in the limitations, significant differences may exist between human preferences and model-predicted preferences, underscoring the importance of human evaluation for generated responses. In cases where human judgment is not feasible, using an LLM-as-a-judge (please refer to Reference) could serve as a practical alternative for conducting pseudo-human evaluation.
>
> **Reply to W3:** Thank you for your constructive feedback on our paper. We sincerely appreciate the opportunity to refine our work based on your suggestions.
>
> Following your suggestion, we also explored using GPT-4o as evaluators. However, a key issue remains: LLMs cannot provide global scalar scores in a timely manner to reflect over-optimization during training. To address this, we considered storing policy checkpoints throughout training. By comparing these checkpoints on the test set to compute the win rate and fitting an ELO score [1] as a global measure, we can effectively observe the over-optimization phenomenon.
>
> Below, we provide a summary of our implementation and results:
>
> **Implementation Details:** Since it is challenging to assign a scalar score to a response by human evaluation, an effective approach uses comparing the generated responses from different checkpoints on a evaluation dataset.
> Specifically, we train a 7B reward model (based on Alpaca-7B) on a preference dataset (UltraFeedback), achieving a test set accuracy of 82.94%.
> This reward model was used to conduct BSPO and baselines (PPO, WCO) training.
>
> To quantify evaluation results, we compared the checkpoints on the test set to calculate the win rate between checkpoints and fit an ELO score [4] as the scalar evaluation metric. The ELO update formula is as follows:
> $$
> R'\_A = R\_A + K \cdot \left(S\_{AB} - \frac{1}{1 + 10^{(R\_B - R\_A)/400}}\right)
> $$
>
> Here, $S_{AB}$ represents the win rate of Model A over Model B on the test set, $K$ is the update coefficient, and $R_A$ and $R_B$ are the ELO scores of Models A and B, respectively.
>
> **Experimental results:**
>
> ***Table 1. Win rate between different checkpoints:***
>
>  | Win Ratio    | Initial | PPO:79step | PPO:158step | WCO:79step | WCO:158step | BSPO:79step | BSPO:158step |
>  | ------------ | ------- | ---------- | ----------- | ---------- | ----------- | ----------- | ------------ |
>  | Initial      | 0.5     | 0.4921     | 0.5316      | 0.4683     | 0.4821      | 0.4246      | 0.3849       |
>  | PPO:79step   | 0.5079  | 0.5        | 0.5258      | 0.4722     | 0.4942      | 0.4227      | 0.3952       |
>  | PPO:158step  | 0.4684  | 0.4742     | 0.5         | 0.4643     | 0.4603      | 0.4048      | 0.369        |
>  | WCO:79step   | 0.5317  | 0.5278     | 0.5357      | 0.5        | 0.5161      | 0.4724      | 0.4365       |
>  | WCO:158step  | 0.5179  | 0.5058     | 0.5397      | 0.4839     | 0.5         | 0.4782      | 0.4325       |
>  | BSPO:79step  | 0.5754  | 0.5773     | 0.5952      | 0.5276     | 0.5218      | 0.5         | 0.4484       |
>  | BSPO:158step | 0.6151  | 0.6048     | 0.631       | 0.5635     | 0.5675      | 0.5516      | 0.5          |
>
> ***Table 2: The ELO scores for different checkpoints:***
> | Model | Initial | PPO:79step | PPO:158step | WCO:79step | WCO:158step | BSPO:79step | BSPO:158step |
> | ----- | ------- | ---------- | ----------- | ---------- | ----------- | ----------- | ------------ |
> | ELO   | 1178.30 | 1181.77    | 1164.17     | 1201.77    | 1195.39     | 1224.92     | 1253.68      |
>
> The curves of proxy reward and elo score in training see link: [Non-Synthetic Setting](https://anonymous.4open.science/r/rebuttal-iclr2025-8235/human/README.md)
>
> **Our experiments reveal interesting insights:** We observed that while the proxy reward exhibits consistently increases across the three approaches, their performance under human evaluation reveals differences:
>
> - **Naïve PPO** shows performance improvement at 79 steps but suffers a severe decline by 158 steps, highlighting the issue of reward over-optimization.
> - **WCO** maintains stable human evaluation performance across both steps, indicating its effectiveness in mitigating over-optimization. However, it does not identify the optimal solution as effectively as BSPO.
> - **BSPO** demonstrates continuous improvement in both proxy reward and human evaluation, indicating its capability to address over-optimization while identifying optimal policies.
>
> We hope these experimental results address your concerns and strengthen our paper's contributions to the field of RLHF.
>
> **References:**
>
> [1] Askell et al. A General Language Assistant as a Laboratory for Alignment

---

> > ### Author Response · Authors · 2024-11-21
> > **Official Reply to Reviewer DPgD (3/3)**
> >
> > > **W4:** Some visual content is challenging to interpret at first glance. For instance, it seems unclear in Figure 1$c$ what "correct/incorrect" specifically indicates, as this is not explicitly mentioned in the main text. Additionally, Figure 2(a) is somewhat confusing, as it places the "query+answer (given past tokens)" at the top and the "behavior distribution (next tokens)" at the bottom, unlike the conventional layout where the past context is on the bottom and the future sequence is on the top. A reversed figure layout might have improved clarity.
> >
> > **Reply to W4:** Thank you very much for your valuable comments, we will refine the visual representations in the paper, especially Figure 1(c) and Figure 2(a) you mentioned:
> >
> > - Figure 1(c): We collect the responses generated during the training process and compare with reference responses. "Correct/Incorrect" indicates whether the proxy model's evaluation of a generated response aligns with the gold model, with "Correct" denoting alignment and "Incorrect" denoting otherwise. We will add this explanation into the revised manuscript.
> > - Figure 2(a): We have placed the past tokens at the bottom and the predicted behavior distribution at the top in the revised manuscript.
> >
> > ---
> >
> > > **W5:** Some explanations were not detailed enough. For example, it would be helpful if there were some sentences describing where the (pretrained) proxy reward model $R_\phi(\cdot)$, which is optimized by $\mathcal{L}_\text{ScoreLM}(\phi;\mathcal{D})$, is placed at the calculation of $\mathcal{L}_V(\varphi;\pi)$.
> >
> > **Reply to W5:** Thank you very much for pointing out this typo. In fact, Equation (7) uses $R(x, y_w; \phi)$ to calculate $\mathcal{T}^\pi_{\beta,V}$, which is then used to calculate $\mathcal L_V(\varphi;\pi)$. The current notation may have caused some confusion, and we will improve the clarity of this expression in the revised manuscript.
> >
> > ---
> >
> > > **Q1:** In Eq (6), is the supervised loss calculated with respect to both $y_w$ and $y_l$?
> >
> > **Reply to Q1:** Yes, the supervised loss is calculated with respect to both $y_w$ and $y_l$. This is because it is equally important for RL to evaluate positive and negative samples. Therefore, we want ScoreLM to model the token distribution of all responses in the training dataset of the Reward Model.
> >
> > ---
> >
> > > **Q2:** Regarding the main results in Figure 3, how were the remaining 27k samples, which were not used for training the proxy reward model, utilized? What was the rationale behind splitting the dataset into 30k and 27k?
> >
> > **Reply to Q2:** Thank you for your question; we appreciate the opportunity to clarify this point. We used 30,000 samples from the dataset to train the proxy reward model, while the full dataset of 57,000 samples was used to train the gold model. This approach was designed to intentionally limit the training data available to the proxy reward model, thereby amplifying the performance gap between the proxy and gold models. This design choice was made to make the reward over-optimization issue in RLHF training more pronounced and easier to study.
> >
> > ---
> >
> > > **Q4:** In the first line of Eq(15)in the Appendix, should it be $a'\sim\pi(\cdot|s')$ instead of $a'\sim\pi(\cdot|\pi)$?
> >
> > **Reply to Q4:** Thank you very much for pointing out this typo. We will correct it in the latest manuscript.
> >
> > ---
> >
> > Finally, thank you for your thoughtful review and for recognizing our paper. Your insights and acknowledgment are greatly appreciated and encourage our ongoing work in this area.

---

> > ### Comment · Reviewer_DPgD · 2024-11-25
> > **Response to the rebuttal**
> >
> > Thank you for addressing my concerns.
> >
> > From what I understand, the authors used Alpaca-7b as the reward model, estimated the rewards for both the baselines and your model at each checkpoint, and then compared the checkpoints to calculate the win rate and fit the ELO score. This approach is interesting and seems to provide consistent and constructive feedback.
> >
> > However, my concern remains partially unresolved. Since your ELO evaluation relies solely on Alpaca-7b, which has its own biases, it may not fully capture the nuances of real human evaluation. If possible, I encourage you to conduct ELO experiments using a variety of LLMs as reward models to ensure that BSPO's performance is consistently validated across different models.
> >
> > At any rate, I truly appreciate the effort the authors put into addressing my concern, and will maintain my original score.

---

> ### Author Response · Authors · 2024-11-25
> **Clarification of a Minor Misunderstanding**
>
> Thank you very much for your response and for recognizing the value of our work. However, we would like to clarify a minor misunderstanding. We apologize for lack of clarity in our previous reply. As outlined in our revised manuscript’s Appendix D.5, **we utilized GPT-4o, not Alpaca-7b, as the LLM for evaluation purposes**. Specifically, Alpaca-7b was used as a proxy model for training RL experiments, not for evaluation.
>
> In response to your new suggestion, we will further enhance the credibility of our experimental results by incorporating evaluations using additional LLMs. For instance, we have included an evaluation conducted with `claude-3-5-sonnet-20240620`, and the results are presented in the following table:
>
> **Table 2: The ELO scores for different checkpoints:**
> | Model                          | Initial | PPO:79step | PPO:158step | WCO:79step | WCO:158step | BSPO:79step | BSPO:158step |
> |--------------------------------|---------|------------|-------------|------------|-------------|-------------|--------------|
> | gpt-4o-2024-08-06              | 1178.30 | 1181.77    | 1164.17     | 1201.77    | 1195.39     | 1224.92     | 1253.68      |
> | claude-3-5-sonnet-20240620     | 1179.29 | 1177.63    | 1158.99     | 1206.28    | 1215.36     | 1219.20     | 1243.26      |
>
> Finally, we sincerely appreciate the time and effort you have devoted to reviewing our paper.

---

### Official Review · Reviewer_PiVm · 2024-11-02

**Soundness:** 3
**Presentation:** 3
**Contribution:** 3
**Rating:** 6
**Confidence:** 5

**Summary:**

This work addresses reward over-optimization in RLHF by penalizing out-of-distribution (OOD) responses, which are a significant source of extrapolation errors during evaluation. The proposed approach, Behavior-Supported Policy Optimization (BSPO), identifies OOD tokens by checking if the predicted probability from the reward model falls below a defined threshold. It integrates an auxiliary loss based on the supervised fine-tuning (SFT) objective into the reward model’s training. The proposed mechanism helps mitigate OOD overestimation without impacting the model’s performance on in-distribution (ID) data.

**Strengths:**

1. The paper effectively addresses a critical and well-motivated issue in RLHF, reward over-optimization due to OOD responses.

2. The proposed Behavior-Supported Policy Optimization (BSPO) method introduces a unique approach by leveraging a behavior policy for OOD detection and integrating it with value regularization.

3. The empirical validation through synthetic experiments demonstrates that BSPO outperforms baseline methods.

4. The paper’s methodology is rigorously supported by theory, with proofs ensuring the convergence and stability of the behavior-supported value functions.

**Weaknesses:**

1. A significant concern is that the experiments are conducted solely on synthetic data. This limitation raises questions about the applicability and effectiveness of the proposed method in real-world scenarios.

2. The experimental results lack an ablation study on the sensitivity of the parameter $\epsilon_\beta$.

3. The paper does not adequately discuss related works that incorporate SFT loss into the training objective, albeit for different purposes, such as in policy loss. Notable examples include:
- Provably Mitigating Overoptimization in RLHF: Your SFT Loss is Implicitly an Adversarial Regularizer
- Value-Incentivized Preference Optimization: A Unified Approach to Online and Offline RLHF

**Questions:**

1. How do you select the value of $V_{min}$ and $\epsilon_\beta$?

2. If the SFT loss is not added to the training of reward model, can the reward model still recognize OOD prompts?

---

> ### Author Response · Authors · 2024-11-21
> **Official Reply to Reviewer PiVm (1/4)**
>
> > **W1:** A significant concern is that the experiments are conducted solely on synthetic data. This limitation raises questions about the applicability and effectiveness of the proposed method in real-world scenarios.
>
> **Reply to W1:** Thank you for your constructive feedback on our paper. We sincerely appreciate the opportunity to refine our work based on your suggestions.
> Studying reward over-optimization requires continuous evaluation during the training and effective comparison with the proxy reward.
> However, human evaluation is not only expensive but also unable to provide timely scalar feedback.
> As a result, to the best of our knowledge, **nearly all prior works [1,2,3] rely on synthetic settings.**
>
> The differences between synthetic settings and real-world scenarios are an important concern. In response, we have conducted additional experiments in non-synthetic environments.
> Below, we provide a summary of our implementation and results:
>
> **Implementation Details:** Since it is challenging to assign a scalar score to a response by human evaluation, an effective approach uses comparing the generated responses from different checkpoints on a evaluation dataset.
> Specifically, we train a 7B reward model (based on Alpaca-7B) on a preference dataset (UltraFeedback), achieving a test set accuracy of 82.94%.
> This reward model was used to conduct BSPO and baselines (PPO, WCO) training.
>
> To quantify evaluation results, we compared the checkpoints on the test set to calculate the win rate between checkpoints and fit an ELO score [4] as the scalar evaluation metric. The ELO update formula is as follows:
> $$
> R'\_A = R\_A + K \cdot \left(S\_{AB} - \frac{1}{1 + 10^{(R\_B - R\_A)/400}}\right)
> $$
>
> Here, $S_{AB}$ represents the win rate of Model A over Model B on the test set, $K$ is the update coefficient, and $R_A$ and $R_B$ are the ELO scores of Models A and B, respectively.
>
> **Experimental results:**
>
> ***Table 1. Win rate between different checkpoints:***
>
>  | Win Ratio    | Initial | PPO:79step | PPO:158step | WCO:79step | WCO:158step | BSPO:79step | BSPO:158step |
>  | ------------ | ------- | ---------- | ----------- | ---------- | ----------- | ----------- | ------------ |
>  | Initial      | 0.5     | 0.4921     | 0.5316      | 0.4683     | 0.4821      | 0.4246      | 0.3849       |
>  | PPO:79step   | 0.5079  | 0.5        | 0.5258      | 0.4722     | 0.4942      | 0.4227      | 0.3952       |
>  | PPO:158step  | 0.4684  | 0.4742     | 0.5         | 0.4643     | 0.4603      | 0.4048      | 0.369        |
>  | WCO:79step   | 0.5317  | 0.5278     | 0.5357      | 0.5        | 0.5161      | 0.4724      | 0.4365       |
>  | WCO:158step  | 0.5179  | 0.5058     | 0.5397      | 0.4839     | 0.5         | 0.4782      | 0.4325       |
>  | BSPO:79step  | 0.5754  | 0.5773     | 0.5952      | 0.5276     | 0.5218      | 0.5         | 0.4484       |
>  | BSPO:158step | 0.6151  | 0.6048     | 0.631       | 0.5635     | 0.5675      | 0.5516      | 0.5          |
>
> ***Table 2: The ELO scores for different checkpoints:***
> | Model | Initial | PPO:79step | PPO:158step | WCO:79step | WCO:158step | BSPO:79step | BSPO:158step |
> | ----- | ------- | ---------- | ----------- | ---------- | ----------- | ----------- | ------------ |
> | ELO   | 1178.30 | 1181.77    | 1164.17     | 1201.77    | 1195.39     | 1224.92     | 1253.68      |
>
> The curves of proxy reward and elo score in training see link: [Non-Synthetic Setting](https://anonymous.4open.science/r/rebuttal-iclr2025-8235/human/README.md)
>
> **Our experiments reveal interesting insights:** We observed that while the proxy reward exhibits consistently increases across the three approaches, their performance under human evaluation reveals differences:
>
> - **Naïve PPO** shows performance improvement at 79 steps but suffers a severe decline by 158 steps, highlighting the issue of reward over-optimization.
> - **WCO** maintains stable human evaluation performance across both steps, indicating its effectiveness in mitigating over-optimization. However, it does not identify the optimal solution as effectively as BSPO.
> - **BSPO** demonstrates continuous improvement in both proxy reward and human evaluation, indicating its capability to address over-optimization while identifying optimal policies.
>
> We hope these experimental results address your concerns and strengthen our paper's contributions to the field of RLHF.
>
> **References:**
>
> [1] Gao et al. Scaling Laws for Reward Model Overoptimization
>
> [2] Coste et al. Reward Model Ensembles Help Mitigate Overoptimization
>
> [3] Eisenstein et al. Helping or Herding? Reward Model Ensembles Mitigate but do not Eliminate Reward Hacking
>
> [4] Askell et al. A General Language Assistant as a Laboratory for Alignment

---

> ### Author Response · Authors · 2024-11-21
> **Official Reply to Reviewer PiVm (2/4)**
>
> > **W2:** The experimental results lack an ablation study on the sensitivity of the parameter $\epsilon_\beta$.
> >
> > **Q1:** How do you select the value of $V_\text{min}$ and $\epsilon_\beta$?
>
> **Reply to W2,Q1:** We sincerely thank you for your valuable question. Complating the experiments related to these two hyperparameters indeed enhances the contribution of our paper. Below, we provide a detailed explanation of the hyerparameter selection, experimental resulsts, and analysis.
>
> - $V\_\text{min}$:
>
> This hyperparameter is designed to suppress the $V$-values corresponding to OOD actions, necessitating that $V_\text{min}$ be lower than all possible $V$-values.
>
> In general, **$V_\text{min}$ can be determined by calculating the minimum $V$-value**, such as $V_\text{min} = \frac{1}{1-\gamma} r_\text{min}$, where $r_\text{min}$ denotes the minimum value of the reward function.
> Specifically, for RLHF of LLMs, where only the final token receives a nonzero reward, we have $V(s) = \mathbb{E}\_{\tau \sim \pi}[\gamma^{\|\tau\|} r(\tau) \mid s\_0 = s] \geq \min(0, r\_\text{min}) = V\_\text{min}$.
>
> On the other hand, empirical evidence suggests that **as long as $V_\text{min}$ remains smaller than all possible $V$-values, the performance of BSPO is not sensitive to the $V_\text{min}$**.
> To evaluate this, we conducted experiments under the condition $r_\text{min} = -10$, testing a range of $V_\text{min}$ values: $V_\text{min} = -10, -15, -20, -25$. The resulting experimental outcomes are summarized in the figure below:
>
> [Robustness of Vmin](https://anonymous.4open.science/r/rebuttal-iclr2025-8235/robust/README.md)
>
> We observe that, across different values of $V_\text{min}$, the training curves of BSPO exhibit similar behavior, consistently improving both the proxy reward and the gold reward while effectively addressing the issue of over-optimization.
> These results demonstrate the robustness of the $V_\text{min}$ selection.
>
> - $\epsilon_\beta$:
>
> Since we use a parameterized model to predict the behavior policy. Thus, to prevent modeling errors, an infinitesimal value $\epsilon_\beta$ is used as a threshold, instead of 0, to determine whether a response is supported.
>
> To analyze the sensitivity of the BSPO algorithm to $\epsilon_\beta$, we conducted experiments with a range of values: $\epsilon_\beta = 1e-3, 1e-4, 1e-5, 1e-6, 1e-7$. The experimental results are presented in the figure below:
>
> [Robustness of eps](https://anonymous.4open.science/r/rebuttal-iclr2025-8235/robust/README.md)
>
> Our observations indicate that for $\epsilon_\beta = 1e-4, 1e-5, 1e-6$, BSPO mitigates reward over-optimization and identifies the optimal policy. However, the performance for $\epsilon_\beta = 1e-3$ and $\epsilon_\beta = 1e-7$ shows slower improvement. This behavior may result from excessively large $\epsilon$ values, which can incorrectly classify some actions as behavior-unsupported, or excessively small $\epsilon$ values, which may fail to account for certain OOD actions.
> In conclusion, **BSPO is robust to $\epsilon_\beta$ within a reasonable range, but extreme values should be avoided.**

---

> > ### Author Response · Authors · 2024-11-21
> > **Official Reply to Reviewer PiVm (3/4)**
> >
> > > **W3:** The paper does not adequately discuss related works that incorporate SFT loss into the training objective, albeit for different purposes, such as in policy loss. Notable examples include:
> > > - Provably Mitigating Overoptimization in RLHF: Your SFT Loss is Implicitly an Adversarial Regularizer
> > > - Value-Incentivized Preference Optimization: A Unified Approach to Online and Offline RLHF
> >
> > **Reply to W3:** We sincerely thank the reviewer for highlighting these two related works. Here and in the revised manuscript, we will provide a discussion of the use of SFT loss in these works and our approach, as well as experimental results.
> >
> > First, **the over-optimization problems of DPO-based algorithms in [1, 2] differ from the reward over-optimization problem that our work addresses**:
> > - [1] and [2] focus on mitigating overfitting to the training dataset in DPO-based algorithms, which results in reduced generalization to the test set. DPO-based algorithms do not use a reward model to evaluate responses generated from exploration. Instead, the policy learns preferences solely from the "chosen" and "rejected" labels in the training dataset. Consequently, the reward over-optimization problem solved in our work does not arise in this context.
> > - In contrast, our work addresses a distinct challenge: in RL-based algorithms, exploration can produce OOD responses for the reward model, leading to reward overestimation and, ultimately, reward hacking.
> >
> > This distinction is also discussed by [3]. Based on the above differences, we did not previously discuss this over-optimization problem of the DPO framework.
> > However, following the reviewer’s suggestion, we recognize that the similarities between the two are worth explicitly discussing in the paper.
> > Thus, we add experiments involving DPO and DPO+SFT training.
> >
> > **Implementation:** DPO+SFT is implemented by minimizing the following objective function:
> > $$
> > \mathcal{L}(\theta) = \eta\beta\cdot\mathbb{E}\_{x\sim d\_0, a^0\sim\pi^\text{base}(\cdot|x)}\left[-\log(\pi\_\theta(a^0|x))\right] + \mathcal{L}\_\mathcal{D}\left(\beta\cdot\log\left(\frac{\pi\_\theta(\cdot|\cdot)}{\pi^\text{ref}(\cdot|\cdot)}\right)\right)
> > $$
> > The objective function is derived from Equation (4.5) in [1] and aligns with the goal described in Equation (21) of [2], where $\mathcal{L}_\mathcal{D}$ represents the preference optimization loss.
> >
> > **Experimental Results:** The win rate of DPO and DPO+SFT results compared to other algorithms is shown in the table below:
> >
> > | vs.     | vs. Alpaca-7B (ref.) | vs. PPO | vs. BSPO (ours) |
> > | ------- | -------------------- | ------- | --------------- |
> > | DPO     | 0.6592               | 0.6226  | 0.2627          |
> > | DPO+SFT | 0.7058               | 0.6664  | 0.297           |
> >
> > **Our experiments reveal interesting insights:**
> >
> > - Both DPO and DPO+SFT exhibit improvements compared to the original Alpaca-7B model, with the integration of SFT loss further enhancing their performance.
> > - The performance of PPO training is relatively poor due to severe reward over-optimization issues. In contrast, DPO and DPO+SFT do not rely on the reward model during training, thereby avoiding reward over-optimization problems and outperforming PPO.
> > - Nevertheless, because DPO and DPO+SFT lack the ability to explore new responses and depend heavily on the quality of the training dataset (similar to supervised learning) [3], their performance falls short of RL-based BSPO.
> >
> > **References:**
> >
> > [1] Liu et al. Provably Mitigating Overoptimization in RLHF: Your SFT Loss is Implicitly an Adversarial Regularizer
> >
> > [2] Cen et al. Value-Incentivized Preference Optimization: A Unified Approach to Online and Offline RLHF
> >
> > [3] Xu et al. Is DPO Superior to PPO for LLM Alignment? A Comprehensive Study

---

> > > ### Author Response · Authors · 2024-11-21
> > > **Official Reply to Reviewer PiVm (4/4)**
> > >
> > > > **Q2:** If the SFT loss is not added to the training of reward model, can the reward model still recognize OOD prompts?
> > >
> > > **Reply to Q2:** We sincerely thank you for your valuable question. In most cases, the reward model will lose the ability to judge OOD actions if SFT loss is not added.
> > >
> > > The reasons are as follows:
> > > - First, although next-token prediction is an in-domain task for the language model head of ScoreLM, **it requires the SFT loss to maintain its language capabilities**. Furthermore, previous work [1] has demonstrated that maintaining these capabilities contributes to the generalization of reward models.
> > > - Secondly, we aim to evaluate whether the actions in the explored responses are OOD to the reward model. However, the distribution of the LLM backbone of the reward model may differ from the distribution of the training dataset. Thus, **additional SFT loss is necessary to fine-tune the model, ensuring its predicted distribution aligns with the training data**.
> > >
> > > Further, we verify this experimentally. It is shown in the figure below.
> > >
> > > [Without sft loss](https://anonymous.4open.science/r/rebuttal-iclr2025-8235/nosft/README.md)
> > >
> > > Our experiments reveal interesting insights:
> > >
> > > - The "Non-SFT" curve represents using ScoreLM without the SFT loss, and the results suggest that the training was notably impacted by reward over-optimization.
> > > - Furthermore, the "Original Head" curve represents training with an additional LLM (the same as the backbone of reward model) as the Behavior Model. Thus, to accurately assess OOD actions, the SFT loss is essential for aligning the predicted distribution of the behavior model with the training dataset distribution.
> > > - We also observe a decrease in the proxy model's performance. This drop may be attributed to an error in OOD judgement that impacts the ID value.
> > >
> > > **References:**
> > >
> > > [1] Yang et al. Regularizing hidden states enables learning generalizable reward model for LLMs.
> > >
> > > ---
> > >
> > > Finally, we express our sincere gratitude for the insightful questions and suggestions, which have been helpful to our work and will be integral to our revised manuscript. **If our efforts can address your concerns, we sincerely hope you will recognize our work.**

---

> > > > ### Author Response · Authors · 2024-11-24
> > > > **Hope to Get Your Reply**
> > > >
> > > > Dear Reviewer PiVm,
> > > >
> > > > As the deadline is nearing, we wanted to gently follow up on our recent submission. Your feedback is highly valuable to us, and we would appreciate any updates or further guidance you might have regarding our revisions and responses.
> > > >
> > > > Thank you for your time and consideration.

---

> > > > > ### Comment · Reviewer_PiVm · 2024-11-27
> > > > >
> > > > > Thanks for the explanations and experiments added. I raise my score after reading those.

---

> ### Author Response · Authors · 2024-11-28
> **Thank Reviewer PiVm for Approving Our Work**
>
> We sincerely appreciate your acknowledgment and are deeply encouraged by your decision to raise our rating. It is our honor to address your concerns, which have been helpful to our work and will be integral to the improvements in our final version.

---

### Official Review · Reviewer_HnWK · 2024-11-03

**Soundness:** 2
**Presentation:** 2
**Contribution:** 2
**Rating:** 6
**Confidence:** 3

**Summary:**

This paper introduces an approach named Behavior-Supported Policy Optimization (BSPO) to address the challenge of reward over-optimization in Reinforcement Learning from Human Feedback. The core issue addressed is the extrapolation error that arises when the reward model evaluates out-of-distribution responses, leading to discrepancies between the performance of LLMs under the reward model and true human objectives. The authors propose using a behavior policy to model the in-distribution region of the reward model and introduce a behavior-supported Bellman operator to regularize the value function, penalizing out-of-distribution values without impacting in-distribution ones. Theoretical proofs are provided to show that BSPO guarantees monotonic improvement of the supported policy until convergence to the optimal behavior-supported policy. Empirical results demonstrate BSPO's effectiveness in preventing reward over-optimization and finding the optimal ID policy.

**Strengths:**

1. The paper is well-organised and articulated with clarity, the complex concepts are explained in a clear and concise manner.
2. The paper provides theoretical justifications, and the results are convincing.

**Weaknesses:**

1. The paper primarily focuses on the synthetic set for evaluating reward over-optimisation, the alignment of the reward with real human annotators’ evaluation remains insufficiently validated.
2. Although the paper did experiments on the robustness of BSPO with noisy data, a more in-depth analysis of how BSPO performs under various types of noise or distributional shifts could strengthen the paper.

**Questions:**

The paper mentions that using V values instead of Q values provides greater stability, mainly because state transitions are deterministic in the context of LLMs. Specifically, for a given input prompt and a sequence of generated tokens, the generation of the next token is deterministic—given the current state and action, the next state is uniquely determined. In this case, there is a direct relationship between V values and Q values. But what if the next token generation is sampled, introducing uncertainty into the transition function? How does BSPO work in such a situation?

There is a lack of analysis of the experimental results. What causes the drop observed during the training of CPPO?

---

> ### Author Response · Authors · 2024-11-21
> **Official Reply to Reviewer HnWK (1/2)**
>
> > **W1:** The paper primarily focuses on the synthetic set for evaluating reward over-optimisation, the alignment of the reward with real human annotators’ evaluation remains insufficiently validated.
>
> **Reply to W1:** Thank you for your constructive feedback on our paper. We sincerely appreciate the opportunity to refine our work based on your suggestions.
> Studying reward over-optimization requires continuous evaluation during the training and effective comparison with the proxy reward.
> However, human evaluation is not only expensive but also unable to provide timely scalar feedback.
> As a result, to the best of our knowledge, **nearly all prior works [1,2,3] rely on synthetic settings.**
>
> The differences between synthetic settings and real-world scenarios are an important concern. In response, we have conducted **additional experiments in non-synthetic environments**.
> Below, we provide a summary of our implementation and results:
>
> **Implementation Details:** Since it is challenging to assign a scalar score to a response by human evaluation, an effective approach uses comparing the generated responses from different checkpoints on a evaluation dataset.
> Specifically, we train a 7B reward model (based on Alpaca-7B) on a preference dataset (UltraFeedback), achieving a test set accuracy of 82.94%.
> This reward model was used to conduct BSPO and baselines (PPO, WCO) training.
>
> To quantify evaluation results, we compared the checkpoints on the test set to calculate the win rate between checkpoints and fit an ELO score [4] as the scalar evaluation metric. The ELO update formula is as follows:
> $$
> R'\_A = R\_A + K \cdot \left(S\_{AB} - \frac{1}{1 + 10^{(R\_B - R\_A)/400}}\right)
> $$
>
> Here, $S_{AB}$ represents the win rate of Model A over Model B on the test set, $K$ is the update coefficient, and $R_A$ and $R_B$ are the ELO scores of Models A and B, respectively.
>
> **Experimental results:**
>
> ***Table 1. Win rate between different checkpoints:***
>
>  | Win Ratio    | Initial | PPO:79step | PPO:158step | WCO:79step | WCO:158step | BSPO:79step | BSPO:158step |
>  | ------------ | ------- | ---------- | ----------- | ---------- | ----------- | ----------- | ------------ |
>  | Initial      | 0.5     | 0.4921     | 0.5316      | 0.4683     | 0.4821      | 0.4246      | 0.3849       |
>  | PPO:79step   | 0.5079  | 0.5        | 0.5258      | 0.4722     | 0.4942      | 0.4227      | 0.3952       |
>  | PPO:158step  | 0.4684  | 0.4742     | 0.5         | 0.4643     | 0.4603      | 0.4048      | 0.369        |
>  | WCO:79step   | 0.5317  | 0.5278     | 0.5357      | 0.5        | 0.5161      | 0.4724      | 0.4365       |
>  | WCO:158step  | 0.5179  | 0.5058     | 0.5397      | 0.4839     | 0.5         | 0.4782      | 0.4325       |
>  | BSPO:79step  | 0.5754  | 0.5773     | 0.5952      | 0.5276     | 0.5218      | 0.5         | 0.4484       |
>  | BSPO:158step | 0.6151  | 0.6048     | 0.631       | 0.5635     | 0.5675      | 0.5516      | 0.5          |
>
> ***Table 2: The ELO scores for different checkpoints:***
> | Model | Initial | PPO:79step | PPO:158step | WCO:79step | WCO:158step | BSPO:79step | BSPO:158step |
> | ----- | ------- | ---------- | ----------- | ---------- | ----------- | ----------- | ------------ |
> | ELO   | 1178.30 | 1181.77    | 1164.17     | 1201.77    | 1195.39     | 1224.92     | 1253.68      |
>
> The curves of proxy reward and elo score in training see link: [Non-Synthetic Setting](https://anonymous.4open.science/r/rebuttal-iclr2025-8235/human/README.md)
>
> **Our experiments reveal interesting insights:** We observed that while the proxy reward exhibits consistently increases across the three approaches, their performance under human evaluation reveals differences:
>
> - **Naïve PPO** shows performance improvement at 79 steps but suffers a severe decline by 158 steps, highlighting the issue of reward over-optimization.
> - **WCO** maintains stable human evaluation performance across both steps, indicating its effectiveness in mitigating over-optimization. However, it does not identify the optimal solution as effectively as BSPO.
> - **BSPO** demonstrates continuous improvement in both proxy reward and human evaluation, indicating its capability to address over-optimization while identifying optimal policies.
>
> We hope these experimental results address your concerns and strengthen our paper's contributions to the field of RLHF.
>
> **References:**
>
> [1] Gao et al. Scaling Laws for Reward Model Overoptimization
>
> [2] Coste et al. Reward Model Ensembles Help Mitigate Overoptimization
>
> [3] Eisenstein et al. Helping or Herding? Reward Model Ensembles Mitigate but do not Eliminate Reward Hacking
>
> [4] Askell et al. A General Language Assistant as a Laboratory for Alignment

---

> > ### Author Response · Authors · 2024-11-21
> > **Official Reply to Reviewer HnWK (2/2)**
> >
> > > **W2:** Although the paper did experiments on the robustness of BSPO with noisy data, a more in-depth analysis of how BSPO performs under various types of noise or distributional shifts could strengthen the paper.
> >
> > **Reply to W2:** Thank you for your valuable suggestion. In response, we have conducted a new set of experiments with larger noise ratio. Specifically, we modified the the training dataset of proxy model by replacing 15% of the samples with noisy data and conducted experiments under the same settings described in the paper.
> >
> > The results of these experiments can be found at the following link: [Exp](https://anonymous.4open.science/r/rebuttal-iclr2025-8235/noise/README.md).
> >
> > We observed the following phenomena:
> >
> > - The inclusion of additional noisy data results in **a decrease in the performance of all algorithms** when evaluated on the gold reward, compared to the Figure 3 of the paper.
> > - Despite this, **BSPO consistently mitigates reward over-optimization and achieves better performance than the baseline methods**, demonstrating its robustness to noise.
> >
> > We hope the experimental results addresses your concerns and further validate the effectiveness of BSPO.
> > Should you have any additional questions, we welcome further discussion.
> >
> > ---
> >
> > > **Q1:** The paper mentions that using V values instead of Q values provides greater stability, mainly because state transitions are deterministic in the context of LLMs. Specifically, for a given input prompt and a sequence of generated tokens, the generation of the next token is deterministic—given the current state and action, the next state is uniquely determined. In this case, there is a direct relationship between V values and Q values. But what if the next token generation is sampled, introducing uncertainty into the transition function? How does BSPO work in such a situation?
> >
> > **Reply to Q1:** Thank you for your question. However, we believe there is a misunderstanding that requires clarification.
> >
> > In our paradigm, **the next token $a$ given the current state $s$ is still sampled rather than deterministic**; that is, $a\sim\pi(\cdot|s)$. Therefore, our BSPO algorithm is applicable to common scenarios in LLM training.
> >
> > In the paper, **the "determinism" we referred to is that given a state $s=a_{0:t-1}$ and the next token $a_t$, the next state $s'=a_{0:t}$ is determined**, as noted in `lines 314-315` of the manuscript.
> > It is important to clarify that this determinism inherently holds due to the autoregressive nature of next-token prediction in LLMs. As a result, the relationship between Q-values and V-values in the MDP can be simplified as follows:
> > $$
> > Q(s,a)=\mathbb{E}_{s'\sim P(\cdot|s,a)}[r(s, a)+\gamma V(s')]=r(s,a)+\gamma V(s'),
> > $$
> > where $P$ denotes the state transition probability. For LLMs, $P(s'|s,a)=1$ if and only if $s'=s\cup a$. It can be observed that $V(s')$ can be directly converted to $Q(s, a)$ without the need for sampling. Based on this finding, we can predict behavior-supported V-values instead of Q-values to achieve greater stability while maintaining equivalent policy evaluation.
> >
> > We hope this explanation resolves your concerns. We will also revise unclear parts of the manuscript. If you have further questions, please feel free to continue the discussion.
> >
> > ---
> >
> > > **Q2:** There is a lack of analysis of the experimental results. What causes the drop observed during the training of CPPO?
> >
> > **Reply to Q2:** Thank you very much for helping us improve our work. We will carefully address your concerns.
> > In our experiments, we strictly followed the implementation outlined in the Constrained PPO [1], which uses a constraint to keep the proxy reward below an empirical threshold where reward over-optimization begins.
> > In the experimental results, we observed that while the proxy reward of CPPO training continuously increases, its gold reward actually declines. We believe the potential reasons for this phenomenon are as follows:
> >
> > 1. During CPPO training, although the constraints successfully maintain the proxy reward near the threshold, the gold reward continues to change. This suggests that training checkpoints with the same proxy reward could correspond to different gold rewards. Therefore, **relying solely on the proxy reward may be insufficient** to establish a threshold that indicates when over-optimization occurs.
> > 2. Lowering the constraint threshold could mitigate the decline in the gold reward. However, **it may also reduce the search space**, leading to suboptimal outcomes.
> >
> > We hope the explanation above addresses your concerns. We will include this analysis in the revised manuscript.
> >
> >
> > [1] Moskovitz et al. Confronting Reward Model Overoptimization with Constrained RLHF
> >
> > ---
> >
> > Finally, we express our sincere gratitude for the insightful questions and suggestions, which have been helpful to our work and will be integral to our revised manuscript. **If our efforts can address your concerns, we sincerely hope you will recognize our work.**

---

> > > ### Author Response · Authors · 2024-11-24
> > > **Hope to Get Your Reply**
> > >
> > > Dear Reviewer HnWK,
> > >
> > > As the deadline is nearing, we wanted to gently follow up on our recent submission. Your feedback is highly valuable to us, and we would appreciate any updates or further guidance you might have regarding our revisions and responses.
> > >
> > > Thank you for your time and consideration.

---

> > > > ### Comment · Reviewer_HnWK · 2024-11-26
> > > > **Official Comment by Reviewer**
> > > >
> > > > The equation $Q(s,a) = r(s,a) + \gamma V(s^\prime)$ is clear, thank you for the clarification. However, I still have concerns regarding the claim of greater stability. Given the large action space (next token), $r(s,a)$ still makes it difficult to see how this approach would offer additional stability. This reasoning does not convince me that it provides additional benefits.
> > > >
> > > > In summary, I choose to maintain the current score.

---

> ### Author Response · Authors · 2024-11-26
> **Clarify Misunderstandings and Desire for Re-assessment**
>
> Thank you for your response. However, it seems there is still a misunderstanding that we would like to address. **Since you did not previously express concerns regarding "additional stability," we were unaware that this aspect required further clarification.**
>
> The term "additional stability" refers to the fact that the V-value estimation is more stable than the Q-value estimation in the actual implementation. **In fact, it is precisely because LLM has a large action space that the use of V value is more favorable.**
>
> The reasons for this are as follows:
>
> 1. **Lower Complexity:** V-value focuses solely on the expected return of a state, whereas Q-value includes the additional complexity of evaluating state-action pairs. LLMs typically operate in massive action spaces，and most of the actions are difficult to access on the same state, which makes the data for training Q-values insufficient.
>
> 2. **Lower Variance:** Estimating Q-values can involve higher variance because it depends on both the immediate reward and the optimality of subsequent actions ***[1]***. V-values, which summarize the expected return from a state without needing action-specific granularity, tend to have lower variance, making them more stable during training. However, Q-values estimate rely on specific actions that may be sampled infrequently, leading to higher variance due to fewer samples.
>
> 3. **Easy to Integrate:** In RLHF, human feedback is often used to train a reward model that evaluates states (e.g., whether a generated response aligns with human preferences). A state-based value function (V-value) naturally aligns with this reward model, simplifying integration and improving performance.
>
> As a result, the RLHF ***[2][3][4][5]*** for almost all LLMs uses V network as the value function estimate. We do not cite this as a reason why we outperform the baselines algorithm; in fact, we simply give matching implementations of BSPO that use the V-value version.
>
>
> Finally, we wish to express our heartfelt regret that a misunderstanding overshadowed our hard work. **We sincerely hope that this response will encourage a fresh re-evaluation to our work, and we extend our deepest gratitude for your consideration.**
>
>
> ---
>
> **Reference:**
>
> [1] Sutton, et al. Reinforcement Learning: An Introduction (2nd ed.).
>
> [2] Ouyang et al. Training language models to follow instructions with human feedback. OpenAI.
>
> [3] Bai et al. Training a Helpful and Harmless Assistant with Reinforcement Learning from Human Feedback. Anthropic.
>
> [4] Kaufmann et al. A Survey of Reinforcement Learning from Human Feedback
>
> [5] Casper et al. Open Problems and Fundamental Limitations of Reinforcement Learning from Human Feedback

---

> > ### Author Response · Authors · 2024-12-01
> > **Sincere Request for Feedback**
> >
> > Dear Reviewer HnWK,
> >
> > As the review deadline approaches, I would like to reach out once more and apologize for any inconvenience caused by the timing of this message. We sincerely appreciate the time and effort you have devoted to reviewing our paper.
> >
> > During the rebuttal period, we conducted additional experiments, including tests in non-synthetic settings and noise experiments, and provided detailed responses to the concerns you raised. These updates will be included in the revised paper.
> >
> > It seems that you found our initial response satisfactory but raised an additional concern. We find this might be due to a misunderstanding, and provide detailed clarification. **We would greatly appreciate it if you could take the time to review our clarification and provide feedback.** Since we have not received further feedback, we are concerned that the misunderstanding may still exists. **Should you wish for us to clarify any further points, we would be happy to do so and ensure they are addressed in the revised version.**
> >
> > We greatly value your thoughtful review and remain committed to addressing your feedback thoroughly.  We sincerely hope that the efforts we have made during the rebuttal phase are acknowledged by you.

---

> > ### Comment · Reviewer_HnWK · 2024-12-02
> > **Still Confusing**
> >
> > I’ve changed my score, but I’m still confused. Since the input to Q and V neural network are exactly the same because $(s,a)=s\cup a=s'$ , I think that learning Q and V are essentially doing the same thing.

---

> ### Author Response · Authors · 2024-12-02
> **Gratitude to Reviewer HnWK for Supporting Our Work**
>
> We sincerely thank you for recognizing our work and for raising our score. It is an honor for us to address your concerns, and your valuable feedback will be incorporated into the final revised version.
>
> Regarding your concerns about the Q and V critic networks, **we will revise the manuscript to provide a more precise description**, such as clarifying their equivalence in BSPO during LLM training. It is important to note that this part pertains to the implementation of our algorithm. **We do not cite it as a reason for outperforming baseline algorithms; rather, we simply present matching implementations of BSPO using the V-value version.**
>
> Once again, we greatly appreciate the time and effort you have dedicated to reviewing our paper. Your insightful comments are highly valuable and will significantly contribute to improving our work.

---

### Author Response · Authors · 2024-11-22
**General Comments to All Reviewers**

We thank the reviewers (Reviewer HnWK, Reviewer PiVm, Reviewer DPgD, Reviewer JZFB) for their valuable feedback.

We are encouraged that the reviewers found that our BSPO is a **novel** (Reviewer JZFB) and **unique** (Reviewer PiVm) approach which **effectively addresses a critical and well-motivated issue** (Reviewer PiVm, Reviewer JZFB); our methodology is **rigorously and strongly supported by theory** (Reviewer PiVm, Reviewer DPgD) which is **convincing** (Reviewer HnWK); our experiments are **well-designed** and **effectively highlighting the strength of BSPO** (Reviewer PiVm, Reviewer DPgD, Reviewer JZFB); our paper is **well-organised and articulated with clarity** (Reviewer HnWK, Reviewer DPgD) and has **thorough literature review** (Reviewer DPgD).

We addressed all the reviewer comments below and will incorporate them into the revision. The revised version primarily includes the following significant updates, with the modified sections marked in **orange**:

- In Appendix D.5, we conduct experiments in **a more realistic (non-synthetic) setting** (Reviewer HnWK, PiVm-W1, and DPgD-W3)
- In Appendix D.6, we conduct experiments to demonstrate **robustness of label noise** (Reviewer HnWK and JZFB).
- We **refine the writing** of the equivalent use of  V-value function (Reviewer HnWK).
- We include **more result anlysis of CPPO** (Reviewer HnWK).
- In Appendix D.7, we include **ablation studies of $V_{min}$ and $\epsilon_\beta$** (Reviewer PiVm and JZFB).
- In Appendix D.8, we include **comparison with DPO-based algorithms** (Reviewer PiVm and DPgD).
- In Appendix D.9, we conduct **experiments on an additional dataset** (Reviewer DPgD).
- We **refine the visual representations** of Figure 1(c) and Figure 2(a) (Reviewer DPgD).
- We **refine explanations of some annotations** (Reviewer DPgD).
- In Introduction Section, We provides **a more accurate expression of enssembles** (Reviewer JZFB-W1)
- In Limitation Section, we include **discussion about OOD prompts** (Reviewer JZFB-W2).
- We fixed some **typo** (Reviewer DPgD-Q4 and JZFB-W7)

If our rebuttal addresses the concerns, we earnestly and kindly ask the reviewers to consider raising the rating and supporting us for acceptance.

---

### Meta-Review · Area_Chair_vA9a · 2024-12-23

**Metareview:**

This paper develops an approach Behavior-Supported Policy Optimization (BSPO) to address the challenge of reward over-optimization by using a behavior policy to model the in-distribution region of the reward model and introduce a behavior-supported Bellman operator to regularize the value function, penalizing out-of-distribution values without impacting in-distribution ones.

The reviewers all liked the paper as is evidenced by all positive scores. As a result, I recommend acceptance of this paper. For suggestions, I would recommend doing some experiments on non-subjective benchmarks such as Math and coding tasks to strengthen the paper.

**Additional Comments On Reviewer Discussion:**

The reviewers all were convinced of the value of the paper after discussion with authors.

---

### Decision · Program_Chairs · 2025-01-22

Accept (Poster)